# Scaling Test-Time Compute Without Verification or RL is Suboptimal

**Amrith Setlur** [1]  **Nived Rajaraman** [2]  **Sergey Levine** [2]  **Aviral Kumar** [1]

## Abstract

Despite substantial advances in scaling test-time compute, an ongoing debate in the community is how it should be scaled up to enable continued and efficient improvements with scaling. There are largely two approaches: (i) distilling successful search or thinking traces; and (ii), using verification (*e.g.*, 0/1 outcome rewards, or verifiers) to guide reinforcement learning (RL) and search algorithms. In this paper, we prove that finetuning LLMs with verifier-based (VB) methods based on RL or search is far superior to verifier-free (VF) approaches based on distilling or cloning search traces, given a fixed amount of compute/-data budget. Further, we show that as we scale test-time compute (measured as the output token length) and training data, suboptimality of VF methods scales poorly compared to VB when the base pre-trained LLM presents a heterogeneous distribution over correct solution traces (*e.g.*, different lengths, styles, etc.) and admits a non-sharp distribution over rewards on traces sampled from it. We formalize this condition using anti-concentration (Erdős, 1945), implying a stronger result that VB methods scale better *asymptotically*, with the performance gap between VB and VF widening as test-time budget grows. We corroborate our theory empirically on didactic and math reasoning problems with 3/8/32B-sized pre-trained LLMs, where we find verification is crucial for scaling test-time compute.

## 1. Introduction

Pre-training and post-training of LLMs rely heavily on access to high-quality "expert" data, but it is projected that by 2028, the availability of such data on the Internet will diminish (Villalobos et al., 2022; Liu et al., 2024), and improving model performance on several domains (*e.g.*, reasoning, safety, *etc.*) often requires more data (Li et al., 2024). As a

[1]Carnegie Mellon University [2]UC Berkeley. Correspondence to: Amrith Setlur <asetlur@cs.cmu.edu>.

*Proceedings of the 42nd International Conference on Machine Learning*, Vancouver, Canada. PMLR 267, 2025. Copyright 2025 by the author(s).

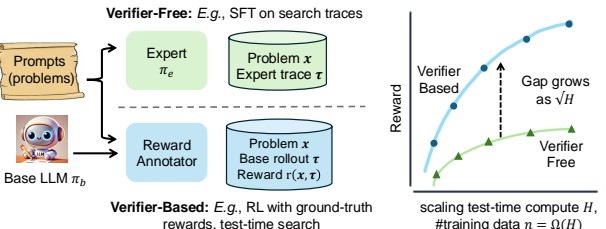

Figure 1: **Scaling test-time compute**: Given a set of problems, verifier-free (VF) methods query expert traces, whereas verifier-based (VB) methods collect reward annotations for rollouts from the base LLM. Crucially, one aims to mimic "good" traces and the other seeks to improve via access to verification. We prove a $\sqrt{H}$ gap between a simple VB method and *any* VF method as we scale data $n$ and compute $H$, and verify this in practice.

result, scaling test-time compute is emerging as an alternate paradigm for improving reasoning performance, where an LLM is made capable of executing search or refinement procedures, either implicitly by training for it, or by explicitly executing the search on top of the LLM outputs. The goal here is to search over responses for a test query, which naturally leads to traces much longer than a direct answer. Broadly, we can classify the set of prevalant approaches for scaling test compute into two categories (see Figure 1).

The first class uses some form of *verification*, *e.g.*, a 0/1 outcome reward or a verifier, for test-time search (Cobbe et al., 2021b) or reinforcement learning (RL), *e.g.*, Deepseek R1 (DeepSeek-AI et al., 2025) uses outcome reward. The second class of approaches circumvents verification altogether and runs supervised fine-tuning on "expert" traces, obtained either by piecing together tree search (Gandhi et al., 2024; Moon et al., 2024), or by querying bigger models trained to generate longer traces, *i.e.*, distillation approaches (*e.g.*, S1 (Muennighoff et al., 2025), OpenThinker (Team, 2025), *etc.*). We refer to these as *"verifier-free"*, as these methods do not query any verification signal for guiding learning. Despite the prevalence of both classes, *i.e.*, verifier-based (VB) and verifier-free (VF), it is not clear which class results in better test-time scaling, especially when the amount of test-time compute increases. We theoretically and empirically show that when fine-tuning a pre-trained LLM, VB methods are expected to perform better.

***Theoretically, we show in this paper that VB methods outperform VF methods as we scale data and compute.*** To do so, we operate in a setting where we are given a base

LLM $\pi_b$ and a dataset of problems. We represent the total available test-time compute in terms of the total number of tokens $H$ that can be used to produce a solution. Our goal is to finetune $\pi_b$ to make efficient use of test-time compute, *i.e.*, attain best performance within a given compute budget. For learning, VF methods are allowed to obtain at most $n$ correct solution traces for these problems by querying an expert (*e.g.*, humans, linearized search (Gandhi et al., 2024), *etc.*). On the other hand, VB methods are allowed to query a reward annotator that measures correctness of a given response on $n$ samples generated from $\pi_b$ but never observes expert traces. When operating in this setup (which reflects practical scenarios), we prove that as we scale training data $n$ and test-time compute budget $H$, the separation between a simple verifier-based approach based on RL, and *any* verifier-free method grows. Since both class of approaches involve finetuning $\pi_b$, the properties of the pre-trained $\pi_b$ plays a key role in our separation result, as we discuss below.

***What properties of the pre-trained LLM enable VB methods to scale better than VF methods?*** To evaluate the performance of different approaches across varying test-time compute budgets, we define rewards that are high when the LLM arrives at the correct solution, and does so without completely exhausting the provided token budget. We show that when the base LLM admits a sufficiently *heterogeneous* distribution over rewards (*i.e.*, it admits coverage over multiple correct sequences of varying lengths for a given problem), then scaling test-time compute by running *any* VF approach is suboptimal. In contrast, under a specific form of heterogeneity (which is also generally satisfied in practice), the performance of the policy obtained by running RL with verifiers (either implicit 0/1 "regex" matching rewards or explicitly trained numerical, generative verifiers) is better at any given test-time compute budget, and, moreover, also scales better as we further increase test-time compute. We term this specific form of heterogeneity as *anti-concentration*: for a given problem, a base LLM is said to be anti-concentrated if it admits non-trivial probability mass over solution traces that achieve rewards slightly better than the mean reward for that problem under the base policy. In Figure 3 we illustrate these two properties: heterogeneity and anti-concentration. Given a problem, if the base LLM samples correct responses of varying response length (heterogeneity), but also samples solution traces that are rewarded higher than the mean reward on that problem (anti-concentration); we prove that VB methods asymptotically dominate VF methods as we proportionately scale both training data $n$ and the test-time compute $H$.

***Overview of how we show the separation between VB and VF.*** For our main result, we first prove an information-theoretic lower bound showing that the suboptimality of *any* VF algorithm scales as the heterogeneity or diversity of the base policy being finetuned, which implies a subop-

timality gap of $\Omega(H/\sqrt{n})$ in the worst case. For this we build on second-order suboptimality bounds in Foster et al. (2024a). That said, we then show that the suboptimality for a simple VB method that runs RL with a trained verifier or actual 0/1 outcome rewards scales as $\mathcal{O}(H/n)$. In fact, we show that the heterogeneity of the base policy is often helpful for VB approaches, as long as the heterogeneity also implies anti-concentration, *i.e.*, there is a reasonable chance to sample traces with rewards slightly better than the mean reward attained under the base policy. Consequently, the performance difference between VB and VF methods scales with the horizon or test compute $H$. This implies the need for training verifiers, running RL, or at the very least, using rewards when finetuning LLMs for test-time scaling.

***Empirical results corroborating theory.*** We corroborate our theoretical results on math reasoning with 3B/8B Llama models, and the S1 (Muennighoff et al., 2025) model. For the S1 model that is trained in a verifier-free manner, we show that a simple verifier-based approach performs better than S1 across a set of test-time compute budgets (Figure 6). For the LLama models, we explicitly control the heterogeneity of the base LLM and show that VF methods perform poorly with more heterogeneous base LLMs, and that the gap between VB and VF performance scales with more test-time compute (Figure 5 in Section 7). Our investigation also reveals that common pre-trained LLMs are indeed heterogeneous and satisfy anti-concentration, which are abstractions we introduce to prove our theoretical results (Figure 7, 8). To the best our knowledge, this is the first theoretical result and systematic study showing a separation between VF and VB methods, under realistic assumptions on the base model.

***Implications and takeaways.*** Our work presents several implications for practitioners. As long as the pre-trained LLM is sufficiently hetergeneous and anti-concentrated, which we verify are both satsified in practice for models we evaluate, our analysis implies: (i) While recent results imply that both verifier-free and verifier-based methods can work well at a large compute budget, we show that VB methods scale better as the test-time budget increases; (ii) This gap can be amplified further when the number of prompts for VB finetuning also scales linearly with the allowed test-time token budget. Another implication of our results is that since VF methods work well when heterogeneity is small, this means that in order for them to scale well, pre-trained LLMs must also exhibit low heterogeneity, but we believe that this is rarely the case with modern LLMs in practice.

## 2. Related Work

**Scaling test-time compute.** Recent works (Sardana et al., 2023; Snell et al., 2024) show that scaling test-time compute can improve performance at rates faster than scaling data (Li et al., 2024) or model size (Hoffmann et al., 2022), either by **(i)** training an LLM to implement a search (Yao et al., 2023;

Gandhi et al., 2024) or refinement (Kumar et al., 2024; Qu et al., 2024) procedure in its long chain-of-thought; or **(ii)** wrapping LLMs with search procedures (Wu et al., 2024; Beeching et al., 2024) that use trained verifiers (Cobbe et al., 2021a; Setlur et al., 2024b) as cost functions for search. We do not study this distinction in this paper, but rather focus on an orthogonal axis that can be used to separate test-time methods: whether or not there is access to a reward or verification signal during training and/or inference.

**Verifier-based (VB) algorithms.** Several works use 0/1 "outcome" reward annotations (Uesato et al., 2022; Bi et al., 2024) for training LLM reasoners. Some also use trained verifiers (Hosseini et al., 2024) or run search test-time search (Welleck et al., 2024; Chen et al., 2024b). More recently, using 0/1 verification signals for RL (Kimi-Team, 2025; DeepSeek-AI et al., 2025) has shown impressive results. Other algorithms use verification by converting 0/1 rewards into a value function (Rafailov et al., 2023; Zelik-man et al., 2022; Singh et al., 2023; Setlur et al., 2024a). Verification can be done generatively (Zhang et al., 2024) and implicitly (Yuan et al., 2024), all within one single LLM alternating between generation and verification. We bucket all of these as VB methods and show that querying verifiers or rewards is critical for scaling test-time compute.

**Verifier-free (VF) algorithms.** With the goal of distilling search procedures multiple works (Yang et al., 2022; Xie et al., 2024; Lehnert et al., 2024; Gandhi et al., 2024), SFT pre-trained LLMs on search traces (Gandhi et al., 2024; Nie et al., 2024) that all succeed eventually. This is done so that the LLM learns to search for solutions on test problems (Sel et al., 2023). These methods are reward-free and are thus forced to mimic heterogeneous search traces with varying token counts. This makes generalization difficult for *any* SFT method, leading to poor test-time scaling, aligning with prior findings (Kumar et al., 2024; Xiang et al., 2025).

## 3. Notation and Preliminaries

**Notation.** We use the usual $\mathcal{O}/\Omega$ notation, where $a = \widetilde{\mathcal{O}}(b)$ when $a = \mathcal{O}(b \cdot \max(1, \text{polylog}(b)))$, and $a \lesssim b$ for $a = \mathcal{O}(b)$. The set of integers $\{1, \ldots, n\}$ is denoted as $[n]$. For a set $\mathcal{S}$, the set of all measures over $\mathcal{S}$ is given by $\Delta(\mathcal{S})$.

**Preliminaries.** Following prior work (Kazemnejad et al., 2024; Setlur et al., 2024a) we model language generation as a token-level Markov decision process (MDP): $\mathcal{M}(\mathcal{S}, \mathcal{A}, r, H)$, with state space $\mathcal{S}$, token space $\mathcal{A}$, binary reward $r : \mathcal{S} \times \mathcal{A} \mapsto \{0, 1\}$ in class $\mathcal{R}$, and horizon (token budget) $H$. Let $\mathcal{S}_h$ denote the set of states at time $h$ (so, $\mathcal{S} =: \cup_{h=1}^{H} \mathcal{S}_h$). The set of initial states $\mathcal{S}_1$ is the set of input problems $\mathcal{X} \ni \mathbf{x}$, sampled from a distribution $\rho$. At time $h$, state $\mathbf{s}_h$ is given exactly by the concatenation of the problem $\mathbf{x}$ and the sequence of tokens sampled till step $h - 1$, *i.e.*, $\mathbf{s}_h = (\mathbf{x}, a_1, \ldots, a_{h-1})$; upon producing

token $a_h$ the environment deterministically transitions to state $\mathbf{s}_{h+1} = (\mathbf{s}_h, a_h)$ obtained by concatenation and collects reward $r_h =: r(\mathbf{s}_h, a_h)$. A policy $\pi \in \Pi$ is a function $\pi_h : \mathcal{S} \mapsto \Delta(\mathcal{A})$ which produces a distribution over tokens at each state. We use $d_h^\pi$ to denote the distribution over $\mathcal{S}_h$ induced by $\pi$. A *solution trace* is a rollout $\tau = (\mathbf{x}, a_1, \ldots a_H)$ in the MDP, and $r(\tau) = \sum_h r(\mathbf{s}_h, a_h)$. We use the notation $\mathbb{E}_{\rho, \pi}[\cdot]$ to denote the expectation $\mathbb{E}_{\mathbf{x} \sim \rho}[\mathbb{E}_{\tau \sim \pi(\cdot | \mathbf{x})}[\cdot]]$.

## 4. Effectively Scaling Test-Time Compute

Our goal is to compare methods that finetune LLMs to most efficiently scale test-time compute. We say that an algorithm is effective at making consistent use of test compute if it attains the best performance possible within a fixed compute budget. In practice, this means that an approach must strike a balance between directly "guessing" an answer, which uses the least number of tokens but is unlikely to succeed, and re-attempting sequentially (*i.e.*, run linearized search), which is less token efficient and wastes compute, but is more likely to succeed at least once. This entails a procedure where models are deployed with an ever-growing upper bound on test-time token budgets in hopes to find more successes for a given prompt, underscoring the necessity of efficient asymptotic scaling as we formalize in this section.

Denoting a base LLM as an autoregressive policy $\pi_b(a|\mathbf{s})$ and a given budget on test-time compute represented in terms of a maximum $H$ output token length, we evaluate a finetuning algorithm by measuring the performance of the policy produced by finetuning $\pi_b$ under a specific reward function $r(\mathbf{s}, a)$. This reward function should capture both the accuracy and the efficiency of attaining the solution. One such family of reward functions is a ***bi-level reward***.

**Bi-level reward.** As discussed in Property 4.1, we say that a reward function is a bi-level reward when on any given trajectory, the reward remains 0 until it reaches a state corresponding to the correct solution, at which point it receives a reward of 1 (for the first time), and then continues to collect 1s in the future no matter what it samples (Figure 2). That is, once the LLM generates a correct solution, it continues to attain high rewards. For a solution trace $\tau = (\mathbf{x}, a_1, \ldots a_H)$ we define the reward $r(\tau) =: \sum_{h=1}^{H} r(\mathbf{s}_h, a_h)$, and the performance (expected reward) of $\pi$ is $J_r(\pi) =: \mathbb{E}_{\rho, \tau}[r(\tau)]$. A *correct trace* $\tau$ is one that gets the answer correct at some point within the budget of $H$ tokens, *i.e.*, $r(\tau) > 0$. To maximize efficiency, we want $r(\tau)$ to be as high as possible in the distribution of the test problem, denoted $\rho$, *i.e.*, $\max_\pi J_r(\pi)$ where $J_r(\pi) =: \mathbb{E}_{\rho, \pi} \sum_{t=0}^{H} r(\mathbf{s}_t, a_t)$. The $Q$-value is defined as : $Q_\pi(\mathbf{s}_h, a_h) =: \mathbb{E}_{\rho, \pi} \left[ \sum_{t=h}^{H} r(\mathbf{s}_t, a_t) \mid \mathbf{s}_h, a_h \right]$.

> **Property 4.1** (Bi-level rewards)**.** For any trajectory $\tau$, rewards are binary and non-decreasing, *i.e.* $\forall h \in [H]$, $r_{h+1}(\mathbf{s}_{h+1}, a_{h+1}) \geq r_h(\mathbf{s}_h, a_h)$, (example in Figure 2).

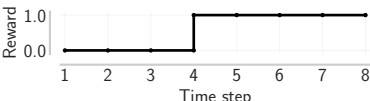

Figure 2: *Example of bi-level rewards:* After the first step where reward is 1, irrespective of future actions reward remains 1.

**Asymptotic test-time compute efficiency.** Having defined how we can measure the efficacy of a finetuning algorithm in scaling test-time compute within a budget of $H$ tokens, we now turn to providing a formal definition that allows us to compare different fine-tuning algorithms. Concretely, Definition 4.2 defines what it means for an algorithm to *"asymptotically"* scale test-time compute by $H^\alpha$, compared to another algorithm. Under our bi-level reward formulation, a higher value of $\alpha$ implies that algorithm $\mathcal{A}_1$ is able to arrive at the correct answers spending $\approx H^\alpha$ less compute on average compared to $\mathcal{A}_2$, as we scale $H$. In the next section, we show that verifier-based algorithms scales test compute by $\tilde{\Omega}(H)$ compared to veriifer-free algorithms.

---

**Definition 4.2** (Scaling test-time compute by $H^\alpha$). Fix any bi-level reward $r$, base policy $\pi_b$, horizon $H$ and data budget $n = \Omega(H)$, we say that algorithm $\mathcal{A}_1$ producing policy $\mathcal{A}_1(H)$, *asymptotically* scales test-time compute by $H^\alpha$ compared to $\mathcal{A}_2$ producing $\mathcal{A}_2(H)$ if:
$$J_r(\mathcal{A}_1(H)) - J_r(\mathcal{A}_2(H)) = \tilde{\Omega}(H^\alpha).$$

---

## 5. Theory: When Does Verification Enable Asymptotic Scaling of Test Compute?

In this section, we theoretically compare *verifier-free* and *verifier-based* algorithms when scaling test-time compute. We show that for any bi-level reward, there are base policies (pre-trained LLMs) that enable verification based algorithms to asymptotically scale test-time compute $H$, by a factor of $\Omega(\sqrt{H})$ relative to *any* verifier-free approach, and quantify these properties of the pre-trained base LLM.

A *verifier-free (VF) algorithm* finetunes the base LLM $\pi_b$ to mimic data from an expert policy $\pi_e$ without using any rewards or verification. The expert $\pi_e$ can produce a solution trace that directly results in the final correct answer (Zelikman et al., 2022) or perform a number of search and/or backtracking operations to eventually end in the final correct answer (Gandhi et al., 2024). The expert policy samples correct traces $\tau$, *i.e.* $r(\tau) > 0$, however these traces are not guaranteed to be the most compute-efficient (i.e., $r(\tau) \neq H$) as each one may get to the answer spending varying number of tokens for search, backtracking, and CoT.

The performance of any verifier-free algorithm is dependent on the **choice of the expert**. So, how do we choose "good" experts for learning? Such experts must satisfy two conditions: **(a)** they should attain high rewards (end in a correct final answer), and **(b)** the expert's distribution should be at least somewhat "close" to the base policy $\pi_b$ to prevent

issues such as memorization and optimization pathologies from finetuning (Kang et al., 2024; Tajwar et al., 2024). For e.g., one predominant way of constructing expert data is to first sample multiple traces from $\pi_b$ and then retain all correct traces (Zelikman et al., 2022; Gulcehre et al., 2023). While existing theoretical abstractions do not prescribe an ideal condition to quantify **(b)**, we formalize this practical constraint by constraining the expert to be the highest reward policy in $\Pi_\kappa$: the set of all policies with $\chi^2$ divergence $\leq \kappa$ w.r.t. the base $\pi_b$. We choose $\chi^2$ over other f-divergences like KL for simplicity (Huang et al., 2024b).

$$D_{\chi^2}(\pi_e \| \pi_b) =: \mathbb{E}_{\rho,\pi_b}\left[\left(\frac{\pi_e(\tau \mid \mathbf{x})}{\pi_b(\tau \mid \mathbf{x})} - 1\right)^2\right] \leq \kappa. \quad (1)$$

We refer to the $\kappa$-$\chi^2$ ball of expert policies as $\Pi_\kappa$, and the optimal expert, i.e., $\arg\max_{\pi \in \Pi_\kappa} J_r(\pi)$, as $\bar{\pi}_\kappa$.

A *verifier-based (VB) algorithm* is one that finetunes the base policy without an expert, but instead queries an annotator to provide rewards to solution traces sampled from $\pi_b$. *E.g.*, RL with outcome rewards (DeepSeek-AI et al., 2025) or using generative verifiers (Zhang et al., 2024) count as VB methods. Note that this definition does *not* necessarily require a learned verifier. In all, these classes of methods *differ in the learning signal being ued*: access to an expert policy vs. access toa a bi-level reward annotator.

We compare VF and VB methods, given $n$ rollouts sampled from expert policy for VF methods and $n$ base policy rollouts with reward annotations for VB. We are interested in evaluting whether VB methods scale test-time compute better than VF as per Definition 4.2. Our main theoretical result, Theorem 5.1, states that for *any* bi-level reward function, there exist base policies $\pi_b$, representative of pre-trained LLMs, where a simple VB method scales test-time compute better than *all* VF methods by at least $\Omega(\sqrt{H})$.

---

**Theorem 5.1** (Main result; informal). *For any bi-level reward $r$ and sufficiently large data budget $n$, there exists a base policy $\pi_b$, verifier-based algorithm $\mathcal{A}$, such that finetuning $\pi_b$ with $\mathcal{A}$ scales test-time compute (Definition 4.2) by $\tilde{\Omega}(\sqrt{H})$ relative to any verifier-free algorithm.*

---

**Key insight.** To prove the above, we establish an instance-dependent information-theoretic lower bound on the suboptimality gap of *any* VF method, which is $H/\sqrt{n}$ when $\pi_b$ is sufficiently *heterogeneous*, *i.e.*, solution traces for a given prompt vary a lot in token efficiency. Then, we show that a simple verifier-based method attains a suboptimality gap upper bound of only $H/n$, even when $\pi_b$ is heterogeneous. For this, $\pi_b$ should cover some high-reward traces with a sufficient probability. Formally, when the distribution over rewards attained by traces sampled from $\pi_b$ is heterogeneous and not too "sharply" concentrated around its mean and $n = \Omega(H)$ (typically the case for best performance),

VB methods scale test-time efficiency by $\sqrt{H}$ over VF methods. A pictorial illustration of these conditions is shown in Figure 3, which we also show holds empirically (Section 7). Then, we use techniques from second-order adaptive bounds to develop an analysis for proving the separation result.

### 5.1. Lower Bounds for Verifier-Free Expert Cloning

We first derive an information-theoretic lower bound for VF methods comparing them to the expert policy $\pi_e$. To understand the implications of our theoretical result, we state our lower bound using a notion of "*base policy heterogeneity*", which measures the variability in the token sequences that all attain the same final answer under $\pi_b$. We define this notion of policy heterogeneity as follows:

---

**Property 5.2** (Policy heterogeneity). Given problem $\mathbf{x}$, the heterogeneity of $\pi \in \Pi$ at $\mathbf{x}$ is given by:

$$\sigma_{\pi,\mathbf{x}}^2 =: \sum_{h=1}^H \mathbb{E}_{\mathbf{s}_h \sim d_h^\pi} \left[ \mathrm{Var}_{a \sim \pi(\cdot|\mathbf{s}_h)} \left[ Q^{\pi_e}(\mathbf{s}_h, a_h) \right] \mid \mathbf{x} \right].$$

Total heterogeneity across problems is $\sigma_\pi^2 =: \mathbb{E}_{\mathbf{x} \sim \rho} \left[ \sigma_{\pi,\mathbf{x}}^2 \right]$, the median is $\widetilde{\sigma}_\pi := \mathrm{Median}(\{\sigma_{\pi,\mathbf{x}} : \mathbf{x} \in \mathcal{X}\})$, and the mean across problems is $\bar{\sigma}_\pi = \mathbb{E}_{\mathbf{x} \sim \rho} \left[ \sigma_{\pi,\mathbf{x}} \right]$.

---

For the expert policy, heterogeneity is non-zero when different solution traces spend different tokens and token budgets to attain the final answer from *any* state-action tuple attained in a trajectory. We expect most practical LLM finetuning datasets obtained by rejection sampling, concatenating search traces, collecting human thinking trace data, or distilling from larger models to induce quite a heterogeneous expert, since a high diversity of solution traces is often a desideratum employed by practitioners when generating training data in supervised finetuning (Chen et al., 2024a). In order to obtain heterogeneous expert traces, we would also need the base policy $\pi_b$ to be heterogeneous. In fact, we show a useful result relating heterogeneity of $\pi_e$ to that of $\pi_b$, which allows us to present our lower bound directly in terms of $\sigma_b$ of the base policy (instead of $\sigma_e$).

**Lemma 5.3** (Lower bound on expert heterogeneity). *Let the heterogeneity of base policy $\pi_b$ be $\sigma_b^2$. For any expert $\pi_e \in \Pi_\kappa$, its heterogeneity $\sigma_e^2$ satisfies $|\sigma_e^2 - \sigma_b^2| \le H \sigma_b \sqrt{\kappa/2}$.*

**Reading Theorem 5.4.** This bound means that a dataset of $n$ datapoints from a fixed expert $\pi_e$ is fundamentally insufficient to resolve uncertainty as to which of the experts $\in \Pi'$ was the one generating the data. Using a verifier-free algorithm here incurs a suboptimality that depends on $\widetilde{\sigma}_e/\sqrt{n}$ for the worst choice of this expert in $\Pi'$, meaning that for any given reward function, base policy and expert, there is a problem instance $\Pi'$ and an alternate reward $r'$ where this guarantee is tight. Further, when the total heterogeneity $\sigma_e^2 \approx \bar{\sigma}_e^2$, then we can replace the median with the expert's heterogeneity $\sigma_e$, which is in turn close to the base LLM's heterogeneity $\sigma_b$, as stated by our Lemma 5.3.

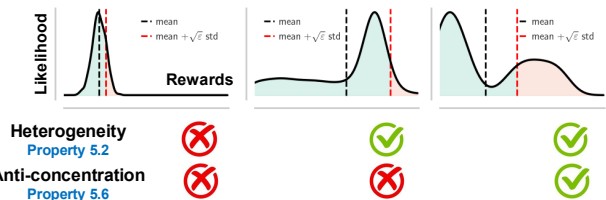

Figure 3: *Illustration of properties of the base model $\pi_b$ that enable VB methods to outperform VF methods:* heterogeneity (Property 5.2) and anti-concentration (Property 5.6).

---

**Theorem 5.4** (**Information-theoretic lower bound on verifier-free algorithms**). *Given any $\rho, r, \pi_b$, expert policy $\pi_e$ and $k \le |\mathcal{X}|/4$, there exists a family of alternate expert policies $\Pi'$ of size $2^k$ and reward class $\mathcal{R}' \subseteq \mathcal{R}$, s.t., for any $\hat{\pi}_n^{\mathrm{vf}}$ returned by any verifier-free algorithm:*

$$\max_{\pi' \in \Pi'} \max_{r' \in \mathcal{R}'} J_r(\pi') - J_{r'}(\hat{\pi}_n^{\mathrm{vf}}) = \Omega\left( \widetilde{\sigma}_e \sqrt{\frac{\log |\Pi'|}{n}} \right),$$

*$\forall \pi' \in \Pi'$, $\sigma_{\pi'}^2 = O(\sigma_e^2)$ under any alternate reward function $r' \in \mathcal{R}'$, and $\Pi' \subseteq \Pi_{\Theta(\kappa)}$. Finally, when the total hetetogeneity $\sigma_e^2 \le (6/5)\bar{\sigma}_e^2$, then the median heterogeneity across problems $\widetilde{\sigma}_e$ scales as $\Omega(\sigma_e)$, i.e., we can replace $\tilde{\sigma}_e$ in the bound above with $\sigma_e$.*

---

To prove this result, we extend the lower bound result from Foster et al. (2024a), which applies to only one prompt, to an instance-dependent lower bound that applies to a setting with more than one prompt and bi-level rewards. See Appendix C.3 for a formal statement and a proof. This result implies that it is challenging to clone highly heterogeneous experts: when $\widetilde{\sigma}_b$ scales as $\Omega(H)$, the bound grows as $\Omega(H/\sqrt{n})$. A linear dependence on horizon is unavoidable, even though the transition dynamics in this problem are trivial (i.e., just concatenation) and the transitions are known. The one scenario where this bound can be reasonable is when $\widetilde{\sigma}_b$ is small, but this is rarely the case in practice because pre-trained LLMs tend to be quite heterogeneous. Due to pathologies from training on narrow data, users prefer using more heterogeneous base models and experts.

### 5.2. A Simple Verifier-Based Algorithm

So far, we saw that heterogeneity can hurt the performance of any VF algorithm that uses expert data without reward annotations. Next, we show that this limitation does not exist for VB methods, by constructing a simple algorithm that trains a verifier using $n$ reward annotations on data sampled from the base policy $\pi_b$ (which need not be an expert). Concretely, our algorithm first trains a verifier to predict sparse $0/1$ correctness of a given solution trace using the provided data, to the best possible accuracy. Then, it finetunes the LLM to maximize the verifier scores on the distribution of problems $\rho$. Note, that at test-time we sample problems *i.i.d.* from $\rho$. Crucially, the algorithm does not assume access to ground-truth rewards over the entire distribution $\rho$, but only a small training dataset $\mathcal{D}_{\mathrm{tr}}$. We present this approach for-

mally in Algorithm 1. In particular, Step 2 produces a class of verifiers $\hat{\mathcal{R}}_\gamma$ that are $\gamma$-optimal as measured by squared loss. Step 3 produces a policy that performs optimally on the worst reward in $\hat{\mathcal{R}}_\gamma$. This technique of optimizing a pessimistic reward is common in both theory and practice of offline RL (Wang et al., 2024), and has also been useful for preventing reward overoptimization (Coste et al., 2024).

In practice, algorithms that use some sort of verification, either train a policy on ground-truth 0/1 rewards on a fixed set of training problems, *e.g.*, in R1 (DeepSeek-AI et al., 2025), DeepScaleR (Luo et al., 2025); or train outcome/process verifiers to predict ground-truth rewards on the same training problems, and use the verifiers at test time to run search over responses sampled from the base policy given test problems, *e.g.*, best-of-N (Cobbe et al., 2021a), beam search (Beeching et al., 2024; Snell et al., 2024; Setlur et al., 2024b). The former trains a policy that generalizes onto test problems, and the latter trains a verifier with the expectation that the verifier's predictions are accurate on the test problems. Algorithm 1 below falls in the latter category of verification-based approaches. Next, we show that this VB

---

**Algorithm 1** Simple Verifier-Based Algorithm

**Require:** Base policy $\pi_b$, dataset $\mathcal{D}_{\mathrm{tr}} =: \{(\mathbf{x}_i, \tau_i)\}_{i=1}^n$ of problems $\mathbf{x}_i \sim \rho$ and traces $\tau_i \sim \pi_b(\cdot \mid \mathbf{x})$.

1: For every $\tau_i$ annotate $(\mathbf{x}_i, \tau_i)$ with bi-level reward $y_i$.
2: Learn set of classifiers $\hat{R}_\gamma \subset \mathcal{R}$ that are $\gamma$-optimal, *i.e.*,

$$\hat{R}_\gamma =: \left\{ r' \in \mathcal{R} \,\middle|\, \frac{1}{n} \sum_{i=1}^n (r'(\tau_i) - r(\tau_i))^2 \leq \gamma \right\}$$

3: Return any optimal pessimistic verifier-based policy,

$$\hat{\pi}_n^{\mathrm{vb}} \in \arg\max_{\pi \in \Pi} \min_{r \in \hat{R}_\gamma} J_r(\hat{\pi}).$$

---

algorithm attains a lower suboptimality gap. To do so, we first prove an intermediate Lemma 5.5, which upper bounds the accuracy of the verifier trained on $\mathcal{D}_{\mathrm{tr}}$ in Algorithm 1.

**Proposition 5.5** (Verifier accuracy). *For any bi-level reward $r$, base policy $\pi_b$, and learned reward function $\hat{r} \in \hat{R}_\gamma$ from Algorithm 1, with probability $1 - \delta$, the following error bound is true:* $\mathbb{E}_{\rho, \pi_b} [|r(\tau) - \hat{r}(\tau)|] \leq \widetilde{\mathcal{O}}_H \left( H \cdot \log(\frac{|\mathcal{R}|}{\delta})/n \right)$.

Equipped with this result, we can now bound the suboptimality of the learned policy $\hat{\pi}_n^{\mathrm{vb}}$ in Algorithm 1. We show that when used with a specific subset of heterogeneous $\pi_b$–which are empirically show are representative of real pre-trained LLMs–this VB algorithm attains a stronger suboptimality guarantee of $H/n$, when compared to the best policy $\bar{\pi}_\kappa$ belonging to the $\chi^2$-ball, $\Pi_\kappa$, around the base policy. Intuitively, this subset of heterogeneous policies are characterized by a condition pertaining to how concentrated or "sharp" is the distribution of rewards induced by sampling

traces from $\pi_b$ on a given prompt. We call this the **anti-concentration** condition. As long as the reward distribution puts a constant probability mass on reward values that are $\approx \sigma_\mathbf{x} \sqrt{\kappa}$ higher than the mean reward $\pi_b$ gets on prompt $\mathbf{x}$, we say that the policy is *anti-concentrated* (Property 5.6; an illustration of this condition is shown in Figure 3).

---

**Property 5.6** (*Anti-concentrated $\pi_b$*). For problem $\mathbf{x}$, horizon $H$, and base policy $\pi_b$, let $c_\mathbf{x}(\varepsilon)$ be the probability mass that reward $r(\tau)$ is larger than the mean $E_{\tau \sim \pi_b(\cdot|\mathbf{x})}[r(\tau)]$ by a margin of $\sigma_{b,\mathbf{x}} \sqrt{\varepsilon}$.

$$c_\mathbf{x}(\varepsilon) =: \Pr_{\tau \sim \pi_b(\cdot|\mathbf{x})}\Big[r(\tau) \geq \mathbb{E}_{\tau \sim \pi_b(\cdot|\mathbf{x})}[r(\tau)] + \sigma_{b,\mathbf{x}} \sqrt{\varepsilon}\Big],$$

The base LLM $\pi_b$ is said to be $c$-anticoncentrated if $\min_\mathbf{x} c_\mathbf{x}(\kappa_\mathbf{x}) \geq c$, where $\kappa_\mathbf{x} =: D_{\chi^2}(\bar{\pi}_\kappa(\cdot|\mathbf{x}) \| \pi_b(\cdot|\mathbf{x}))$ and $\bar{\pi}_\kappa$ denotes the best policy in $\Pi_\kappa$ (highest performance). The value of $\kappa_\mathbf{x}$ depends on how much an expert is allowed to deviate from $\pi_b$ on problem $\mathbf{x}$.

---

Even under high heterogeneity (Property 5.2), an anti-concentrated $\pi_b$ covers–with a constant mass–a policy that improves over its own mean. This means that an algorithm using the reward signal to fine-tune $\pi_b$ should be able to discover this "better" policy. VF algorithms that do not utilize to the reward signal fail at finding this high-rewarding policy. While a non-heterogeneous base policy (for e.g., one that always samples a single trace for a given $\mathbf{x}$) will not satisfy Property 5.6, hetergeneous distributions can easily be anti-concentrated since heterogeneity is a property of a moment (i.e., variance) of the reward distribution whereas Property 5.6 fundamentally relates to the shape or the CDF of the reward distribution. We demonstrate in our experiments that pre-trained LLMs often satisfy this property.

***How can VB algorithms benefit from anti-concentration of $\pi_b$?*** As discussed above, Property 5.6 ensures the existence of a good policy that is covered by the base policy, with high probability. Intuitively, running RL should be able to then sample traces that attain high rewards and learn to pick up on this reward with more training. From a theoretical perspective, note that the suboptimality gap of any VB method depends on the distribution shift between the data-generating policy ($\pi_b$ in our case) and the comparator policy that we wish to provide the guarantee against ($\bar{\pi}_\kappa$), since this shift dictates how generalization error during training gets amplified when the model is deployed. This notion of distribution shift is typically formalized as a bounded *coverage coefficient* (Rashidinejad et al., 2021) of an unknown comparator policy, which is restrictive. We strengthen the notion of coverage coefficient by leveraging anti-concentration, which allows us to optimally construct a high-reward comparator policy covered by the base policy.

Note that our simple VB method admits no direct dependency in $\sigma_b$ (base policy's heterogeneity), which scales as

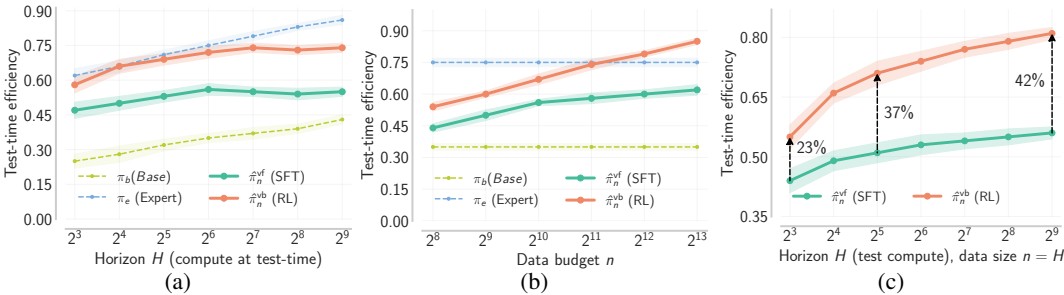

Figure 4: *Contextualized planted subsequence:* We setup a heterogeneous base policy $\pi_b$, and induce an expert by rejection sampling correct traces from $\pi_b$. (a) Fixing data size at $2^{10}$ we scale test compute, training separate SFT, RL policies for each compute budget. (b) For a fixed compute budget of $2^6$ we scale data, and train a set of SFT and RL policies for each $n$. In (a), (b) we find RL scales both data and test-time compute efficiency over SFT. In (c) we scale both test compute and training data and note that the gap between the performance of RL and SFT grows super linearly, as predicted by our result in Theorem 5.8.

$\Omega(H)$ in the worst case. This implies that as long as $\pi_b$ satisfies Property 5.6 for some $h_0 \ll H$, VB methods only incur suboptimality that scales as $O(1)$ when $n = \Omega(H)$ whereas for any VF method this is $\Omega(\sqrt{H})$. Mathematically, this is because once Property 5.6 is satisfied for some $c_0$ at a given horizon $h_0$, then it continues to hold for $c_0$ and $\forall H > h_0$. This is a consequence of the structure of the bi-level reward as we show in Lemma C.22 in Appendix C.4.

> **Theorem 5.7** (Suboptimality upper bound for VB against any expert). *Consider a bi-level reward $r$, base policy $\pi_b$ that is $c_0$-anticoncentrated at some horizon $h_0 \leq H$. Then, w.p. $1 - \delta$, for the policy $\hat{\pi}_n^{\mathrm{vb}}$ returned by Algorithm 1, the suboptimality gap w.r.t. the best expert: $\bar{\pi}_\kappa$:*
>
> $$J_r(\bar{\pi}_\kappa) - J_r(\hat{\pi}_n^{\mathrm{vb}}) \lesssim \frac{1}{c_0} \cdot \frac{H \log(|\mathcal{R}|/\delta)}{n}.$$

**Overall**, Theorem 5.7 implies that if $\pi_b$ covers some correct solution traces for a given prompt, then VB methods can find these traces and minimize suboptimality, whereas VF methods may not be able to discover them and will try to mimic multiple traces, which also naturally increases the chances of failing at the problem. Combining the upper and lower bounds (Theorem 5.7 and 5.4) allows us to bound the efficacy of test-time scaling with VB and VF methods.

**Theorem 5.8** (*Separation between test-time scaling of VB and VF methods*). *For any heterogeneous $\pi_b$ with $\tilde{\sigma}_b = \Omega(H)$, and is $c_0$-anticoncentrated for horizon $h_0 \ll H$, the policy $\hat{\pi}_n^{\mathrm{vb}}$ returned by the simple verifier-based Algorithm 1 and $\hat{\pi}_n^{\mathrm{vf}}$ returned by any verifier-free method satisfy:*

$$J_r(\hat{\pi}_n^{\mathrm{vb}}) - J_r(\hat{\pi}_n^{\mathrm{vf}}) = \tilde{\Omega}\left(H/\sqrt{n}\right),$$

*which implies our test-time scaling result in Theorem 5.1.*

> **Takeaways: Verification enables test-time scaling**
>
> VF suffer when the base policy is heterogeneous. VB outperform *any* VF algorithm given that the base policy is heterogeneous and the induced reward distribution is anti-concentrated.

*Remark* 5.9 (***VB methods improve over VF by solving more***

*problems*). We first note that the performance of the best expert policy $\bar{\pi}_\kappa$ which belongs the the $\kappa \chi^2$-ball around $\pi_b$ will only continue to improve as a function of $H$ by solving more questions (*i.e.*, by finding new $\mathbf{x}$ where $r$ flips from 0 to 1). To see why, note that $\mathrm{D}_{\chi^2}$ grows in $H$ meaning that at large $H$, the expert $\bar{\pi}_\kappa$ cannot be simultaneously close to the base policy and maximize reward unless it solves new questions too. Now note that if $n = \Omega(H)$ samples are used for training, then the VB algorithm attains a suboptimality of $\mathcal{O}(1)$ compared to this best expert, but the VF algorithm still suffers from a horizon-dependent suboptimality (if $\pi_b$ is heterogeneous). This means that if the suboptimality gap with respect to the best expert continues to be $\mathcal{O}(1)$ as we increase the token budget, then we are solving harder problems, not just being more efficient on easier problems.

# 6. Illustrating Theory in Practice

Our theoretical results in Section 5 show that when the base policy is heterogeneous, VF approaches perform poorly. However, this can still be favorable for VB Algorithm 1, as long as the anti-concentration condition (Property 5.6) holds. We now use a didactic setting representative of typical LLM reasoning problems to validate our theoretical results, and study real math reasoning problems in the next section.

**Didactic setup.** We extend the planted subsequence problem from Setlur et al. (2024b) to a contextual version. Concretely, for an input problem $\mathbf{x} = (x_1, .., x_5)$, we say that a response $\mathbf{y}$ with $H$ tokens is a correct trace if there exists a *gold* contiguous subsequence $(g(x_1), .., g(x_5))$ planted in $\mathbf{y}$. Here, the underlying mapping $g : [10] \mapsto [30]$ is fixed but unknown. For a state $\mathbf{s} =: (\mathbf{x}, a_1, .., a_h)$, the bi-level reward $r(\mathbf{s}) = 1$ if and only if there exists some $h' \leq h$ such that the last 5 tokens before $h'$ match the gold subsequence. To use the same scale to compare methods trained for different horizon $H$ values (test-time budget), we $J_r(\pi)$ and divide it by the maximum reward of $H - 4$. Details in Appendix D.

**Base policy.** We wish to construct base policies $\pi_b$ that: **(i)** differ in heterogeneity, and **(ii)** satisfy the anti-concentration condition. To do so, we finetune GPT2-xl (Radford et al.,

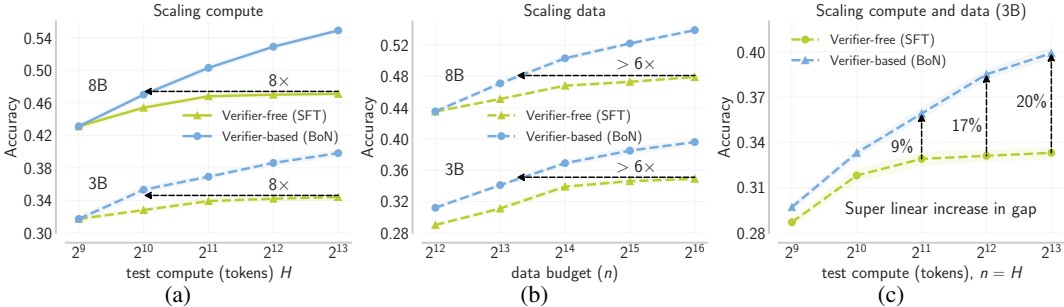

Figure 5: **Scaling test compute $H$ and training data $n$ on MATH:** We compare two common algorithms for spending test compute: (i) verifier-free SFT on manually stitched sequential revisions (Qu et al., 2024; Muennighoff et al., 2025) from an expert, and (ii) BoN (Cobbe et al., 2021a) search using a verifier trained on base LLM. In (a), we scale $H$, with data size $n=2^{14}$, and find BoN scales test-compute by $8\times$ over SFT. In (b), we fix $H=2^{12}$, scale $n$, and note the $6\times$ gain in sample efficiency for BoN. In (c), we compare RL and SFT following Definition 4.2 where we scale both $n$ and $H$, and corroborating Theorem 5.8 the gap between RL and SFT grows super linearly.

2019) on samples obtained from a mixture of hand-designed "procedural" policies. Inspired from Setlur et al. (2024b), a procedural policy $\mu_\gamma(\mathbf{y}^\star_{k+1}|\mathbf{s})\propto \gamma$, when the last $k$ tokens in the state $\mathbf{s}$, match the first $k$ tokens in the gold subsequence $\mathbf{y}^\star$. Thus, the normalized return for $\mu_\gamma\to 1$, as $\gamma\to\infty$. We vary the heterogeneity of $\pi_b$ by finetuning GPT2-xl on data from a mixture of procedural policies with $\gamma \in [1000]$.

**Verifier-free SFT & verifier-based RL.** Given $n$ prompts, we collect trajectories from an expert by running rejection sampling over $\pi_b$, *i.e.*, for each prompt, we sample responses from $\pi_b$ until a correct trace is sampled. Next, we run SFT on this dataset in a verifier-free manner to obtain $\hat{\pi}^{\mathrm{vf}}_n$, similar to Zelikman et al. (2022). For RL, we implement a practical version of Algorithm 1. We train a reward model (GPT2-xl) as a multiclass classifier that predicts the bi-level reward over $H+1$ values: 0 to $H$. To collect training data, we draw a response $\tau \sim \pi_b(\cdot \mid \mathbf{x})$ for each of the $n$ prompts and annotate it ground-truth $r(\tau)$. Using this, we train a reward model $\hat{r}$, and learn policy $\hat{\pi}^{\mathrm{vb}}_n$ by running REINFORCE (with a KL constraint) against $\hat{r}$ (Ahmadian et al., 2024).

**Results: scaling test-time compute.** In Figure 4(a), we compare the test-time efficiency (normalized $J_r$) of SFT and RL as we scale test-time token budget $H$, fixing $n=2^{10}$. The performance of any procedural policy $\mu_\gamma$ improves with $H$, since there is a greater chance of sampling the gold subsequence. A similar argument applies to base and expert policies that are mixtures over $\mu_\gamma$. But perhaps counterintuitively, the gap between SFT and expert policy worsens as $H$ increases, matching our result in Theorem 5.4 where the gap grows with $H$. This is because the heterogeneity of each procedural policy (and hence $\sigma_b$) scales with $H$. On the flip side, RL nearly matches the expert (Theorem 5.7 shows suboptimality gap that is independent of $\sigma_b$), until a much higher $H$, after which it deviates slightly, likely because of decline in verifier accuracy at higher $H$ (Appendix D), resulting in reward hacking (Gao et al., 2023) during RL.

**Scaling data budget.** In Figure 4(b), we fix the test-time compute to $2^6$ tokens, and scale the data budget $n$. Expect-

edly, we see the performance of both SFT and RL improve, but the slope for the RL curve is much higher than that of SFT, which agrees with our theoretical result on VB being more sample efficient ($1/n$) than VF ($\sqrt{1/n}$ in Theorem 5.4).

**Effect of policy heterogeneity.** In Figure 4(c), we compare the performance of SFT and RL policies as we reduce the heterogeneity of the base policy. Consistent with our discussion in Section 5.1, the suboptimality gap for SFT reduces with the base policy's heterogeneity. In this regime we also find that VF methods outperform VB, primarily because of the decline in verifier accuracy (Appendix D), and perhaps the anti-concentration property is also not satisfied.

## 7. Results: Large-Scale Math Reasoning

Next, we extend our empirical results to math reasoning problems where we compare VF supervised finetuning on manually stitched search traces, and VB best-of-$N$ search (BoN) (Cobbe et al., 2021b). In BoN, we sample multiple responses from the base LLM, and choose the best one with an outcome verifier trained to predict 0/1 correctness labels. Here, the verifier is trained on $n$ samples generated from the base LLM for questions in the training data. Thus, BoN mimics the first few iterations of a VB online RL algorithm, initialized with the base LLM, and that maximizes rewards from a trained verifier. We mainly evaluate performance on the MATH (Hendrycks et al., 2021) reasoning benchmark, and use LLama-3.1/3.2 8B/3B instruct models (Dubey et al., 2024) supervised finetuned on MATH as the base LLMs. We vary the test-time compute budget from $2^9$ to $2^{13}$ tokens, and also vary the training data budget $n$ from $2^{12}$ to $2^{16}$.

**Verifier-free approach: SFT on stitched search traces.** Motivated by the approach of scaling test-time compute via iterative revisions (Qu et al., 2024), in this setting, we SFT total test-time budget $H$ on running as many rounds of revision as possible within the budget. To construct SFT data, we follow Snell et al. (2024) and construct an expert policy that is "close" to $\pi_b$ by first sampling a bunch of correct/incorrect *solution* traces from $\pi_b$, and then manually

stitching a uniformly random number of incorrect solutions followed by the correct into one *search* trace.

**Verifier-based approach: Best-of-N sampling against a verifier.** For each training problem, we collect a given number of traces $\sim \pi_b$, and label them with a 0/1 correctness score based on final answer match. We then train a verifier with binary cross-entropy loss. On a test problem, we use the verifier to rank $N$ solutions from $\pi_b(\cdot|\mathbf{x})$, at temperature 1.0 and choose the best one ($N$ scales linearly in budget $H$). While we run online RL in Section 6, due to computational constraints at higher $H$, we only compare with BoN here, which runs 1-step of policy improvement.

**VB BoN scales compute by $8\times$, data by $6\times$ of VF SFT.** At a fixed data budget of $2^{14}$ samples, BoN scales test-time compute by $8\times$ over SFT, and at a fixed test compute of $2^{12}$ tokens, VB scales data efficiency by $6\times$ (Figure 5(a)(b)). Revisiting Definition 4.2, we scale $n$ with $H$ and analyze the gap between BoN and SFT. We find that the accuracy gap grows super linearly in $\log H$, *i.e.*, the reward gap grows as $\Omega(\sqrt{H})$ (Figure 5(c)), matching Theorem 5.1.

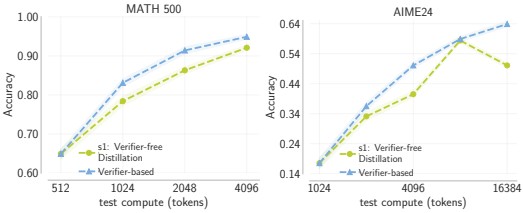

Figure 6: *Results with s1:* Fixing training data, and scaling test-time compute budget, we compare the performance of s1 (Muennighoff et al., 2025) trained with a verifier-free approach: supervised distillation, and our simple VB method: best-of-N search. In a compute matched evaluation, sampling $N$ short responses and selecting the best one with the trained verifier outperforms the budget forcing approach used in Muennighoff et al. (2025).

**s1 trained with verifier-free distillation performs worse than BoN.** In Figure 6, across different test compute budgets, we plot the performance of budget forcing in Muennighoff et al. (2025), that scales test compute over the s1 model. s1 itself was trained by running supervised distillation over traces from the Gemini Thinking (Google, 2024) model in a verifier-free manner. We compare this with BoN, where we sample $N$ responses of length $2^9$ (MATH500) or $2^{10}$ (AIME) and choose the best one with a trained outcome verifier. In a compute matched evaluation, we find that even when we fix the training data $n$, the verification based BoN approach improves over budget-constrained s1.

**VF generalizes on less heterogeneous problems, but memorizes heterogeneous ones.** We analyze the performance of running SFT/BoN on different problem buckets, where each bucket consists of problems of low, medium or high value of heterogeneity, at token budget $2^{10}$ (Figure 7). When $\sigma_{\mathbf{x}}$ is small, VF SFT clones the trace well and improves over

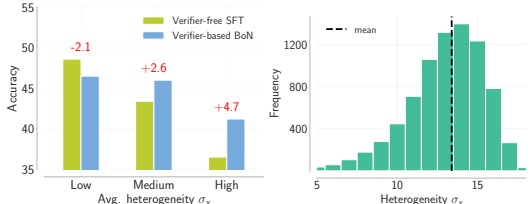

Figure 7: *Heterogeneity hurts SFT, but SFT outperforms BoN on homogeneous problems:* Across problems, we plot the distribution of $\sigma_{\mathbf{x}}$ (Definition 5.2), bucket problems by heterogeneity, and run SFT, BoN on each bucket. We find that verifier-free SFT can outperform BoN when the heterogeneity measured by $\sigma_{\mathbf{x}}$ is low, but the opposite is true when $\sigma_{\mathbf{x}}$ is high.

VB BoN, which can suffer from lack of coverage or inaccuracy of verifier (Appendix E). In contrast, when $\sigma_{\mathbf{x}}$ is larger, VB BoN dominates since VF SFT fails to generalize under heterogeneity and mainly memorizes responses. The distribution of $\sigma_{\mathbf{x}}$ is also skewed towards higher values, resulting in VB methods performing better on average (Figure 5).

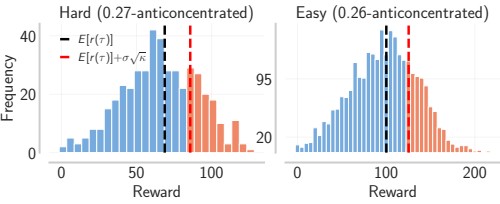

Figure 8: *Anti-concentration coefficient in practice:* For easy and hard problem sets in MATH, we compute the distribution of bi-level rewards on the correct traces sampled from base LLM. We find that for $\kappa = 0.5$ (controls the $\chi^2$ ball of the expert policy), there is a non-trivial ($\approx 1/4$) probability of observing rewards better than the mean reward by atleast $\sigma\sqrt{\kappa}$ ($\sigma$ computed by averaging over prompts in the easy/hard bucket), implying that base policy is roughly 0.25-anti-concentrated (Property 5.6).

**Base LLM is anti-concentrated in practice.** In Figure 8, we plot the distribution over bi-level rewards (Property 4.1) that measure test-compute scaling, conditioned on correct answers. With $\kappa = 0.5$, we mark in red the performance needed for trained LLM to improve over any expert in $\kappa$-$\chi^2$ ball around $\pi_b$. On both easy (acc.$>0.3$) and hard problems (acc.$<0.3$), the region beyond the red mark is $\approx 1/4$, implying that $\pi_b$ has an anti-concentration coefficient of $\approx$ acc. $\times 0.25$ (Property 5.6). Thus, the VB BoN is able to cover correct answers, which only improves with more test compute. Theorem 5.7 suggests that with $H/\eta$ samples BoN can outperform a policy that is $\eta$ close to the red mark.

> **Takeaways: Trends on MATH match our theory.**
> Base LLMs (*e.g.*, Llama-3.1-8B) exhibit heterogeneous and anticoncentrated reward distributions. VB methods outperform VF for a fixed test compute budget and the gap only grows as we increase training data and test budget. Although, when heterogeneity is indeed low, VF can outperform VB.

# Acknowledgements

All experiments in this work were run at Carnegie Mellon University. We thank Max Simchowitz, Andrea Zanette, Yuxiao Qu, Max Sobol Mark, Kwanyoung Park, Matthew Yang, Bhavya Agrawalla, Christina Baek, Charlie Snell, Yifei Zhou, Yi Su, Paria Rashidinejad, Ahmad Beirami and members of the CMU AIRe lab for feedback on an earlier version of this paper and informative discussions. AK is thankful for the support from Office of Naval Research under N00014-24-12206. AS is thankful for the generous support of JP Morgan AI PhD Fellowship. NR is thankful for support from NSF Grants IIS-1901252 and CCF-2211209. The authors thank the TRC program at Google Cloud and Lambda labs for providing compute resources that supported this work.

# Impact Statement

This paper presents work whose goal is to advance the field of Machine Learning. There are many potential societal consequences of our work, none which we feel must be specifically highlighted here.

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

# Appendices

## A. Discussion, Limitations, and Future Work

Recent results show that capabilities of foundation models improve as we sample more tokens from them at test time. But, this paradigm of scaling test-time compute is only sustainable if there exist learning algorithms that can learn policies, which make efficient use of test-time compute and keep improving as we scale test-time budgets. To study this, we first formalize the problem of optimizing test-time compute efficiency under our bi-level rewards (Property 4.1). Then, we define what it means to scale test-time compute efficiency asymptotically, mainly when comparing a pair of algorithms (Definition 4.2).

Based on these definitions, we present a novel theoretical study that analyzes two classes of popular algorithms. These algorithms train LLMs to use higher compute budgets at test-time, much higher than the length of correct answers for typical problems. Crucially, we separate these classes along the axis of access to reward or verification signals, and find that without access to verification (which can be in the form of 0/1 rewards during training, or trained verifiers at test-time), the performance of learning algorithms may not scale efficiently to large budgets compares to a simple verification-based approach. We prove this separation under two conditions on the base pre-trained LLM we start with. In particular, we show that when the base policy is heterogeneous (i.e., conditioned on a problem, the distribution of bi-level rewards has a high variance), no verifier-free learning algorithm can accurately learn any expert in a $\chi^2$ ball around the base policy. While every verifier-free algorithm suffers from a heterogeneous base policy, we show that when the base policy satisfies a weak anti-concentration condition: for all problems, the pretrained LLM puts a constant mass on a region of rewards, slightly higher than mean performance on the problem, then a simple verifier-based algorithm we analyze is already good enough to closely approximate the expert policy, which is supposed to scale well as we scale test-time compute further. We verify that the above conditions of base policy heterogeneity and anti-concentration are satisfied in practice, which neatly ties our theoretical abstractions and results to practical settings and empirical observations. We also compare our theoretical predictions on the gap between VF and VB methods on the MATH and AIME benchmarks (with the s1 model (Muennighoff et al., 2025) and a sequential self-correction model (Snell et al., 2024)), and a didactic setting which allows us to control the heterogeneity explicitly.

**Limitations and future work.** In this work, we mainly group algorithms based on whether or not they utilize access to verification signals for learning. Future work should consider building on our analysis to compare verifier-based algorithms that query sparse vs. dense forms of verification, *e.g.*, process-based rewards. Theoretically, it would also be interesting to extend our analysis of verifier-based algorithms with bi-level rewards to other classes of reward functions, including generative rewards (Zhang et al., 2024). Finally, since it is very expensive to train LLMs to use long contexts at test-time ($> 32k$) an analysis of scaling behaviors for RL with outcome, or dense rewards, and other verifier-based approaches would be crucial for making progress in this area. We believe that our analysis could provide certain "intuitions" about how to set up the right problems for such a scaling study.

## B. Additional Related Work

**Scaling test-time compute.** Recent works (Sardana et al., 2023; Snell et al., 2024) have shown that scaling test-time compute can improve performance at a rate faster than that afforded by traditional approaches of scaling data (Li et al., 2024) or model parameters (Hoffmann et al., 2022), implying that training compute can often be traded off optimally for test-compute (Villalobos & Atkinson, 2023; Jones, 2021). There are two popular ways of spending test compute. First, is to autoregressively sample from the LLM long "chains-of-thought" that resemble linearized search traces (Yao et al., 2023; Gandhi et al., 2024) or an iterative refinement of answers (Qu et al., 2024; Kumar et al., 2024). Second, is to explicitly implement search procedures (Wu et al., 2024; Beeching et al., 2024) with trained verifiers (Cobbe et al., 2021a; Setlur et al., 2024b). In our work, we empirically show that either of these approaches can scale well, and both theoretically and

empirically examine a different and critical axis of separating these approaches: access to verification during training or inference. Additionally, recent works (Chen et al., 2024c; Setlur et al., 2025) raise concerns about the unncessary wastage of test-time compute by sampling overly long responses for even simple questions (Yang et al., 2024). In our work, we use a "bi-level" reward formulation to capture what it means to efficiently use test-compute, and how to compare the asymptotic compute efficiency of verifier-free and verfier-based algorithms.

**Access to verification.** We say that a finetuning algorithm has access to verification if it directly uses ground truth rewards, *e.g.*, the 0/1 correctness labels on math solutions (Uesato et al., 2022; Bi et al., 2024); or if it queries trained verifiers for collecting training data (Hosseini et al., 2024) and running search procedures at test-time (Welleck et al., 2024; Chen et al., 2024b; Chow et al., 2024). The former approach of training LLMs to generate long "chains of thought" with final reward on-policy RL (Kimi-Team, 2025; DeepSeek-AI et al., 2025) has shown impressive gains on reasoning benchmarks. For off-policy RL algorithms (Rafailov et al., 2023; Zelikman et al., 2022; Singh et al., 2023) that utilize verification, converting the same 0/1 rewards into value function based process verification has been shown to be critical (Setlur et al., 2024a). Apart from these verification can also be generative (Zhang et al., 2024) and implicit (Yuan et al., 2024) where the same LLM is trained to generate and self-verify responses iteratively. In this work, we bucket all the above as verifier-based algorithms, and formally show that the asymptotic performance of this class scales test-compute more efficiently than approaches that do not query any sort of rewards, highlighting the critical role played by access to verification.

**Verifier-free algorithms.** Multiple works have proposed to scale test-time compute by finetuning pre-trained LLMs on manually stitched search traces (Gandhi et al., 2024; Nie et al., 2024) that all lead to the correct solution. The goal here is to force the LLM to mimic known search procedures like Monte-Carlo tree search (Yang et al., 2022; Xie et al., 2024) or $A^\star$ (Lehnert et al., 2024) on training questions, with the hope that the LLM learns to search for solutions on test problems too (Sel et al., 2023). Crucially these algorithms do not annotate search trajectories in the training data with any reward, and the LLM is forced to mimic multipe search traces that are "heterogeneous" in nature, *i.e.*, different traces spending varying number of tokens (for search) to arrive at the same final solution. In our work, we analyze how this heterogeneous nature makes it hard for *any* supervised finetuning algorithm to generalize, resulting in a poor test-time scaling law for these, matching observations in practice (Kumar et al., 2024; Xiang et al., 2025).

**Formalizing self-improvement in LLMs.** Some recent works analyze self-improvement in LLMs under the assumption of a verification-generation gap (Song et al., 2024), or frame sharpening model confidence (Huang et al., 2024a) as a form of self-improvement, where the most likely solution is assumed to be correct. In contrast, we do not assume that learning to verify is easier than learning to generate the correct trace, but instead show much weaker yet realistic conditions on the pre-trained LLM, where such a gap exists.

# C. Proofs from Section 5

## C.1. Useful Lemmas

For a pair of probability measures $P$ and $Q$, we define the total variation distance as $D_{\mathrm{TV}}(P, Q) = \frac{1}{2} \int |\mathrm{d}P - \mathrm{d}Q|$, and define the $\chi^2$-divergence by $D_{\chi^2}(P\|Q) = \int \frac{(\mathrm{d}Q - \mathrm{d}Q)^2}{\mathrm{d}Q}$ if $P \ll Q$ and $\chi^2(P\|Q) = +\infty$ otherwise. We define the KL divergence by $D_{\mathrm{KL}}(P\|Q) = \int \mathrm{d}P \log\left(\frac{\mathrm{d}P}{\mathrm{d}Q}\right)$ if $P \ll Q$ and $D_{\mathrm{KL}}(P\|Q) = +\infty$ otherwise.

**Lemma C.1** ((Polyanskiy & Wu, 2014))**.** *The following inequalities hold:*

- $D_{\mathrm{TV}}(P, Q) \leq D_{\mathrm{H}^2}(P, Q) \leq 2D_{\mathrm{TV}}(P, Q)$.

- $\frac{1}{6} D_{\mathrm{H}^2}(P, Q) \leq D_{\chi^2}\left(P \| \frac{1}{2}(P + Q)\right) \leq D_{\mathrm{H}^2}(P, Q)$.

- $D_{\mathrm{TV}}(P, Q) \leq \sqrt{\frac{1}{2} D_{\mathrm{KL}}(P\|Q)}$

**Lemma C.2** (Change of measure (Polyanskiy & Wu, 2014; Foster et al., 2024a))**.** *Let $P$ and $Q$ be probability distributions over a measurable space $(\mathcal{Y}, \mathscr{F})$. Then for all functions $h : \mathcal{Y} \to \mathbb{R}$,*

$$|\mathbb{E}_P[h(Y)] - \mathbb{E}_Q[h(Y)]| \leq \sqrt{\mathrm{Var}_Q[h(Y)] \cdot D_{\chi^2}(P\|Q)} \qquad (\chi^2\text{-CoM})$$

$$\leq \sqrt{\frac{1}{2}\left(\mathbb{E}_P[h^2(Y)] + \mathbb{E}_Q[h^2(Y)]\right) \cdot D_{\mathrm{H}}^2(P, Q)} \qquad (\mathbb{H}\text{-CoM})$$

**Lemma C.3** (Total expert heterogeneity). *For any policy $\pi$, recall the definition of heterogeneity in Definition 5.2. For this definition of heterogeneity the following equivalance to the expected conditional variance of rewards is true:*

$$\sigma_\pi^2 = \mathbb{E}_{\mathbf{x} \sim \rho} \mathrm{Var}_{\tau \sim \pi(\cdot|\mathbf{x})} [r(\tau)].$$

*Proof.* Let us begin by recalling the definition of $\sigma_\pi^2$.

$$\sigma_\pi^2 =: \sum_{h=1}^H \mathbb{E}_{\mathbf{s}_h \sim d_h^\pi} \left[ \mathrm{Var}_{\pi(\cdot|\mathbf{s}_h)} [Q_\pi(\mathbf{s}_h, a_h)] \right].$$

Now let us expand $\mathrm{Var}_{\pi(\cdot|\mathbf{s}_h)} [Q_\pi(\mathbf{s}_h, a_h)]$ in the following way.

$$\mathrm{Var}_\pi \left[ \sum_{h'=h}^H r(\mathbf{s}_{h'}, a_{h'}) \middle| \mathbf{s}_h \right]$$

$$= \mathrm{Var}_\pi \left[ r(\mathbf{s}_h, a_h) + \sum_{h'=h+1}^H r(\mathbf{s}_{h'}, a_{h'}) \middle| \mathbf{s}_h \right]$$

$$= \mathbb{E}_\pi \left[ \left( r(\mathbf{s}_h, a_h) - V_\pi(\mathbf{s}_h) + \sum_{h'=h+1}^H r(\mathbf{s}_{h'}, a_{h'}) \right)^2 \middle| \mathbf{s}_h \right]$$

$$= \mathbb{E}_\pi \left[ \left( r(\mathbf{s}_h, a_h) + V_\pi(\mathbf{s}_{h+1}) - V_\pi(\mathbf{s}_h) + \sum_{h'=h+1}^H r(\mathbf{s}_{h'}, a_{h'}) - V_\pi(\mathbf{s}_{h+1}) \right)^2 \middle| \mathbf{s}_h \right]$$

$$= \mathbb{E}_\pi \left[ \left( Q_\pi(\mathbf{s}_h, a_h) - V_\pi(\mathbf{s}_h) + \sum_{h'=h+1}^H r(\mathbf{s}_{h'}, a_{h'}) - V_\pi(\mathbf{s}_{h+1}) \right)^2 \middle| \mathbf{s}_h \right]$$

Breaking the above expectation into three terms by expanding the square, note that the third term is zero because, $\mathbb{E}_\pi [Q_\pi(\mathbf{s}_h, a_h) - V_\pi(\mathbf{s}_{h+1}) \mid \mathbf{s}_h] = 0$, for any state $\mathbf{s}_h$ and in our autoregressive MDP with deterministic dynamics,

$$Q_\pi(\mathbf{s}_h, a_h) = r(\mathbf{s}_h, a_h) + V_\pi(\mathbf{s}_{h+1}),$$

also for every state $\mathbf{s}_h$. Recall that, here the state $\mathbf{s}_{h+1} = (\mathbf{s}_h, a_h)$. Additionally, we also take the expecation over the state distribution of $\mathbf{s}_h \sim d_h^\pi$, and since the equality is true individually for each value of $\mathbf{s}_h$, it also holds under the expectation over $\mathbf{s}_h$. This gives us the following.

$$\mathbb{E}_{\mathbf{s}_h \sim d_h^\pi} \left[ \mathbb{E}_\pi \left[ \left( r(\mathbf{s}_h, a_h) + V_\pi(\mathbf{s}_{h+1}) - V_\pi(\mathbf{s}_h) + \sum_{h'=h+1}^H r(\mathbf{s}_{h'}, a_{h'}) - V_\pi(\mathbf{s}_{h+1}) \right)^2 \middle| \mathbf{s}_h \right] \right]$$

$$= \mathbb{E}_{\mathbf{s}_h \sim d_h^\pi} \left[ \mathbb{E}_\pi \left[ (r(\mathbf{s}_h, a_h) + V_\pi(\mathbf{s}_{h+1}) - V_\pi(\mathbf{s}_h))^2 \middle| \mathbf{s}_h \right] + \mathbb{E}_\pi \left[ \left( \sum_{h'=h+1}^H r(\mathbf{s}_{h'}, a_{h'}) - V_\pi(\mathbf{s}_{h+1}) \right)^2 \middle| \mathbf{s}_h \right] \right]$$

$$+ 2 \cdot \mathbb{E}_{\mathbf{s}_h \sim d_h^\pi} \left[ \mathbb{E}_\pi [r(\mathbf{s}_h, a_h) + V_\pi(\mathbf{s}_{h+1}) - V_\pi(\mathbf{s}_h) | \mathbf{s}_h] \cdot \mathbb{E}_\pi \left[ \sum_{h'=h+1}^H r(\mathbf{s}_{h'}, a_{h'}) - V_\pi(\mathbf{s}_{h+1}) \middle| \mathbf{s}_h \right] \right]$$

As we noted above, the third term in the summation above is zero. Thus,

$$\mathbb{E}_{\mathbf{s}_h \sim d_h^\pi} \left[ \mathrm{Var}_\pi \left[ \sum_{h'=h}^H r(\mathbf{s}_{h'}, a_{h'}) \middle| \mathbf{s}_h \right] \right] = \mathbb{E}_{\mathbf{s}_{h+1} \sim d_{h+1}^\pi} \left[ \mathrm{Var}_\pi \left[ \sum_{h'=h+1}^H r(\mathbf{s}_{h'}, a_{h'}) \right] \middle| \mathbf{s}_h \right]$$

$$+ \mathbb{E}_{\mathbf{s}_h \sim d_h^\pi} [\mathrm{Var}_\pi [Q_\pi(\mathbf{s}_h, a_h)] | \mathbf{s}_h]$$

The above induction is true for all values of $h$. Now, taking the sum over $h$, from $h = 1$ to $h = H$ on both left and right sides of the equation and using the definition of $\sigma_\pi^2$, we get:

$$\sigma_\pi^2 = \mathbb{E}_{\mathbf{s}_1 \sim d_1^\pi} \left[ \text{Var}_\pi \left[ \sum_{h=1}^{H} r(\mathbf{s}_h, a_h) \middle| \mathbf{s}_1 \right] \right].$$

Recall from Section 3 that the first state $\mathbf{s}_1$ is simply the input prompt $\mathbf{x}$. Thus $d_1^\pi$ is indpendent of $\pi$ and is simply the distribution over the input prompts $\mathbf{x}$, which is defined as $\rho$. Plugging this into the above equation we get:

$$\sigma^2 = \mathbb{E}_{\mathbf{x} \sim \rho} \left[ \text{Var}_\pi \left[ \sum_{h=1}^{H} r(\mathbf{s}_h, a_h) \middle| \mathbf{x} \right] \right] = \mathbb{E}_{\mathbf{x} \sim \rho} \left[ \text{Var}_{\tau \sim \pi(\cdot | \mathbf{x})} \left[ r(\tau) \right] \right].$$

$\square$

**Lemma C.4.** *Consider a random variable $A$ which is almost surely non-negative and has mean $\mu$ and variance $\sigma^2$. For any $\theta \geq 0$,*

$$\mathbb{E} \left[ \frac{\theta(\mu - A)}{\sigma + \theta A} \right] \leq 2\theta^2 \tag{2}$$

*Proof.* Let $f(\theta) = \mathbb{E} \left[ \frac{\theta(\mu - A)}{\sigma + \theta A} \right]$. Observe that,

$$f'(\theta) = \mathbb{E} \left[ \frac{\mu - A}{\sigma + \theta A} \right] - \mathbb{E} \left[ \frac{\theta(\mu - A)A}{(\sigma + \theta A)^2} \right]$$

$$f''(\theta) = -2\mathbb{E} \left[ \frac{(\mu - A)A}{(\sigma + \theta A)^2} \right] + 2\mathbb{E} \left[ \frac{\theta(\mu - A)A^2}{(\sigma + \theta A)^3} \right]$$

$$= 2\mathbb{E} \left[ \frac{\theta(\mu - A)A^2 - (\mu - A)A(\sigma + \theta A)}{(\sigma + \theta A)^3} \right]$$

$$= 2\sigma \mathbb{E} \left[ \frac{(A - \mu)A}{(\sigma + \theta A)^3} \right]$$

$$= 2\sigma \mathbb{E} \left[ \frac{\mu(A - \mu)}{(\sigma + \theta A)^3} \right] + 2\sigma \mathbb{E} \left[ \frac{(A - \mu)^2}{(\sigma + \theta A)^3} \right]$$

$$\leq 2\sigma \mathbb{E} \left[ \frac{\mu(A - \mu)}{(\sigma + \theta A)^3} \right] + 2\sigma \mathbb{E} \left[ \frac{(A - \mu)^2}{\sigma^3} \right]$$

$$= 2\sigma \mathbb{E} \left[ \frac{\mu(A - \mu)}{(\sigma + \theta A)^3} \right] + 2$$

Note that $\mu(A - \mu)$ and $(\sigma + \theta A)^3$ are both increasing functions in $A$, and therefore,

$$\mathbb{E} \left[ \frac{\mu(A - \mu)}{(\sigma + \theta A)^3} \right] \leq 2\sigma \mathbb{E} \left[ \mu(A - \mu) \right] \mathbb{E} \left[ \frac{1}{(\sigma + \theta A)^3} \right] = 0.$$

This results in the upper bound $\|f''\|_\infty \leq 2$. Since $f(0) = 0$ and $f'(0) = 0$, we have that,

$$f(\theta) = \int_0^\theta f''(\alpha) \mathrm{d}\alpha \leq 2\theta^2.$$

$\square$

### C.2. Lower bound on $\sigma_e$: Proof of Lemma 5.3

In this section, we show that for any base policy $\pi_b$, and any expert policy $\pi_e$ such that $D_{\text{KL}} \left( \pi_e \| \pi_b \right) \leq \kappa$,

$$\sigma_e^2 \geq \sigma_b^2 - H\sigma_b \sqrt{\kappa/2}.$$

Since $D_{\mathrm{KL}}\left(\cdot\|\cdot\right) \leq \chi^2(\cdot\|\cdot)$ pointwise, this implies the lower bound on $\sigma_e$ within the $\chi^2$ ball.

By definition, observe that,

$$\sigma_\pi^2 = \mathbb{E}_{\mathbf{x}\sim\rho}[\mathrm{Var}_{\tau\sim\pi(\cdot|\mathbf{x})}\left[r(\tau)\right]]$$
$$= \frac{1}{2}\mathbb{E}_{\mathbf{x}\sim\rho}\left[\mathbb{E}_{\tau,\tau'\sim\pi(\cdot|\mathbf{x})}[(r(\tau)-r(\tau'))^2]\right]$$

Note that the squared Hellinger divergence $D_{\mathrm{H}}^2$ satisfies $D_{\mathrm{H}}^2(\cdot,\cdot) \leq D_{\mathrm{KL}}(\cdot,\cdot)$ pointwise (cf. Lemma 2.4 in Tsybakov (2009)). With the choice $Y = (\tau,\tau')$ in the change-of-measure argument in Equation ($\mathbb{H}$-CoM) of Lemma C.2, $h(Y) = (r(\tau) - r(\tau'))^2$ and $P$ denote the distribution over trajectories $\pi_b(\cdot \mid \mathbf{x})$ and $Q$ denote the distribution over trajectories induced by $\pi_e(\cdot \mid \mathbf{x})$,

$$|\mathrm{Var}_{\tau\sim\pi_b(\cdot|\mathbf{x})}[r(\tau)] - \mathrm{Var}_{\tau\sim\pi_e(\cdot|\mathbf{x})}[r(\tau)]| \leq \frac{1}{2}\sqrt{\frac{1}{2}\left(\mathbb{E}_P\left[h^2(Y)\right] + \mathbb{E}_Q\left[h^2(Y)\right]\right)\cdot D_{\mathrm{KL}}\left((\tau_e,\tau_e')\|(\tau_b,\tau_b')\right)}$$

$$\leq \frac{1}{2}\sqrt{\left(\mathbb{E}_P\left[h^2(Y)\right] + \mathbb{E}_Q\left[h^2(Y)\right]\right)\cdot D_{\mathrm{KL}}\left(\tau_e\|\tau_b\right)} \tag{3}$$

where in the last inequality, we use the fact that $\tau_e$ and $\tau_e'$ are i.i.d. $\sim \pi_e(\cdot \mid \mathbf{x})$, and likewise $\tau_b$ and $\tau_b'$ are i.i.d. $\sim \pi_b(\cdot \mid \mathbf{x})$, and the chain rule of KL divergence. What remains is to bound $\mathbb{E}_{\tau\sim\pi(\cdot|\mathbf{x})}[(r(\tau) - r(\tau'))^4]$ for $\pi = \pi_e$ and $\pi = \pi_b$. Since $|r(\tau) - r(\tau')| \leq H$ almost surely,

$$\mathbb{E}_{\tau\sim\pi(\cdot|\mathbf{x})}[(r(\tau) - r(\tau'))^4] \leq 2H^2\mathrm{Var}_{\tau\sim\pi(\cdot|\mathbf{x})}[r(\tau)]$$

Let's denote $A = \mathrm{Var}_{\tau\sim\pi_e(\cdot|\mathbf{x})}[r(\tau)]$ and $B = \mathrm{Var}_{\tau\sim\pi_b(\cdot|\mathbf{x})}[r(\tau)]$. Combining with Equation (3) and squaring, and denoting $D_{\mathrm{KL}}\left(\tau_e\|\tau_b\right) = \kappa_{\mathbf{x}}$,

$$(A-B)^2 \leq \frac{H^2}{4}(A+B)\cdot\kappa_{\mathbf{x}}$$
$$\implies A^2 - \left(2B + \frac{\kappa_{\mathbf{x}}H^2}{4}\right)A + \left(B^2 - \frac{\kappa_{\mathbf{x}}H^2}{4}B\right) \leq 0 \tag{4}$$

This is a quadratic equation in $A$. Solving, we get,

$$A \geq \left(B + \frac{\kappa_{\mathbf{x}}H^2}{8}\right) - \sqrt{\left(B + \frac{\kappa_{\mathbf{x}}H^2}{8}\right)^2 - \left(B^2 - \frac{\kappa_{\mathbf{x}}H^2}{4}B\right)}$$
$$= \left(B + \frac{\kappa_{\mathbf{x}}H^2}{8}\right) - \sqrt{\frac{\kappa_{\mathbf{x}}H^2}{2}B + \frac{\kappa_{\mathbf{x}}^2 H^4}{64}}$$
$$\geq B - H\sqrt{\kappa_{\mathbf{x}}B/2}$$

where the last inequality uses the subadditivity of the $\sqrt{\cdot}$ function. This implies that,

$$\mathrm{Var}_{\tau\sim\pi_e(\cdot|\mathbf{x})}[r(\tau)] \geq \mathrm{Var}_{\tau\sim\pi_b(\cdot|\mathbf{x})}[r(\tau)] - H\sqrt{(\kappa_{\mathbf{x}}/2)\mathrm{Var}_{\tau\sim\pi_b(\cdot|\mathbf{x})}[r(\tau)]}$$

Taking an expectation over $\mathbf{x}\sim\rho$ on both sides, and using Jensen's inequality,

$$\sigma_e^2 \geq \sigma_b^2 - H\mathbb{E}_{\mathbf{x}\sim\rho}\left[\sqrt{(\kappa_{\mathbf{x}}/2)\mathrm{Var}_{\tau\sim\pi_b(\cdot|\mathbf{x})}[r(\tau)]}\right]$$
$$\geq \sigma_b^2 - H\sqrt{\mathbb{E}_{\mathbf{x}\sim\rho}[\kappa_{\mathbf{x}}/2]\mathbb{E}_{\mathbf{x}\sim\rho}\left[\mathrm{Var}_{\tau\sim\pi_b(\cdot|\mathbf{x})}[r(\tau)]\right]}$$
$$= \sigma_b^2 - H\sigma_b\sqrt{\kappa/2}$$

Noting that $\mathbb{E}_{\mathbf{x}\sim\rho}[\kappa_{\mathbf{x}}] \leq \kappa$. Solving for the larger root of the quadratic in Equation (4), we also arrive at the upper bound,

$$A \leq B + H\sqrt{\kappa_{\mathbf{x}}B/2} + \frac{\kappa_{\mathbf{x}}H^2}{4}$$
$$\implies \sigma_e^2 \leq \sigma_b^2 + H\sigma_b\sqrt{\kappa/2} + \frac{\kappa H^2}{4}. \tag{5}$$

which follows by taking an expectation over $\mathbf{x}\sim\rho$.

**Optimality of Lemma 5.3.** The above result is tight up to constants. Consider an autoregressive MDP with a single prompt, where picking action $a_0$ at time 1 results in hitting a bi-level (so, regardless of future actions, a reward of 1 is collected at each step) and picking action $a_1$ results in a reward of 0 forever. $\pi_b$ picks the first branch with probability $p$ and the second with probability $1-p$ at $t=1$. Then, $\sigma_b^2 = p(1-p)H^2$ and by scaling $p$ from 0 to 1/2, any $0 \le \sigma_b^2 \le H^2/4$ can be achieved. On the other hand, consider the policy $\pi_e$ which plays $a_0$ with probability $p - \theta$ at $t = 1$. Suppose $p$ is a constant. Then,

$$
\begin{aligned}
\chi^2(\pi_e \| \pi_b) &= \frac{(p-\theta)^2}{p} + \frac{1 - 2(p-\theta) + (p-\theta)^2}{1-p} - 1 \\
&= \frac{p^2 - 2p\theta + \theta^2}{p} + \frac{(1-p)^2 + 2\theta(1-p) + \theta^2}{1-p} - 1 \\
&= \frac{\theta^2}{p} + \frac{\theta^2}{1-p} \\
&= \frac{\theta^2}{p(1-p)}
\end{aligned}
$$

Therefore, choosing $\theta = \min\{p, \sqrt{\kappa p(1-p)}\}$, we get,

$$
\chi^2(\pi_e \| \pi_b) \le \kappa
$$

And furthermore that, $\sigma_e^2 = (p-\theta)(1 - (p-\theta))H^2$ and therefore,

$$
\begin{aligned}
\sigma_e^2 - \sigma_b^2 &= (p-\theta)(1 - (p-\theta))H^2 - p(1-p)H^2 \\
&= -(\theta + \theta^2 - 2p\theta)H^2,
\end{aligned}
$$

when $\theta = p$, we get $\sigma_e^2 = 0$. When $\theta = \sqrt{\kappa p(1-p)}$, this is assumed to be in the regime $\theta > p$ and so,

$$
\begin{aligned}
\sigma_e^2 - \sigma_b^2 &\le -(\theta + p\theta - 2p\theta)H^2 \\
&\le -\frac{\theta}{2}H^2
\end{aligned}
$$

where in the last equation we recall the assumption that $p \le 1/2$. Plugging in $\theta$ and observing that $H^2\theta = H\sigma_b\sqrt{\kappa}$ completes the proof.

### C.3. Proof of Theorem 5.4

We will state a slightly more formal version of Theorem 5.4 below in Appendix C.3.3. Prior to this, we introduce some relevant notation necessary to state the main result.

#### C.3.1. MEASURE OF COMPLEXITY: $L_k^\star$

Consider an arbitrary partitioning of the prompt space $\mathcal{X}$ into $k$ disjoint parts, denoted $\{\mathcal{X}_i\}_{i=1}^k$. Let $\{\mathcal{X}_i^\star\}_{i=1}^k$ denote the partitioning of the prompt space which maximizes,

$$
L(\{\mathcal{X}_i\}_{i=1}^k) =: \min\left\{ \mathbb{E}_{\mathbf{x} \sim \rho}[\sigma_{e,\mathbf{x}} \mathbb{I}(\mathbf{x} \in \cup_{i \in K} \mathcal{X}_i)] : K \subseteq [k] \text{ and } |K| \ge k/4 \right\}. \tag{6}
$$

And let $L_k^\star = L(\{\mathcal{X}_i^\star\}_{i=1}^k)$. Our construction, and lower bounds derived therafter are stated in terms of $\{\mathcal{X}_i^\star\}_{i=1}^k$ and $L_k^\star$. We devote the first part of this section toward interpretations of $L_k^\star$.

Recall that $\tilde{\sigma}_e = \text{Median}(\{\sigma_{\pi,\mathbf{x}} : \mathbf{x} \in \mathcal{X}\})$, $\bar{\sigma}_e = \mathbb{E}_{\mathbf{x} \sim \rho}[\sigma_{\pi_e,\mathbf{x}}]$ and $\sigma_e^2 = \mathbb{E}_{\mathbf{x} \sim \rho}[\sigma_{\pi_e,\mathbf{x}}^2]$. We will first show that $L_k^\star \gtrsim \tilde{\sigma}_e$ always. Later, we will show that if $\sigma_e^2 \le c\bar{\sigma}_e^2$ for a sufficiently small constant $c > 1$, $L_k^\star \gtrsim \sigma_e$.

#### C.3.2. INTERPRETATIONS OF, AND BOUNDS ON $L_k^\star$

**Lemma C.5.** *Consider any $8 \le k \le |\mathcal{X}|/4$. Then, $L_k^\star \ge \frac{1}{32}\tilde{\sigma}_e$.*

*Proof.* We will prove this statement in two assertions,

1. When $k$ is a power of two, $L^\star_{k/2} \geq L^\star_k$.

2. When $k$ is any power of two, and any $k/2 \leq k' \leq k$, $L^\star_{k'} \geq \frac{1}{2}L^\star_k$.

For a subset $X \subseteq \mathcal{X}$, define its score $s(X) = \mathbb{E}_{\mathbf{x} \sim \rho}[\sigma_{e,\mathbf{x}}\mathbb{I}(\mathbf{x} \in X)]$. Assuming these two assertions, we will prove the main lemma first, and then come back to proving them.

**Proof of the main lemma.** Consider $k$ as the largest power of 2 between $|\mathcal{X}|/4$ and $|\mathcal{X}|/2$. For this choice, consider the partition of $\mathcal{X}$ into $k$ sets by choosing the first $k$ parts as singleton sets, consisting of the top $k$ prompts $\mathbf{x} \in \mathcal{X}$ with the highest values of $\sigma_{e,\mathbf{x}}$; the remaining prompts are distributed among sets in the partition arbitrarily. Notably, the score of each part $\mathcal{X}_i$ in this partition satisfies $s(\mathcal{X}_i) \geq \text{Median}(\{s(\{\mathbf{x}\}) : \mathbf{x} \in \mathcal{X}\})$; by implication, for any such value of $k$,

$$L^\star_k \geq \frac{k}{4}\text{Median}(\{s(\mathbf{x}) : \mathbf{x} \in \mathcal{X}\}) \geq \frac{|\mathcal{X}|}{16}\text{Median}(\{s(\{\mathbf{x}\}) : \mathbf{x} \in \mathcal{X}\}) = \frac{1}{16}\text{Median}(\{\sigma_{e,\mathbf{x}} : \mathbf{x} \in \mathcal{X}\}) \quad (7)$$

where the last equation uses the fact that $\rho$ is the uniform distribution over $\mathcal{X}$. Therefore, for any $k' \leq k$, we have that $L^\star_{k'} \geq \frac{1}{2}L^\star_k = \geq \frac{1}{32}\text{Median}(\{\sigma_{e,\mathbf{x}} : \mathbf{x} \in \mathcal{X}\})$.

**Proof of the first assertion.** Consider the optimal partition which induces $L^\star_k$, $\{\mathcal{X}^\star_i\}^k_{i=1}$, arranged in increasing order of scores. Note then, that $L^\star_k = \sum^{k/4}_{i=1} s(\mathcal{X}_i)$ Consider the partition of $\mathcal{X}$ into $k/2$ parts, as $\{\mathcal{X}^\star_1 \cup \mathcal{X}^\star_2, \mathcal{X}^\star_3 \cup \mathcal{X}^\star_4, \cdots, \mathcal{X}^\star_{k-1} \cup \mathcal{X}^\star_k\}$. Since scores are additive, the $k/8$ parts with the lowest scores must be $\{\mathcal{X}^\star_i \cup \mathcal{X}^\star_{i+1}\}^{k/8}_{i=1}$. This implies the first assertion.

**Proof of the second assertion.** Consider the optimal partition which induces $L^\star_k$, $\{\mathcal{X}^\star_i\}^k_{i=1}$. By dissolving the bottom $k - k'$ parts (in terms of score) of $\{\mathcal{X}^\star_i\}^k_{i=1}$ and merging them with other parts, this results in a partitioning of $\mathcal{X}$ such that the sum of $k'/4$ worst scores of the parts must be at least $(k'/k)L^\star_k \geq L^\star_k/2$.

$\square$

**Lemma C.6.** *Suppose $\sigma^2_e \leq \frac{4}{3}\overline{\sigma}^2_e$, then $\widetilde{\sigma}_e \geq \frac{1}{10}\overline{\sigma}_e \geq \frac{1}{15}\sigma_e$.*

*Proof.* By the Paley-Zygmund inequality,

$$\Pr_{\mathbf{x} \sim \rho}\left[\sigma_{e,\mathbf{x}} \geq \frac{1}{10}\overline{\sigma}_e\right] \geq \frac{4}{5} \times \frac{\overline{\sigma}^2_e}{\sigma^2_e} \quad (8)$$

When $\sigma^2_e \leq \frac{4}{3}\overline{\sigma}^2_e$, the LHS is at least $3/5$. This means that at least $3|\mathcal{X}|/5$ of the prompts satisfy $\sigma_{e,\mathbf{x}} \geq \frac{1}{10}\overline{\sigma}_e$, and so $\text{Median}(\{\sigma_{e,\mathbf{x}} : \mathbf{x} \in \mathcal{X}\}) \geq \frac{1}{10}\overline{\sigma}_e$. $\square$

As a corollary of this lemma, we have that,

**Corollary C.7.** *Under the condition $\sigma^2_e \leq (4/3)\overline{\sigma}^2_e$, for every $k \leq |\mathcal{X}|/4$, we have that $L^\star_k \geq c\sigma_e$ for some absolute constant $c > 0$.*

Having introduced these interpretations of $L^\star_k$, we prove the following instance-dependent lower bound on the suboptimality of any verifier-free algorithm.

### C.3.3. LOWER BOUNDS ON VERIFIER-FREE APPROACHES

Below we introduce the class of rewards for which we prove the instance-dependent lower bound in Theorem 5.4.

**Definition C.8** (Half-bi-level rewards). Define the class of half-bi-level rewards, $\mathcal{R}_{1/2}$, as those reward functions such that every trajectory contains a bi-level at or before time $t = \lfloor H/2 \rfloor$. Namely, for any trajectory $(s_1, a_1, \cdots, s_H, a_H)$, $r(s_t, a_t) = 1$ for every $t \geq \lfloor H/2 \rfloor$ for any reward $r \in \mathcal{R}_{1/2}$.

*Remark* C.9. Although half-bi-level rewards are constrained to have all their bi-levels before time $H/2$, this does not preclude there from existing policies having high variance under rewards from this class. In particular, there exists a policy $\pi$ and a reward $r \in \mathcal{R}_{1/2}$ such that $\sigma^2_\pi = H^2/16$.

**Theorem C.10.** *Suppose $|\mathcal{X}| \geq 16$ and choose any $4 \leq k \leq |\mathcal{X}|/4$. Consider any autoregressive MDP and assume that $\rho = \text{Unif}(\mathcal{X})$. For any choice of reward $r \in \mathcal{R}_{1/2}$, base policy $\pi_b$ and expert policy $\pi_e \in \Pi_\varepsilon$, there exists an alternate family of expert policies $\Pi'$ of size $\lceil 2^{k/4} \rceil$ and reward class $\mathcal{R}' \subset \mathcal{R}$ (also of the same size), such that,*

1. *$\pi_e \in \Pi'$ and $r \in \mathcal{R}'$,*

2. *$\Pi' \subseteq \Pi_{\varepsilon'}$ corresponds to a family of feasible expert policies with $\varepsilon' = 3(1 + \varepsilon) \cdot \max\left\{\frac{H\sqrt{\varepsilon_{stat}}}{\sigma_{\min}}, \frac{H^2 \varepsilon_{stat}}{\sigma_{\min}^2}\right\}$. Here, $\sigma_{\min} = \min_{\mathbf{x} \in \mathcal{X}} \sigma_{e,\mathbf{x}}$.*

3. *For every $r' \in \mathcal{R}'$ and policy $\pi' \in \Pi'$, $\sigma_{r'}^2(\pi') \leq \sigma_e^2 + H\sigma_e\sqrt{\varepsilon_{stat}} + H^2 \varepsilon_{stat}$.*

4. *For any realizable verifier-based learning algorithm, satisfying $\hat{\pi}_n^{vf} \in \Pi'$,*

$$\max_{\pi' \in \Pi'} \max_{r' \in \mathcal{R}} \Pr\left(J_{r'}(\pi') - J_{r'}(\hat{\pi}_n^{vf}) \geq L_k^\star \sqrt{\varepsilon_{stat}}\right) \geq 1/8 \tag{9}$$

*Here, we define $\varepsilon_{stat} = \frac{\log(|\Pi'|)}{16n}$ and assume that $n$ is sufficiently large so that $\varepsilon_{stat} \leq \min_{\mathbf{x} \in \mathcal{X}} \sigma_{e,\mathbf{x}}^2 / (J_r(\pi_e|\mathbf{x}))^2$.*

*Proof structure.* We define the alternate policy class $\Pi'$ across Lemma C.14 and Lemma C.15, culminating in Appendix C.3.4. Property 2 (i.e., $\Pi' \subseteq \Pi_{\varepsilon'}$) and Property 3 (i.e., the bound on the variance of policies in $\Pi'$ on rewards in $\mathcal{R}'$) are established in Lemma C.14. $\square$

*Remark* C.11. The results of (Foster et al., 2024a) establish a similar lower bound for autoregressive MDPs. However their construction specifically assumes, either $(i)$ there is a single prompt, or $(ii)$ the adversary constructing an alternate hard instance can change the initial state distribution $\rho$. This follows from the fact that their alternate policy is constructed in a way which does not preserve the initial state distribution of the MDP (cf. Lemma G.1 in their paper).

Our lower bound scales with $L_k^\star \gtrsim \widetilde{\sigma}_e$ where $\widetilde{\sigma}_e = \text{Median}(\{\sigma_{e,\mathbf{x}} : \mathbf{x} \in \mathcal{X}\})$, rather than $\sigma_e$, as previous work (Foster et al., 2024a) hints in the case of a single prompt. In general, it turns out that it is not possible to have an instance-dependent lower bound that scales as $\Omega(\sigma_e\sqrt{\log(|\Pi|)/n})$. There exist a class of MDPs where verifier free approaches achieve an error of $\mathcal{O}(\widetilde{\sigma}_e\sqrt{\log(|\Pi|)/n})$, even under the worst case choice of policy class, and improve over the suggested $\Theta(\sigma_e\sqrt{\log(|\Pi|)/n})$ instance-dependent error.

**Theorem C.12.** *Consider an autoregressive MDP with $|\mathcal{A}| = 2$ and $H = 1$. There exists an expert policy $\pi_e$, such that **for any policy class** $\Pi \ni \pi_e$ of size $|\Pi| \geq 2^{\Omega(|\mathcal{X}|)}$, there exists a verifier-free learner such that with probability at least $1 - \delta$,*

$$\max_{r \in \mathcal{R}} J_r(\pi_e) - J_r(\hat{\pi}_n^{vf}) \leq \widetilde{\mathcal{O}}_{|\mathcal{X}|,\delta}\left(\widetilde{\sigma}_e\sqrt{\frac{\log(|\Pi|)}{n}} + \frac{\log(|\Pi|)}{n}\right)$$

$$= \widetilde{\Theta}_{|\mathcal{X}|,\delta}\left(\frac{\sigma_e}{\sqrt{|\mathcal{X}|}} \cdot \sqrt{\frac{\log(|\Pi|)}{n}} + \frac{\log(|\Pi|)}{n}\right)$$

*as long as $\delta \geq |\mathcal{X}| \exp(-\frac{1}{2}\sqrt{n/|\mathcal{X}|})$.*

*Proof.* WLOG, assume $\mathcal{A} = \{0, 1\}$. Consider the following expert: for the $i^{\text{th}}$ prompt, arranged in arbitrary order, let $\pi_e(1|\mathbf{x}_i) = \frac{1}{2i^2}$. Observe that,

$$\widetilde{\sigma}_e = \Theta\left(\frac{1}{|\mathcal{X}|}\right)$$

$$\overline{\sigma}_e = \mathbb{E}_\rho[\sigma_{e,\mathbf{x}}] \leq \mathbb{E}_\rho[\sqrt{\pi_e(1|\mathbf{x})}] = \Theta\left(\frac{\log(|\mathcal{X}|)}{|\mathcal{X}|}\right)$$

$$\sigma_e = \sqrt{\mathbb{E}_\rho[\sigma_{e,\mathbf{x}}^2]} \geq \sqrt{\frac{1}{2}\mathbb{E}_\rho[\pi_e(1|\mathbf{x})]} = \Theta\left(\frac{1}{2\sqrt{|\mathcal{X}|}}\right)$$

For each action, construct the empirical distribution estimator, and return this policy as $\hat{\pi}_n^{\mathrm{vf}}(0|\mathbf{x})$. Then, with probability at least $1 - \delta$, conditioning on the number of samples $n_{\mathbf{x}}$ observed with prompt $\mathbf{x}$,

$$|\hat{\pi}_n^{\mathrm{vf}}(0|\mathbf{x}) - \pi_e(0|\mathbf{x})| \leq \min\left\{1, \sqrt{\frac{\pi_e(0|\mathbf{x})\log(2/\delta)}{n_{\mathbf{x}}}} + \frac{\log(2/\delta)}{n_{\mathbf{x}}}\right\}$$

Therefore, with probability at least $1 - \delta$,

$$\max_{r\in\mathcal{R}} J_r(\pi_e) - J_r(\hat{\pi}_n^{\mathrm{vf}}) = \mathbb{E}_\rho\left[D_{\mathrm{TV}}\left(\hat{\pi}_n^{\mathrm{vf}}(\cdot|\mathbf{x}), \pi_e(\cdot|\mathbf{x})\right)\right]$$

$$\leq \mathbb{E}_\rho\left[\left\{1, \sqrt{\frac{\pi_e(0|\mathbf{x})\log(2|\mathcal{X}|/\delta)}{n_{\mathbf{x}}}} + \frac{\log(2/\delta)}{n_{\mathbf{x}}}\right\}\right] \quad (10)$$

With probability $1 - \delta$, we have that $n_{\mathbf{x}} \geq \frac{n}{|\mathcal{X}|} - \sqrt{\frac{n}{|\mathcal{X}|}\log(1/\delta)}$ for every $\mathbf{x} \in \mathcal{X}$. Assuming $\delta \geq |\mathcal{X}|\exp(-\frac{1}{2}\sqrt{n/|\mathcal{X}|})$, by union bounding, we have that with probability at least $1 - \delta$, for all $\mathbf{x} \in \mathcal{X}$, $n_{\mathbf{x}} \geq \frac{n}{2|\mathcal{X}|}$. Combining with Equation (10), with probability at least $1 - 2\delta$,

$$\max_{r\in\mathcal{R}} J_r(\pi_e) - J_r(\hat{\pi}_n^{\mathrm{vf}}) \leq 2\sum_{\mathbf{x}\in\mathcal{X}}\sqrt{\frac{\pi_e(0|\mathbf{x})\log(|\mathcal{X}|/\delta)}{n|\mathcal{X}|}} + \frac{\log(2/\delta)}{n}$$

$$\leq 2\log(|\mathcal{X}|)\sqrt{\frac{\log(|\mathcal{X}|/\delta)}{n|\mathcal{X}|}} + 2\frac{|\mathcal{X}|\log(2/\delta)}{n}$$

$$\leq 2\widetilde{\sigma}_e \cdot \sqrt{\log(|\Pi|)\frac{\log(|\mathcal{X}|/\delta)}{n}} + \frac{2\log(|\Pi|)\log(2/\delta)}{n}$$

where the last inequality uses the fact that $|\Pi| \geq 2^{\Omega(|\mathcal{X}|)}$ and by construction, the value of $\widetilde{\sigma}_e$.

$\square$

**Lemma C.13.** *For any reward $r \in \mathcal{R}_{1/2}$, there exists another reward $\widetilde{r} \in \mathcal{R}$ such that, for any policy $\pi \in \Pi$ and input distribution $\rho$,*

$$\mathbb{E}_{\rho,\pi}[r(\tau)] = H - \mathbb{E}_{\rho,\pi}[\widetilde{r}(\tau)]$$
$$\mathrm{Var}_{\rho,\pi}[r(\tau)] = \mathrm{Var}_{\rho,\pi}[\widetilde{r}(\tau)]$$

*Proof.* Consider the bi-level reward $r$, and consider the set of minimal states: $\cup_{\tau\in\mathcal{A}^H}\{s_{t^\star}$ where $t^\star = \min\{1 \leq t \leq H : r(s_{t-1}, a_t) > r(s_{t-2}, a_{t-1})\}$. These are the states where a bi-level may be first visited. For each such minimal state, the bi-level property implies that any trajectory which visits this state collects a reward of 1 at every point in time regardless of the sequence of actions played. Based on this construction, we define the reward $\widetilde{r}$ as follows: for every minimal state $s$ which appears at time $t$, consider the subtree rooted at this node (i.e., the set of trajectories which visit this state). Delete this minimal state, and replace it by the set of all $2^{H-t}$ new minimal states corresponding to the set of all states in the subtree at depth $H - t$. Let $\widetilde{r}$ be induced by this new set of minimal states; moreover, it is feasible to construct this set because of the assumption that $r \in \mathcal{R}_{1/2}$: every minimal state appears at some value of $t \leq H/2$.

Consider any trajectory $\tau$. Suppose this trajectory visits a bi-level at time $t \leq H/2$. Now the same trajectory is guaranteed to visit a bi-level at time $H - t \geq H/2$. Thus, $\widetilde{r}(\tau) = H - r(\tau)$, and the assertions about $\mathbb{E}_{\rho,\pi}[\widetilde{r}(\tau)]$ and $\mathrm{Var}_{\rho,\pi}[\widetilde{r}(\tau)]$ follow suit. $\square$

**Lemma C.14.** *For any policy $\pi$ and reward $r$, and $0 \leq \xi \leq \min_{\mathbf{x}\in\mathcal{X}}\frac{\sigma_{e,\mathbf{x}}^2}{4(J_r(\pi_e|\mathbf{x}))^2}$, there exists a class of $2^k$ policies, $\Pi_k = \{\pi_{\mathbf{z}} : \mathbf{z} \in \{0,1\}^k\}$ indexed by binary vectors, and a class of $2^k$ rewards indexed similarly as $\mathcal{R}_k = \{r_{\mathbf{z}} : \mathbf{z} \in \{0,1\}^k\}$, such that,*

1. *For any $\mathbf{z}, \mathbf{z}' \in \{0,1\}^k$, $D_{\chi^2}\left(\pi_{\mathbf{z}}\|\pi_{\mathbf{z}'}\right) \leq 8\xi$. Furthermore, $D_{\chi^2}\left(\pi_{\mathbf{z}}\|\pi_e\right) \leq 8\xi$.*

2. $J_{r_{\mathbf{z}}}(\pi_{\mathbf{z}}) - J_{r_{\mathbf{z}}}(\pi_{\mathbf{z}'}) = \sqrt{\xi} \sum_{i=1}^{k} \mathbb{I}(\mathbf{z}_i \neq \mathbf{z}_i'(\mathbf{x})) \cdot \mathbb{E}_{\mathbf{x} \sim \rho}[\sigma_{e,\mathbf{x}} \mathbb{I}(\mathbf{x} \in \mathcal{X}_i^{\star})]$,

3. *For every reward* $r' \in \mathcal{R}_k$ *and every* $\pi' \in \Pi_k$: $\sigma_{e,\mathbf{x}}^2(\pi', r') \leq \sigma_{e,\mathbf{x}}^2 + H\sigma_{e,\mathbf{x}}\sqrt{\xi} + H^2\xi$.

4. *Recall that* $\pi_e \in \Pi_\varepsilon$, *the* $\varepsilon$-*radius KL ball around* $\pi_b$. *Then, every* $\pi' \in \Pi_k$ *belongs in the ball* $\Pi_{\varepsilon'}$, *where,*

$$\varepsilon' = 3(1+\varepsilon) \cdot \max\left\{ \frac{\sqrt{\xi}H}{\sigma_{\min}}, \frac{\xi H^2}{\sigma_{\min}^2} \right\}. \tag{11}$$

*and where* $\sigma_{\min} = \min_{\mathbf{x} \in \mathcal{X}} \sigma_{e,\mathbf{x}}$.

*Proof.* The policy $\pi_{\mathbf{z}}$ is defined as follows. For each $i \in [k]$ and $\mathbf{x} \in \mathcal{X}_i$,

$$\pi_{\mathbf{z}}(\tau|\mathbf{x}) \propto \begin{cases} (\sigma_{e,\mathbf{x}} + \theta_{\mathbf{x}} r(\tau))\pi_e(\tau|\mathbf{x}), & \text{if } \mathbf{z}_i = 1 \\ \pi_e(\tau|\mathbf{x}), & \text{otherwise.} \end{cases} \tag{12}$$

where $\theta_{\mathbf{x}} \geq 0$ is a parameter to be determined later. Likewise, the reward $r_{\mathbf{z}}$ is defined as follows. For each $\mathbf{x} \in \mathcal{X}_i$,

$$r_{\mathbf{z}}(\tau|\mathbf{x}) \propto \begin{cases} r(\tau), & \text{if } \mathbf{z}_i = 1 \\ \widetilde{r}(\tau|\mathbf{x}), & \text{otherwise.} \end{cases} \tag{13}$$

where $\widetilde{r}$ is the reward defined in Lemma C.13. Since we only care about values and variances, for all intents and purposes, $\widetilde{r}$ is the same as $1 - r$ (which itself may not be a bi-level reward).

**Assertion 1: Bounding the $\chi^2$-divergence between $\pi_{\mathbf{z}}$ and $\pi_{\mathbf{z}'}$.** Consider any pair of binary vectors $\mathbf{z}, \mathbf{z}' \in \{0,1\}^k$. If $\mathbf{z}_i = \mathbf{z}_i'$, then $D_{\chi^2}(\pi_{\mathbf{z}}(\cdot|\mathbf{x})\|\pi_{\mathbf{z}'}(\cdot|\mathbf{x})) = 0$ for any $\mathbf{x} \in \mathcal{X}_i$. Otherwise, if $\mathbf{z}_i = 1$ and $\mathbf{z}_i' = 0$, for any $\mathbf{x} \in \mathcal{X}_i$,

$$\begin{aligned} D_{\chi^2}(\pi_{\mathbf{z}}(\cdot|\mathbf{x})\|\pi_{\mathbf{z}'}(\cdot|\mathbf{x})) &= D_{\chi^2}(\pi_{\mathbf{z}}(\cdot|\mathbf{x})\|\pi_e(\cdot|\mathbf{x})) \\ &= \frac{\mathbb{E}_{\pi_e}[(\sigma_{e,\mathbf{x}} + \theta_{\mathbf{x}} r(\tau))^2|\mathbf{x}]}{\mathbb{E}_{\pi_e}[\sigma_{e,\mathbf{x}} + \theta_{\mathbf{x}} r(\tau)|\mathbf{x}]^2} - 1 \\ &= \frac{\sigma_{e,\mathbf{x}}^2 + 2\theta_{\mathbf{x}}\sigma_{e,\mathbf{x}}J_r(\pi_e|\mathbf{x}) + \theta_{\mathbf{x}}^2((J_r(\pi_e|\mathbf{x}))^2 + \sigma_{e,\mathbf{x}}^2)}{(\sigma_e + \theta_{\mathbf{x}}J_r(\pi_e|\mathbf{x}))^2} - 1 \\ &= \frac{\theta_{\mathbf{x}}^2\sigma_{e,\mathbf{x}}^2}{(\sigma_{e,\mathbf{x}} + \theta_{\mathbf{x}}J_r(\pi_e|\mathbf{x}))^2} \\ &= \xi \end{aligned} \tag{14}$$

where the last equation follows by choosing $\theta_{\mathbf{x}}$ such that $\theta_{\mathbf{x}}\sigma_{e,\mathbf{x}} = \sqrt{\xi}(\sigma_{e,\mathbf{x}} + \theta_{\mathbf{x}}J_r(\pi_e|\mathbf{x}))$. There will always exist a feasible choice of $\theta_{\mathbf{x}} \geq 0$ satisfying this equation as long as the condition $\sqrt{\xi} \leq \sigma_{e,\mathbf{x}}/J_r(\pi_e|\mathbf{x})$ is satisfied, and under the stronger restriction $\sqrt{\xi} \leq \sigma_{e,\mathbf{x}}/2J_r(\pi_e|\mathbf{x})$ we will have that $\theta_{\mathbf{x}} \leq 2\sqrt{\xi}$. On the other hand, if $\mathbf{z}(\mathbf{x}) = 0$ and $\mathbf{z}'(\mathbf{x}) = 1$, for any $\mathbf{x} \in \mathcal{X}_i$,

$$\begin{aligned} D_{\chi^2}(\pi_{\mathbf{z}}(\cdot|\mathbf{x})\|\pi_{\mathbf{z}'}(\cdot|\mathbf{x})) &= D_{\chi^2}(\pi_e(\cdot|\mathbf{x})\|\pi_{\mathbf{z}}(\cdot|\mathbf{x})) \\ &= \mathbb{E}_\pi[\sigma_{e,\mathbf{x}} + \theta_{\mathbf{x}} r(\tau)|\mathbf{x}] \cdot \mathbb{E}_\pi\left[\frac{1}{\sigma_{e,\mathbf{x}} + \theta_{\mathbf{x}} r(\tau)}\bigg|\mathbf{x}\right] - 1 \\ &= \mathbb{E}_\pi\left[\frac{\sigma_{e,\mathbf{x}} + \theta_{\mathbf{x}}J_r(\pi_e|\mathbf{x})}{\sigma_{e,\mathbf{x}} + \theta_{\mathbf{x}} r(\tau)}\bigg|\mathbf{x}\right] - 1 \\ &= \mathbb{E}_\pi\left[\frac{\theta_{\mathbf{x}}(J_r(\pi_e|\mathbf{x}) - r(\tau))}{\sigma_{e,\mathbf{x}} + \theta_{\mathbf{x}} r(\tau)}\bigg|\mathbf{x}\right] \\ &\overset{(i)}{\leq} 2\theta_{\mathbf{x}}^2 \\ &\leq 8\xi \end{aligned} \tag{15}$$

where $(i)$ follows from Lemma C.4 and the last inequality relies on the choice of $\theta_{\mathbf{x}} \leq 2\sqrt{\xi}$. Combining Equations (14) and (15) with an expectation over $\mathbf{x} \sim \rho$ results in a proof of the first assertion.

**Assertion 2: Bounding the value gap.** Observe that $J_r(\pi_{\mathbf{z}}|\mathbf{x}) - J_r(\pi_{\mathbf{z}'}|\mathbf{x}) = 0$ for any $\mathbf{x} \in \mathcal{X}_i$ if $z_i = z_i'$. In case $z_i = 1$ and $z_i' = 0$ and any $\mathbf{x} \in \mathcal{X}_i$, $r_{\mathbf{z}}(\tau) = r(\tau)$ for any $\tau$ which visits $\mathbf{x}$ and,

$$
\begin{aligned}
J_{r_{\mathbf{z}}}(\pi_{\mathbf{z}}|\mathbf{x}) - J_{r_{\mathbf{z}}}(\pi_{\mathbf{z}'}|\mathbf{x}) &= \frac{\mathbb{E}_\pi[\sigma_e r(\tau) + \theta_{\mathbf{x}}(r(\tau))^2|\mathbf{x}]}{\mathbb{E}_\pi[\sigma_{e,\mathbf{x}} + \theta_{\mathbf{x}} r(\tau)|\mathbf{x}]} - J_{r_{\mathbf{z}}}(\pi_e|\mathbf{x}) \\
&= \frac{\sigma_{e,\mathbf{x}} J_r(\pi_e|\mathbf{x}) + \theta_{\mathbf{x}}(J_r(\pi_e|\mathbf{x}))^2 + \sigma_{e,\mathbf{x}}^2)}{\sigma_{e,\mathbf{x}} + \theta_{\mathbf{x}} J_r(\pi_e|\mathbf{x})} - J_r(\pi_e|\mathbf{x}) \\
&= \frac{\theta_{\mathbf{x}} \sigma_{e,\mathbf{x}}^2}{\sigma_{e,\mathbf{x}} + \theta_{\mathbf{x}} J_r(\pi_e|\mathbf{x})} \\
&= \sigma_{e,\mathbf{x}} \sqrt{\xi}
\end{aligned}
$$
(16)

where the last equation follows by choice of $\theta_{\mathbf{x}}$. When $z_i = 0$ and $z_i' = 1$, the same analysis results in the same bound $J_r(\pi_{\mathbf{z}}|\mathbf{x}) - J_r(\pi_{\mathbf{z}'}|\mathbf{x}) = \sigma_{e,\mathbf{x}}\sqrt{\xi}$ for any $\mathbf{x} \in \mathcal{X}_i$, and taking an expectation over $\mathbf{x} \sim \rho$ proves the second assertion.

**Assertion 3: Bound on variance of $\pi_{\mathbf{z}}$.** This follows from Equation (5), which bounds the variance of a policy which lies within a radius $\kappa$ $\chi^2$ ball of another: in particular, $\pi_{\mathbf{z}}(\cdot|\mathbf{x})$ lies in a $\xi$-sized KL ball around $\pi_e(\cdot|\mathbf{x})$, which has variance $\sigma_{e,\mathbf{x}}^2$, and taking an expectation over $\mathbf{x} \sim \rho$. Note also that the reward $r_{\mathbf{z}}$ preserves variances across policies compared to $r$ (cf. Lemma C.13 and the fact that $r_{\mathbf{z}}$ uses either $r$ or $\widetilde{r}$), so it suffices to carry out the variance computation under $r$.

**Assertion 4: Bound on $D_{\chi^2}(\pi\|\pi_b)$ for $\pi \in \Pi_k$.** For any $\mathbf{z} \in \{0,1\}^k$, note that $\pi_{\mathbf{z}}$ and $\pi_e$ have density ratio upper bounded by,

$$
\begin{aligned}
\left\|\frac{\pi_{\mathbf{z}}(\tau|\mathbf{x})}{\pi_e(\tau|\mathbf{x})}\right\|_\infty &\leq \frac{\sigma_{e,\mathbf{x}} + \theta_{\mathbf{x}} H}{\sigma_{e,\mathbf{x}} + \theta_{\mathbf{x}} J_r(\pi_e|\mathbf{x})} \\
&\leq 1 + \frac{2\sqrt{\xi}H}{\sigma_{\min}}
\end{aligned}
$$

This upper bound on the density ratio implies that,

$$
\begin{aligned}
D_{\chi^2}(\pi_{\mathbf{z}}\|\pi_b) &= \mathbb{E}_{\mathbf{x}\sim\rho}\left[D_{\chi^2}(\pi_{\mathbf{z}}(\cdot|\mathbf{x})\|\pi_b(\cdot|\mathbf{x}))\right] \\
&\leq \left(1 + \frac{2\sqrt{\xi}H}{\sigma_{\min}}\right)^2 (1 + D_{\chi^2}(\pi_e\|\pi_b)) - 1 \\
&\leq 3(1+\varepsilon) \cdot \max\left\{\frac{\sqrt{\xi}H}{\sigma_{\min}}, \frac{\xi H^2}{\sigma_{\min}^2}\right\}
\end{aligned}
$$

$\square$

**Lemma C.15.** *There exists a subset $\mathcal{Z} \subseteq \{0,1\}^k$ with $|\mathcal{Z}| = \lceil 2^{k/4} \rceil$ and such that every pair $\mathbf{z}, \mathbf{z}' \in \mathcal{Z}$ satisfies,*

$$
\sum_{i=1}^k \mathbb{I}(z_i \neq z_i') \geq k/4
$$

*Proof.* This statement essentially follows from the Gilbert-Varshamov bound (cf. Theorem 5.2.6 in (Ling & Xing, 2004)).

$\square$

### C.3.4. CONSTRUCTION OF POLICY CLASS $\Pi'$ AND REWARD CLASS $\mathcal{R}'$

Consider the set of policies $\Pi' = \{\pi_{\mathbf{z}} : \mathbf{z} \in \mathcal{Z}\} \subseteq \Pi_k$ and $\mathcal{R}' = \{r_{\mathbf{z}} : \mathbf{z} \in \mathcal{Z}\}$ (see the proof of Lemma C.14 for a definition of $\pi_{\mathbf{z}}, \Pi_k$ and $\mathcal{R}_k$). By Lemma C.15, $|\Pi'| \approx 2^{k/4}$, and furthermore, for any $\mathbf{z}, \mathbf{z}' \in \mathcal{Z}$,

$$
J_{r_{\mathbf{z}}}(\pi_{\mathbf{z}}) - J_{r_{\mathbf{z}}}(\pi_{\mathbf{z}'}) \geq L_k^\star \sqrt{\xi}
$$
(17)

where $L_k^\star$ is defined in Equation (6). This bound follows from the first assertion in Lemma C.14 and the fact that $\mathbf{z}$ and $\mathbf{z}'$ differ in at least $k/4$ coordinates; $L_k^\star$, by definition, captures the deviation for the worst-case choice of $k/4$ coordinates.

**Definition C.16** ((Chen et al., 2016; Rajaraman et al., 2024))**.** The $\chi^2$-informativity is defined as,

$$I_{\chi^2}(X;Y) \triangleq \inf_{Q_Y} \chi^2 \left( P_{XY} \| P_X \times Q_Y \right)$$

**Theorem C.17.** *Consider the family of policies $\Pi'$ defined above. Let $p_{\Pi'}$ denote the uniform prior over them (alternately, the distribution over $\pi_z$ for $z \sim \mathrm{Unif}(\mathcal{Z})$). Let the policy $\hat{\pi}$ be constructed via a dataset $D$ and assume that the verifier-free learner is realizable, satisfying $\hat{\pi}_n^{vf} \in \Pi'$. Then,*

$$\Pr(J_{r_z}(\pi_z) - J_{r_z}(\hat{\pi}_n^{vf}) \geq L_k^\star \sqrt{\xi}) \geq 1 - \frac{1}{|\Pi'|} \sqrt{I_{\chi^2}(z;D) + 1}$$

*Proof.* Let $P$ be the joint distribution of $z$ and $D$. Let $Q$ be the distribution $\mathrm{Unif}(\mathcal{Z}) \times Q_{\mathrm{data}}$ for a generic (arbitrary) data distribution $Q_{\mathrm{data}}$. Let $T : (z, D) \mapsto \mathbb{I}(J_{r_z}(\pi_z) - J_{r_z}(\hat{\pi}_n^{vf}) \geq L_k^\star \sqrt{\xi})$ be a generic map, and $P \circ T^{-1}$ and $Q \circ T^{-1}$ be the pushforward measures of $P$ and $Q$ by $T$. Letting $\mathcal{E}(z, D) = \{J_{r_z}(\pi_z) - J_{r_z}(\hat{\pi}_n^{vf}) \geq L_k^\star \sqrt{\xi}\}$, the data-processing inequality gives,

$$\begin{aligned}
D_{\chi^2}\left(P \| Q\right) &\geq D_{\chi^2}\left(P \circ T^{-1} \| Q \circ T^{-1}\right) \\
&= \frac{(P(\mathcal{E}(z, D)) - Q(\mathcal{E}(z, D)))^2}{Q(\mathcal{E}(z, D))(1 - Q(\mathcal{E}(z, D)))}
\end{aligned} \tag{18}$$

Let us assume that the learner's policy $\hat{\pi}$ is realizable, and satisfies $\hat{\pi} \in \Pi'$. By the product structure of $Q$, we have that,

$$Q(\mathcal{E}(z, D)) \leq \sup_{\pi \in \Pi'} \Pr\left( J_{r_z}(\pi_z) - J_{r_z}(\pi) \geq L_k^\star \sqrt{\xi} \right) = 1 - \frac{1}{|\Pi'|}.$$

where the last inequality uses the fact that for any $z' \neq z$, $J_{r_z}(\pi_z) - J_{r_z}(\pi_{z'}) \geq L_k^\star \sqrt{\xi}$ (cf. Equation (17)). Combining with Equation (18), rearranging, simplifying and taking the infimum over $Q_{\mathrm{data}}$ completes the proof. $\square$

**Lemma C.18.** *Consider any realizable verifier-free learner, satisfying $\hat{\pi}_n^{vf} \in \Pi'$. Then,*

$$\Pr\left( J_{r_z}(\pi_z) - J_{r_z}(\hat{\pi}_n^{vf}) \geq L_k^\star \sqrt{\frac{\log(|\Pi'|)}{16n}} \right) \geq \frac{1}{4}$$

*Proof.* Observe that,

$$\begin{aligned}
I_{\chi^2}(z;D) + 1 &= \inf_{Q_{\mathrm{data}}} \int \left[ \frac{(p_\Pi(\pi_z))^2 \left(\prod_{\tau \in D} \pi_z(\tau)\right)^2}{p_\Pi(\pi_z) Q_{\mathrm{data}}(D)} \right] dD d\pi \\
&\overset{(i)}{\leq} \int \left[ \frac{p_\Pi(\pi_z) \left(\prod_{\tau \in D} \pi_z(\tau)\right)^2}{\prod_{\tau \in D} \pi_e(\tau)} \right] dD d\pi \\
&= \int \left[ \frac{p_\Pi(\pi_z) \left(\prod_{\tau \in D} \pi_z(\tau)\right)^2}{\prod_{\tau \in D} \pi_e(\tau)} \right] dD d\pi \\
&= \mathbb{E}_{\pi \sim p_\Pi}[(1 + D_{\chi^2}\left(\pi_z \| \pi_e\right))^n] \\
&\overset{(ii)}{\leq} (1 + 8\xi)^n
\end{aligned}$$

where in $(i)$ we choose $Q_{\mathrm{data}}$ as the data distribution realized by $\pi_e$ and in $(ii)$, we use the first assertion of Lemma C.14. Choose $\xi = \varepsilon_{\mathrm{stat}} = \frac{\log(|\Pi'|)}{16n}$, we get,

$$\Pr\left( J_{r_z}(\pi_z) - J_{r_z}(\hat{\pi}_n^{vf}) < L_k^\star \sqrt{\frac{\log(|\Pi'|)}{n}} \right) \geq \frac{1}{4}$$

$\square$

## C.4. Bounding the performance of Algorithm 1

### C.4.1. UNDERSTANDING THE ANTI-CONCENTRATION ASSUMPTION

Recall that the anticoncentration assumption controls the probability of the reward $r(\tau)$ for $\tau \sim \pi_b(\cdot|\mathbf{x})$ of exceeding its mean by a margin of $\sqrt{\varepsilon}$ times its standard deviation. Namely,

$$c_\mathbf{x}(\varepsilon) =: \Pr_{\pi_b(\cdot|\mathbf{x})}\big(r(\tau) \geq \mathbb{E}_{\pi_b(\cdot|\mathbf{x})}\left[r(\tau)\right] + \sigma_{b,\mathbf{x}}\sqrt{\varepsilon}\big).$$

The interpretation of $c_\mathbf{x}(\varepsilon)$ is natural, as a prompt-conditional measure of anticoncentration of the rewards $r(\tau)$ collected by the base policy. However, as we discuss in the next lemma, the deviation term $\mathbb{E}_{\pi_b(\cdot|\mathbf{x})}\left[r(\tau)\right] + \sigma_{b,\mathbf{x}}\sqrt{\varepsilon}$ serves a dual purpose: it precisely captures the maximum value achievable in a $\chi^2$ ball around $\pi_b$ of radius $\varepsilon$.

**Lemma C.19** (Characterizing the optimal value within the $\chi^2$ ball). *For a single prompt $\mathbf{x} \in \mathcal{X}$, consider the set of policies* $\Pi_{\varepsilon,\mathbf{x}} = \{\pi : D_{\chi^2}\left(\pi(\cdot|\mathbf{s})\|\pi_b(\cdot|\mathbf{x})\right) \leq \varepsilon\}$. *Then,*

$$\sup_{\pi \in \Pi_{\varepsilon,\mathbf{x}}} \mathbb{E}_{\tau \sim \pi(\cdot|\mathbf{x})}[r(\tau)] \geq \mathbb{E}_{\pi_b(\cdot|\mathbf{x})}[r(\tau)] + \sigma_{b,\mathbf{x}}\sqrt{\varepsilon}. \tag{19}$$

*Furthermore, as long as $\varepsilon \leq \frac{\sigma_{b,\mathbf{x}}^2}{(J_r(\pi_b|\mathbf{x}))^2}$, this inequality is an equality.*

*Proof.* Consider the candidate policy $\pi(\tau|\mathbf{x}) \propto (\sigma_{b,\mathbf{x}} + \theta r(\tau))\pi_b(\cdot|\mathbf{x})$ for $\theta$ to be chosen later. Mirroring the calculation in Equation (14) (with $\pi_e$ replaced by $\pi_b$), we see that,

$$D_{\chi^2}\left(\pi(\cdot|\mathbf{x})\|\pi_b(\cdot|\mathbf{x})\right) = \frac{\theta^2 \sigma_{b,\mathbf{x}}^2}{(\sigma_{b,\mathbf{x}} + \theta J_r(\pi_b|\mathbf{x}))^2}$$

The maximum achievable value of the $\chi^2$ divergence by this policy is $\frac{\sigma_{b,\mathbf{x}}^2}{(J_r(\pi_b|\mathbf{x}))^2}$. Likewise, mirroring the calculation in Equation (16),

$$J_r(\pi|\mathbf{x}) - J_r(\pi_b|\mathbf{x}) = \frac{\theta \sigma_{b,\mathbf{x}}^2}{\sigma_\mathbf{x} + \theta J_r(\pi_b|\mathbf{x})} = \sigma_{b,\mathbf{x}}\sqrt{D_{\chi^2}\left(\pi(\cdot|\mathbf{x})\|\pi_b(\cdot|\mathbf{x})\right)} = \sigma_{b,\mathbf{x}}\sqrt{\varepsilon}$$

Therefore, with the appropriate choice of $\theta$, this policy is a feasible policy achieving the supremum in the statement. What remains is to show that the supremum can be no larger. By Lemma C.2, with the choice of $Y = r(\tau)$, $P$ as the distribution over $\tau$ induced by $\pi(\cdot|\mathbf{x})$ and $Q$ the distribution over trajectories induced by $\pi_b(\cdot|\mathbf{x})$. Then,

$$\left|\mathbb{E}_{\tau \sim \pi(\cdot|\mathbf{x})}[r(\tau)] - \mathbb{E}_{\tau \sim \pi_b(\cdot|\mathbf{x})}[r(\tau)]\right| \leq \sqrt{\mathrm{Var}_{\tau \sim \pi_b(\cdot|\mathbf{x})}[r(\tau)] \cdot D_{\chi^2}\left(\pi(\cdot|\mathbf{x})\|\pi_b(\cdot|\mathbf{x})\right)} = \sigma_{b,\mathbf{x}}\sqrt{\varepsilon}$$

This shows that the supremizing value is exactly $\sigma_{b,\mathbf{x}}\sqrt{\varepsilon}$. $\qquad\square$

**Property C.20** (Regularity). Assume that for each $\mathbf{x} \in \mathcal{X}$ that $J_r(\pi_b|\mathbf{x}) > 0$ and,

$$\varepsilon_\mathbf{x} =: D_{\chi^2}\left(\bar{\pi}_\kappa(\cdot|\mathbf{x})\|\pi_b(\cdot|\mathbf{x})\right) \leq \frac{\sigma_{b,\mathbf{x}}^2}{(J_r(\pi_b|\mathbf{x}))^2}.$$

where $\bar{\pi}_\kappa$ is any policy which collects the maximum value, while remaining within $\Pi_\kappa$.

**Lemma C.21.** *Suppose $\pi_b$ is $c_0$-anticoncentrated for some problem horizon $h_0$ and assume that Property C.20 holds true for the base policy at this value of $h_0$. Define a collection of parameters, $\lambda = \{\lambda_\mathbf{x} : \mathbf{x} \in \mathcal{X}\}$ where $\mathbb{R} \ni \lambda_\mathbf{x} \in (0, \sigma_b\sqrt{2/c_0}]$. Then, there exists a policy $\pi_c$ such that,*

1. *Almost surely, $r(\tau) > 0$ for $\tau \sim \pi_c(\cdot|\mathbf{x})$ and any $\mathbf{x} \in \mathcal{X}$.*

2. *$\pi_c$ is no worse than $\pi_e$. Namely, $J_r(\pi_c) \geq \sup_{\pi \in \Pi_\kappa} J_r(\pi) \geq J_r(\pi_e)$.*

3. *For every $\mathbf{x} \in \mathcal{X}$, $\sup_{\tau : \Pr_{\pi_b}(\tau|\mathbf{x}) > 0} \frac{\Pr_{\pi_c}(\tau|\mathbf{x})}{\Pr_{\pi_b}(\tau|\mathbf{x})} \leq c_0^{-1}$*

*Proof.* Fix a prompt $\mathbf{x} \in \mathcal{X}$. We will construct $\pi_c$ separately for each prompt and later argue about each of these three assertions. Since $\pi_b$ is $c_0$-anticoncentrated for some problem horizon $h_0$, as long as $\varepsilon_{\mathbf{x}} =: D_{\chi^2}\left(\bar{\pi}_\kappa(\cdot|\mathbf{x})\|\pi_b(\cdot|\mathbf{x})\right) \leq \frac{\sigma_{b,\mathbf{x}}^2}{(J_r(\pi_b|\mathbf{x}))^2}$, by Lemma C.19, defining $\mathcal{T}$ as the set of trajectories $\{r(\tau) \geq \sup_{\pi \in \Pi_{\varepsilon_{\mathbf{x}},\mathbf{x}}} \mathbb{E}_{\tau \sim \pi(\cdot|\mathbf{x})}[r(\tau)]\}$. Then,

$$\Pr_{\tau \sim \pi_b(\cdot|\mathbf{x})}(\tau \in \mathcal{T}) \geq c_0 \tag{20}$$

Consider the policy $\pi_c(\cdot|\mathbf{x})$ which is the mixture over the trajectories $\mathcal{T} = \{\tau : r(\tau) \geq \mathbb{E}_{\tau \sim \pi_b(\cdot|\mathbf{x})}[r(\tau)] + \sigma_{b,\mathbf{x}}\sqrt{\varepsilon_{\mathbf{x}}}\}$ with mixture weights $w_\tau \propto \Pr_{\pi_b(\cdot|\mathbf{x})}(\tau)$. Since the MDP is autoregressive (i.e., tree-like), $\pi_c(\cdot|\mathbf{x})$ corresponds to a simple policy (as opposed to a mixture over policies), since two trajectories in $\mathcal{T}$ can not visit the same state again after a different action is played between them, i.e., a breakpoint. This implies that the mixture of these two trajectories is the same as the policies which agrees with them until the breakpoint and picks one of the trajectories to follow at the breakpoint, proportional to its weight. The same argument applies when considering a mixture over more than two trajectories. Next, we prove the three assertions of this lemma.

**Assertion 1: Rewards are strictly positive.** $\pi_c(\cdot|\mathbf{x})$ is only supported on trajectories which collect rewards which exceed $\sup_{\pi \in \Pi_{\varepsilon_{\mathbf{x}},\mathbf{x}}} \mathbb{E}_{\tau \sim \pi(\cdot|\mathbf{x})}[r(\tau)] \geq \mathbb{E}_{\tau \sim \pi_b(\cdot|\mathbf{x})}[r(\tau)]$. By Property C.20, we have that $\mathbb{E}_{\tau \sim \pi_b(\cdot|\mathbf{x})}[r(\tau)] > 0$; this implies that the reward collected by every such trajectory is not only strictly positive, but must be at least 1 (by the bi-level property of the rewards).

**Assertion 2: Value bound.** $\pi_c(\cdot|\mathbf{x})$ is supported on trajectories which collect reward at least:

$$\sup_{\pi \in \Pi_{\varepsilon_{\mathbf{x}},\mathbf{x}}} \mathbb{E}_{\tau \sim \pi(\cdot|\mathbf{x})}[r(\tau)].$$

Thus, with probability 1, for any trajectory $\tau$ sampled from $\pi_c(\cdot|\mathbf{x})$, $r(\tau) \geq \sup_{\pi \in \Pi_{\varepsilon_{\mathbf{x}},\mathbf{x}}} \mathbb{E}_{\tau \sim \pi(\cdot|\mathbf{x})}[r(\tau)]$. Taking an expectation over $\tau \sim \pi_c(\cdot|\mathbf{x})$, we get, $\mathbb{E}_{\tau \sim \pi_c(\cdot|\mathbf{x})}[r(\tau)] \geq \sup_{\pi \in \Pi_{\varepsilon_{\mathbf{x}},\mathbf{x}}} \mathbb{E}_{\tau \sim \pi(\cdot|\mathbf{x})}[r(\tau)]$. Further, taking an expectation over $\mathbf{x} \sim \rho$,

$$\begin{aligned}
\mathbb{E}_{\rho,\pi_c}[r(\tau)] &\geq \mathbb{E}_{\mathbf{x} \sim \rho}\left[\sup_{\pi \in \Pi_{\varepsilon_{\mathbf{x}},\mathbf{x}}} \mathbb{E}_{\tau \sim \pi(\cdot|\mathbf{x})}[r(\tau)]\right] \\
&\geq \sup_{\pi \in \bigcap_{\mathbf{x} \in \mathcal{X}} \Pi_{\varepsilon_{\mathbf{x}},\mathbf{x}}} \mathbb{E}_{\rho,\pi}[r(\tau)] \\
&= \sup_{\pi \in \Pi_\kappa} \mathbb{E}_{\rho,\pi}[r(\tau)]
\end{aligned}$$

where the last equation follows by definition of $\varepsilon_{\mathbf{x}}$ (cf. Property C.20).

**Assertion 3: Bounds on coverage.** Note that $\pi_c(\cdot|\mathbf{x})$ is the policy $\sum_{\tau \in \mathcal{T}} w_\tau \delta_\tau$. In particular, for any trajectory $\tau$ in the support of $\pi_c(\cdot|\mathbf{x})$,

$$\frac{\Pr_{\pi_c}(\tau|\mathbf{x})}{\Pr_{\pi_b}(\tau|\mathbf{x})} = \frac{w_\tau}{\Pr_{\pi_b(\cdot|\mathbf{x})}(\tau)} = \frac{1}{\sum_{\tau \in \mathcal{T}} \Pr_{\pi_b(\cdot|\mathbf{x})}(\tau)} \tag{21}$$

where the last equation follows by definition of $w_\tau$. By Equation (20), $\sum_{\tau \in \mathcal{T}} \Pr_{\pi_b(\cdot|\mathbf{x})}(\tau) \geq c_0$. This completes the proof of the last assertion. $\qquad \square$

**Lemma C.22.** *Suppose $\pi_b$ is $c_0$-anticoncentrated for some problem horizon $h_0$ and assume that Property C.20 holds true for the base policy $\pi_b$ at this value of $h_0$. Consider the policy $\pi_c$ introduced in Lemma C.21 at this value $h_0$. For any horizon $H > h_0$, there exists a policy $\widetilde{\pi}_c$ which satisfies essentially the same conditions,*

1. *Almost surely, $r(\tau) > 0$ for $\tau \sim \widetilde{\pi}_c(\cdot|\mathbf{x})$ for any $\mathbf{x} \in \mathcal{X}$,*

2. *$\widetilde{\pi}_c$ is no worse than $\pi_e$ when deployed on horizon $H$. Namely, $J_r^H(\widetilde{\pi}_c) \geq \sup_{\pi \in \Pi_\kappa^H} J_r^H(\pi) \geq J_r^H(\pi_e)$.*

3. *$\sup_{\tau : \Pr_{\pi_b}(\tau|\mathbf{x})>0} \frac{\Pr_{\pi_c}(\tau|\mathbf{x})}{\Pr_{\pi_b}(\tau|\mathbf{x})} \leq c_0^{-1}$.*

*Here, we point out that the in the third assertion (coverage), $(a)$ trajectories $\tau$ are of length $H$, and $(b)$ the variance term $\sigma_b(h_0)$ that appears is that of the base policy evaluated on the horizon $h_0$. Everywhere, we take care to superscript $J_r$ and $\Pi_\kappa$ to indicate the horizon over which the policies are considered.*

*Proof.* Consider the "extension" of $\pi_c$, defined till time $h_0$, by $\pi_b$ (which we assume is defined for every $t \in \mathbb{N}$). Namely, consider the policy $\widetilde{\pi}_c$ which follows $\pi_c$ till time $h_0$ and plays actions according to $\pi_b$ thereon.

The first three assertions follow from the fact that $\pi_c$ is only supported on trajectories with strictly positive reward. By the bi-level property, each of these trajectories collect 1 unit of reward at every $t > h_0$. Thus, $J_r^H(\widetilde{\pi}_c) = J_r^{h_0}(\widetilde{\pi}_c) + (H - h_0)$, while $\sup_{\pi \in \Pi_\kappa^H} J_r^H(\pi) \leq \sup_{\pi \in \Pi_\kappa^{h_0}} J_r^H(\pi) + (H - h_0)$. This follows from the fact that the supremizing policy for the $H$ horizon problem can be truncated to the first $h_0$ steps to result in a candidate policy in $\Pi_\kappa^{h_0}$; in the process the value of the policy decreases by at most $H - h_0$. The last assertion follows from the fact that $\widetilde{\pi}_c$ and $\pi_b$ agree after time $h_0$, so the worst-case density ratio cannot increase as $H$ increases beyond $h_0$. $\square$

### C.4.2. ANALYSIS OF ALGORITHM 1: PROOF OF THEOREM 5.7

Below, we provide implementation details of Algorithm 1 and a slightly more formal version of Theorem 5.7. We will define the confidence set $\hat{R}_\gamma$ below, and choose $\gamma$ appropriately as any upper bound to $\mathbf{Est}_n^{\mathrm{Off}}(\delta)$ (see Equation (22)). One such upper bound is provided in Lemma C.25. For the purpose of this section, we will assume that Algorithm 1 carries out least square estimation with respect to some reward class $\mathcal{R}_{\mathrm{vb}}$ such that $r$ belongs to this class, and may be a subset or superset of the set of all bi-level rewards, $\mathcal{R}$.

**Theorem C.23** (Formal version of Theorem 5.7). *Consider a bi-level reward $r$, base policy $\pi_b$ that is $c_0$-anticoncentrated at some horizon $h_0 \leq H$ and assume that Property C.20 is satisfied at $h_0$. Suppose the verifier is used to label the cumulative reward of every trajectory and results in a dataset of noisy reward annotations, $\{(\mathbf{x}_i, \tau_i, y_i)\}_{i=1}^n$: assume that the reward annotations are of the form $y_i = r(\tau_i) + Z_i$ where the $Z_i$'s are independent and standard normal with trajectory level variance $\mathrm{Var}[Z_i] \leq \sigma_{noise}^2$. Then, the policy $\hat{\pi}_n^{\mathrm{vb}}$ returned by Algorithm 1, the suboptimality gap w.r.t. the best expert $\bar{\pi}_\kappa \in \Pi_\kappa$ satisfies: with probability $\geq 1 - \delta$,*

$$J_r(\bar{\pi}_\kappa) - J_r(\hat{\pi}_n^{\mathrm{vb}}) \lesssim \frac{(H + \sigma_{noise}^2) \log(|\mathcal{R}_{vb}|/\delta)}{n c_0},$$

*With independent $O(1)$-variance noise at steps of a trajectory, note that $\sigma_{noise}^2 \leq O(H)$.*

Below we instantiate the confidence set $\hat{R}_\gamma$ in Algorithm 1. Recall that we assume that Algorithm 1 carries out least square estimation with respect to some reward class $\mathcal{R}_{\mathrm{vb}}$: with $\hat{r}_{\mathrm{ls}}$ as the least squares estimator,

$$\hat{r}_{\mathrm{ls}} \leftarrow \inf_{r' \in \mathcal{R}_{\mathrm{vb}}} \frac{1}{n} \sum_{i=1}^n (r'(\tau_i) - y_i)^2$$

$$\widetilde{R}_\gamma = \left\{ r' \in \mathcal{R}_{\mathrm{vb}} \,\middle|\, \frac{1}{n} \sum_{i=1}^n (r'(\tau_i) - \hat{r}_{\mathrm{ls}}(\tau_i))^2 \leq \gamma \right\}$$

$$\hat{R}_\gamma = \left\{ \{\mathsf{round}(r'(\cdot))\} : r' \in \widetilde{R}_\gamma \right\}$$

Where $\mathsf{round}(r(\cdot))$ is the "rounding" of the reward $r$, for every $\tau$, $r(\tau)$ is rounded to the nearest integer, breaking ties arbitrarily. We define the offline estimation error of the least-squares estimator below. Define $\mathcal{E}_\delta$ as the event,

$$\frac{1}{n} \sum_{i=1}^n (\hat{r}_{\mathrm{ls}}(\tau_i) - r(\tau_i))^2 \leq \mathbf{Est}_n^{\mathrm{Off}}(\delta) \tag{22}$$

And suppose $\Pr(\mathcal{E}_\delta) \geq 1 - \delta$ where the probability is computed over the randomness of the training dataset $\{(\mathbf{x}_i, \tau_i)\}_{i=1}^n$.

The analysis of the verifier-based learner in Algorithm 1 follows the standard analysis of pessimism-based algorithms. For

an arbitrary comparator policy $\pi_c$,

$$
\begin{aligned}
J_r(\pi_c) - J_r(\hat{\pi}_n^{\text{vb}}) &\leq J_r(\pi_c) - \min_{\hat{r} \in \hat{R}_\gamma} J_{\hat{r}}(\hat{\pi}_n^{\text{vb}}) \\
&\leq J_r(\pi_c) - \min_{\hat{r} \in \hat{R}_\gamma} J_{\hat{r}}(\pi_c) \\
&\leq \sup_{\hat{r} \in \hat{R}_\gamma} \mathbb{E}_{\rho, \pi_c} \left[ |r(\tau) - \hat{r}(\tau)| \right]
\end{aligned}
\tag{23}
$$

With the choice of the comparator policy $\pi_c = \widetilde{\pi}_c$, as defined in Lemma C.22,

$$
\sup_{\pi \in \Pi_\kappa} J_r(\pi) - J_r(\hat{\pi}_n^{\text{vb}}) \leq \sup_{\hat{r} \in \hat{R}_\gamma} c_0^{-1} \mathbb{E}_{\rho, \pi_b} \left[ |r(\tau) - \hat{r}(\tau)| \right].
$$

where note that the base policy is assumed to be $c_0$-anticoncentrated for the horizon $h_0$. The performance of the algorithm thus relies on establishing a generalization bound for the reward estimation problem, which is proved below in Theorem C.24. In conjunction, this results in the upper bound: with probability $1 - \delta$,

$$
\sup_{\pi \in \Pi_\kappa} J_r(\pi) - J_r(\hat{\pi}_n^{\text{vb}}) \leq \mathcal{O} \left( \frac{(H + \sigma_{\text{noise}}^2) \cdot \log(|\mathcal{R}_{\text{vb}}|/\delta)}{c_0 n} \right)
$$

**Theorem C.24.** *Recall that the reward annotations are of the form $y_i = r(\tau_i) + Z_i$ where the noise $Z_i$ is assumed to be independent and standard normal with trajectory level variance $\sigma_{noise}^2$. Consider any $\delta \in (0, 1)$. Then, with probability $1 - \delta$, simultaneously for all $r' \in \hat{R}_\gamma$,*

$$
\mathbb{E}_{\rho, \pi_b}[|r(\tau) - r'(\tau)|] \leq \mathcal{O} \left( \frac{(H + \sigma_{noise}^2) \cdot \log(|\mathcal{R}_{vb}|/\delta)}{n} \right)
$$

*Note that with independent noise at each step, $\sigma_{noise}^2 \leq O(H)$.*

*Proof.* This result is a direct combination of Lemmas C.25 and C.27. □

**Lemma C.25** (Lemma C.1 in (Foster et al., 2024b)). *It suffices to choose,*

$$
\boldsymbol{Est}_n^{Off}(\delta) = \frac{8\sigma_{noise}^2 \log(|\mathcal{R}_{vb}|/\delta)}{n}
\tag{24}
$$

*to guarantee that $\Pr(\mathcal{E}_\delta) \geq 1 - \delta$.*

**Lemma C.26.** *With the choice $\gamma = \boldsymbol{Est}_n^{Off}(\delta)$, under the event $\mathcal{E}_\delta$, $r \in \hat{R}_\gamma$. Under the same event, for every reward $r'' \in \hat{R}_\gamma$,*

$$
\frac{1}{n} \sum_{i=1}^{n} |r''(\tau_i) - r(\tau_i)| \leq 16 \cdot \boldsymbol{Est}_n^{Off}(\delta)
$$

*Proof.* The first assertion follows by definition of $\widetilde{R}_\gamma$ and Equation (22), and the fact that $r$ is a bi-level reward, so it is unperturbed by the $\text{round}(\cdot)$ operation. For the second assertion: under $\mathcal{E}_\delta$, for any reward $r' \in \widetilde{R}_\gamma$,

$$
\frac{1}{n} \sum_{i=1}^{n} (r'(\tau_i) - r(\tau_i))^2 \leq \frac{2}{n} \sum_{i=1}^{n} (r'(\tau_i) - \hat{r}_{\text{ls}}(\tau_i))^2 + (r(\tau_i) - \hat{r}_{\text{ls}}(\tau_i)))^2 \leq 4\boldsymbol{Est}_n^{\text{Off}}(\delta)
\tag{25}
$$

Consider the $r'' = \text{round}(r') \in \hat{R}_\gamma$, for this choice of reward, observe that $r''(\tau) - r(\tau) \in \mathbb{Z}$, since both rewards only take integer values. Furthermore, $(a)$ if $|r'(\tau) - r(\tau)| < 1/2$, then we know that $r''(\tau) - r(\tau) = 0$ surely, and $(b)$ if

$|r'(\tau) - r(\tau)| \geq 1/2$, then $|r''(\tau) - r(\tau)| \leq 2|r'(\tau) - r(\tau)|$. This implies,

$$\frac{1}{n}\sum_{i=1}^{n}|r''(\tau_i) - r(\tau_i)| = \frac{1}{n}\sum_{i=1}^{n}|r''(\tau_i) - r(\tau_i)| \cdot \mathbb{I}(|r'(\tau) - r(\tau)| > 1/2)$$

$$\leq \frac{2}{n}\sum_{i=1}^{n}|r'(\tau_i) - r(\tau_i)| \cdot \mathbb{I}(|r'(\tau) - r(\tau)| > 1/2)$$

$$\leq \frac{4}{n}\sum_{i=1}^{n}|r'(\tau_i) - r(\tau_i)|^2 \cdot \mathbb{I}(|r'(\tau) - r(\tau)| > 1/2)$$

$$\leq 16 \cdot \textbf{Est}_n^{\text{Off}}(\delta)$$

where the last inequality follows from Equation (25). □

### C.4.3. PROOF OF PROPOSITION 5.5

**Lemma C.27** (Generalization bound for learning in $L_1$-error). *With probability $1 - 2\delta$, simultaneously for all $r' \in \hat{R}_\gamma$,*

$$\mathbb{E}_{\rho,\pi_b}[|r(\tau) - r'(\tau)|] \leq \mathcal{O}\left(\frac{H \cdot \log(|\mathcal{R}_{vb}|/\delta)}{n} + \textbf{Est}_n^{Off}(\delta)\right)$$

*Proof.* For any fixed reward $r' \in \mathcal{R}_{vb}$, by Bernstein concentration, with probability $\geq 1 - \delta$,

$$\mathbb{E}_{\rho,\pi_b}[|r(\tau) - r'(\tau)|] - \frac{1}{n}\sum_{i=1}^{n}[|r(\tau_i) - r'(\tau_i)|] \leq \sqrt{\frac{\text{Var}_{\rho,\pi_b}[|r(\tau) - r'(\tau)|] \cdot \log(1/\delta)}{n}}$$

$$\leq \sqrt{\frac{\mathbb{E}_{\rho,\pi_b}[(r(\tau) - r'(\tau))^2] \cdot \log(1/\delta)}{n}}$$

$$\leq \sqrt{\frac{H \cdot \mathbb{E}_{\rho,\pi_b}[|r(\tau) - r'(\tau)|] \cdot \log(1/\delta)}{n}}$$

Union bounding over rewards in $\mathcal{R}_{vb}$, and choosing an arbitrary $r' \in \hat{R}_\gamma$, by Lemma C.26, with probability $\geq 1 - 2\delta$,

$$\mathbb{E}_{\rho,\pi_b}[|r(\tau) - r'(\tau)|] \leq 16 \cdot \textbf{Est}_n^{\text{Off}}(\delta) + \sqrt{\frac{H \cdot \mathbb{E}_{\rho,\pi_b}[|r(\tau) - r'(\tau)|] \cdot \log(|\mathcal{R}_{vb}|/\delta)}{n}}$$

Solving the quadratic equation results in the upper bound: with probability $\geq 1 - 2\delta$,

$$\forall r' \in \hat{R}_\gamma, \quad \mathbb{E}_{\rho,\pi_b}[|r(\tau) - r'(\tau)|] \leq \mathcal{O}\left(\frac{H \cdot \log(|\mathcal{R}_{vb}|/\delta)}{n} + \textbf{Est}_n^{\text{Off}}(\delta)\right)$$

□

### C.5. Weakening the notion of anti-concentration

The notion of anti-concentration we consider can be weakened to accommodate "average-case" notions of anti-concentration at the cost of a slightly weaker dependency of the reward gap, as a function of $n$. In particular, using Hölder's inequality we may show that,

$$\sup_{\pi \in \Pi_\kappa} J_r(\pi) - J_r(\hat{\pi}_n^{\text{vb}}) \leq \mathbb{E}_x[1/c_\mathbf{x}^p]^{1/p}\frac{H}{n^{1-1/p}}, \qquad p \geq 2. \tag{26}$$

This change is purely analytical, and does not require any modification to the algorithm. In comparison, our current result shows the bound:

$$\sup_{\pi \in \Pi_\kappa} J_r(\pi) - J_r(\hat{\pi}_n^{\text{vb}}) \leq \|1/c_\mathbf{x}\|_\infty \frac{H}{n}, \tag{27}$$

which corresponds to letting $p \to \infty$ in Equation (26). The result in Equation (26) has an improved dependency on the individual anti-concentration parameters, $c_{\mathbf{x}}$, and no longer depends on the smallest value, but this comes at the cost of worse dependency on $n$. In particular, with $p = 3$, we get a bound of the form:

$$\sup_{\pi \in \Pi_\kappa} J_r(\pi) - J_r(\hat{\pi}_n^{\text{vb}}) \leq \mathbb{E}_x[1/c_{\mathbf{x}}^3]^{1/3} \frac{H}{n^{2/3}}. \tag{28}$$

For any constant $p$ strictly greater than 2, say $2 + \varepsilon$ the verifier based bound decays as $H/n^{1/2+\varepsilon}$, which is strictly faster than the $H/\sqrt{n}$ rate exhibited by SFT. In particular, this results in the same conclusion as Theorem 5.8 in the current paper: the advantage now, is that the bound depends on a weaker notion of anti-concentration than the worst case bound considered in the paper.

## C.6. Proof of Theorem 5.8

The proof of this result follows directly from the instance lower bound in Theorem 5.4 and suboptimality upper bound result in Theorem 5.7. When, $\tilde{\sigma}_b = \Omega(H)$, the lower bound on the suboptimality gap of any VF method scales as $H \log(|\Pi|)/n$, with respect to any expert in a $O(1)$-$\chi^2$ ball around the base policy $\pi_b$, where as if $\pi_b$ is $c_0$ anti-concentrated, then there exists an algorithm that yields an upper bound on the suboptimality gap of $H \log |\mathcal{R}|/n$, with constant probability. Thus, in compliance with the definition of scaling test-time compute in Definition 4.2, as we scale $n = \Omega(H)$, we get the result in Theorem 5.1.

As an example of such a $\pi_b$, consider a single prompt, and a base policy that gets a reward of 1 with probability $> \frac{3}{5}$ on any trajectory rolled out till horizon $H = H_0$, and that this mass remains constant as we scale $H \to \infty$, i.e., the fraction of in correct trajectories (in the set $\mathcal{S}_{H_0}$) remain incorrect no matter how much we rollout $\pi_b$. For this distribution, it is easy to see that $\tilde{\sigma}_b = \Omega(H)$, but is 0.5-anti-concentrated.

## C.7. Analyzing Verifier Accuracy Under 0/1 Loss

Consider the following modified version of Algorithm 1.

---

**Algorithm 2** Simple Verifier-Based Algorithm with $\ell_{0/1}$ loss

---

**Require:** Base policy $\pi_b$, dataset $\{(\mathbf{x}_i, \tau_i)\}_{i=1}^n$ of prompts $\mathbf{x}_i \sim \rho$ and traces $\tau_i \sim \pi_b(\cdot \mid \mathbf{x})$.
1: For every $\tau_i$ annotate $(\mathbf{x}_i, \tau_i)$ with bi-level reward $r(\tau_i)$.
2: Learn set of classifiers $\hat{R}_\gamma \subset \mathcal{R}$ that are $\gamma$-optimal, *i.e.*,

$$\hat{R}_\gamma =: \left\{ r' \in \mathcal{R} \,\middle|\, \frac{1}{n} \sum_{i=1}^n \ell_{0/1}(r'(\tau_i), r(\tau_i)) \leq \gamma \right\}$$

3: Return any optimal pessimistic verifier-based policy,

$$\hat{\pi}_n^{\text{vb}} \in \arg\max_{\pi \in \Pi} \min_{r \in \hat{R}_\gamma} J_r(\hat{\pi}).$$

---

**Proposition C.28** (Verifier accuracy). *For any bi-level reward $r$, base policy $\pi_b$, there exists an algorithm querying the at most reward annotator $n$ times to learn $\hat{r} \in \mathcal{R}$, s.t. w.p. $1 - \delta$,*

$$\mathbb{E}_{\rho, \pi_b}[\ell_{0/1}(r(\tau), \hat{r}(\tau))] = \widetilde{\mathcal{O}}_n\left(\frac{\log(|\mathcal{R}|/\delta) \log H}{n}\right) =: \gamma_{stat}.$$

*In Algorithm 1, setting $\gamma = \gamma_{stat} \implies r \in \hat{R}_\gamma$ w.p. $\geq 1 - \delta$.*

**Definition C.29** (Graph dimension). Let $\mathcal{H}$ be a hypothesis class on an input space $\mathcal{X}$ and label space $\mathcal{Y}$. Let $S \subseteq \mathcal{X}$. The class $\mathcal{H}$ is said to G-shatter $S$ if there exists an $f : S \to \mathcal{Y}$ such that for every $T \subseteq S$, there is a $g \in \mathcal{H}$ such that $\forall x \in T$, $g(x) = f(x)$, and $\forall x \in S \setminus T$, $g(x) \neq f(x)$. The graph dimension of $\mathcal{H}$, denoted $d_G(\mathcal{H})$, is the maximal cardinality of a set that is G-shattered by $\mathcal{H}$.

**Theorem C.30** (Sample complexity of multiclass classification (Daniely et al., 2011)). *There exists an absolute constant*

$C > 0$ *such that for every hypothesis class* $\mathcal{H}$, *given a* $\mathcal{H}$-*realizable i.i.d. dataset* $D$ *of size* $n \geq n(\varepsilon)$, *where*,

$$n(\varepsilon) = C \left( \frac{d_G(\mathcal{H}) \log(1/\varepsilon) + \log(1/\delta)}{\varepsilon} \right), \tag{29}$$

*empirical risk minimization on* $D$ *with the hypothesis class* $\mathcal{H}$ *incurs* 0-1 *loss of at most* $\varepsilon$ *with probability at least* $1 - \delta$.

**Lemma C.31** (Upper bound on the graph dimension). *For any hypothesis class* $\mathcal{H}$, $d_G(\mathcal{H}) \leq \log_2(|\mathcal{H}|)$.

*Proof.* For a set $S \subseteq \mathcal{X}$ to be $G$-shattered by $\mathcal{H}$ if there exists a function $f$ such that for any subset $T \subseteq S$ there exists an discriminator $g_T \in \mathcal{H}$ that agrees with $f$ on $T$ and disagrees with it on $S \setminus T$. Across different choices of the subset $T \subseteq S$, the discriminating $g_T$ cannot be the same: indeed for $T_1 \neq T_2 \subseteq S$, $g_{T_1}$ and $g_{T_2}$ must disagree on points in $(T_1 \setminus T_2) \cup (T_2 \setminus T_1)$, the symmetric difference of the two subsets. This is simply because on points in $T_1 \setminus T_2$, $g_{T_1}$ agrees with $f$ and $g_{T_2}$ disagrees with $f$, while on points in $T_2 \setminus T_1$, $g_{T_2}$ agrees with $f$ and $g_{T_1}$ disagrees with $f$. Since the map $T \to g_T$ is injective, and there are $2^{|S|}$ choices of $T$, this means that $S$ can only be $G$-shattered if $|\mathcal{H}| \geq 2^{|S|}$. $\square$

**Theorem C.32.** *Given a dataset of* $n(\varepsilon)$ *trajectories from* $\pi_b$, *there exists an algorithm which calls the verifier* $n(\varepsilon) \lceil \log_2(H) \rceil$ *times and learns a reward model such that*,

$$\mathbb{E}_{\rho, \pi_b} \left[ \mathbb{I}(r(\tau) \neq \hat{r}(\tau)) \right] \leq \varepsilon. \tag{30}$$

*Proof.* Recall that $\mathcal{R}$ is assumed to be a bi-level reward class. For each $r \in \mathcal{R}$, consider the multiclass classifier $f_r : (\mathcal{S} \times \mathcal{A})^H \to [H + 1]$ which maps a trajectory $\tau = \{(s_1, a_1), \cdots, (s_H, a_H)\}$ to the value of $h \in [H]$ such that $h$ is the first point in the trajectory where $r(s_h, a_h) = 1$, i.e., the location of the bi-level in the trajectory. If the reward stays $0$ entirely through the trajectory, then $f_r(\tau) = H + 1$. First, we relate the 0-1 error of a reward estimator $\hat{r}$ to the multiclass classification error of $f_r$, assuming the labels come from $f_r$. Observe that,

$$\mathbb{E}_{\rho, \pi_b} \left[ \mathbb{I}(r(\tau) \neq \hat{r}(\tau)) \right] \leq \mathbb{E}_{\rho, \pi_b} \left[ \mathbb{I}(f_r(\tau) \neq f_{\hat{r}}(\tau)) \right]. \tag{31}$$

This follows from the fact that, if $r(\tau) \neq \hat{r}(\tau)$, then the bi-level in this trajectory $\tau$ is identified incorrectly, implying that $f_r(\tau) \neq f_{\hat{r}}(\tau)$. Recall that the expert dataset is composed of $n = n(\varepsilon)$ trajectories $D = \{(\mathbf{x}_i, \tau_i)\}_{i=1}^n$ for some $\varepsilon > 0$ (see Equation (29) for the definition of $n(\varepsilon)$). Using the verifier to annotate rewards, by a binary searching, the location of the bi-level in any of these $n$ trajectories may be located: thus with $n \lceil \log_2(H) \rceil$ calls to the verifier, a dataset of $n$ examples may be constructed of the form $\{(\tau_i, f_r(\tau))\}_{i=1}^n$ for the ground truth reward $r$. By carrying out empirical risk minimization over the hypothesis class $\mathcal{F} = \{f_r : r \in \mathcal{R}\}$ to learn a hypothesis $\hat{f}$, and invoking Theorem C.30, with probability $\geq 1 - \delta$,

$$\mathbb{E}_{\rho, \pi_b} \left[ \mathbb{I}(f_r(\tau) \neq \hat{f}(\tau)) \right] \leq \varepsilon. \tag{32}$$

$\square$

### C.8. Proof of Theorem 5.4 for the single problem instance

This result follows using a similar approach as the instance-dependent lower bound against behavior cloning proved in (Foster et al., 2024a). For the case, where we have a single prompt $\mathbf{x}$, we use the following lemma to argue that given an expert policy $\pi_e$, we can always construct another policy $\tilde{\pi}_e$, and a pair of rewards $\{r, \tilde{r}\}$ that satisfy certain properties, while ensuring that each policy observes a variance of $\sigma^2$ in the range $(0, H^2/4]$ for either of the rewards.

Next, we consider the following inequality, which holds for any $\Delta > 0$:

$$\min_{\text{Alg}} \max_{\pi \in \{\pi_e, \tilde{\pi}_e\}} \max_{r \in \{r, \tilde{r}\}} \mathbb{P}\left[J_r(\pi) - J_r(\hat{\pi}) \geq \Delta\right] \geq \min_{\text{Alg}} \max_{\pi \in \{\pi_e, \tilde{\pi}_e\}} \mathbb{P}\left[|J_r(\pi) - J_r(\hat{\pi})| \geq \Delta\right].$$

Here, $J_r(\pi)$ denotes the expected reward under the reward function $r$, and for convenience, we abbreviate $J(\pi) \equiv J_r(\pi)$ going forward. Let $\mathbb{P}_n^\pi$ represent the probability distribution of the offline imitation learning dataset when the data is collected under policy $\pi$. By choosing $\Delta = \frac{|J(\pi_e) - J(\tilde{\pi}_e)|}{2}$, and applying the standard Le Cam two-point argument, we can conclude that:

$$\max \left\{ \mathbb{P}_n^{\pi_e} \left[ |J(\pi_e) - J(\hat{\pi})| \geq \Delta \right], \mathbb{P}_n^{\tilde{\pi}_e} \left[ |J(\tilde{\pi}_e) - J(\hat{\pi})| \geq \Delta \right] \right\}$$

is bounded below by:

$$\frac{1}{2} \left( 1 - \mathbb{P}_n^{\pi_e} \left[ |J(\pi_e) - J(\hat{\pi})| < \Delta \right] + \mathbb{P}_n^{\tilde{\pi}_e} \left[ |J(\tilde{\pi}_e) - J(\hat{\pi})| \geq \Delta \right] \right).$$

This, in turn, is further bounded below by:

$$\frac{1}{2} \left( 1 - \mathbb{P}_n^{\pi_e} \left[ |J(\tilde{\pi}_e) - J(\hat{\pi})| \geq \Delta \right] + \mathbb{P}_n^{\tilde{\pi}_e} \left[ |J(\tilde{\pi}_e) - J(\hat{\pi})| \geq \Delta \right] \right),$$

and by a standard application of the data processing inequality for the total variation distance, we have:

$$\frac{1}{2} \left( 1 - D_{\mathrm{TV}} \left( \mathbb{P}_n^{\pi_e}, \mathbb{P}_n^{\tilde{\pi}_e} \right) \right).$$

Utilizing the tensorization property of the Hellinger distance (Wainwright, 2019), we further lower bound this by:

$$\frac{1}{2} \left( 1 - \sqrt{n \cdot D_{\mathrm{H}} \left( \mathbb{P}^{\pi_e}, \mathbb{P}^{\tilde{\pi}_e} \right)} \right).$$

Next, we proceed to show the following key inequality:

$$\omega_{\pi_e}(\varepsilon) := \sup_{\pi} \left\{ |J(\pi) - J(\pi_e)| \,\big|\, D_{\mathrm{H}} \left( \mathbb{P}^{\pi_e}, \mathbb{P}^{\pi} \right) \leq \varepsilon^2 \right\} \geq \Omega(1) \cdot \sqrt{\sigma_{\pi_e}^2 \cdot \varepsilon^2},$$

for any $\varepsilon > 0$ sufficiently small. The final result follows by setting $\varepsilon^2 \propto \frac{1}{n}$, and defining:

$$\tilde{\pi}_e = \arg\max_{\pi} \left\{ |J(\pi) - J(\pi_e)| \,\big|\, D_{\mathrm{H}} \left( \mathbb{P}^{\pi_e}, \mathbb{P}^{\pi} \right) \leq \varepsilon^2 \right\}.$$

To prove this, we invoke the following technical lemma:

**Lemma C.33** (Lemma G.1 in Foster et al. (2024a)). *For any distribution $\mathbb{Q}$ and any function $h$ satisfying $|h| \leq R$ almost surely, it holds that for all $0 \leq \varepsilon^2 \leq \frac{\mathrm{Var}_{\mathbb{Q}}[h]}{4R^2}$, there exists a distribution $\mathbb{P}$ such that:*

*1. $\mathbb{E}_{\mathbb{P}}[h] - \mathbb{E}_{\mathbb{Q}}[h] \geq 2^{-3} \sqrt{\mathrm{Var}_{\mathbb{Q}}[h] \cdot \varepsilon^2}$,*

*2. $D_{KL}(\mathbb{Q}\|\mathbb{P}) \leq \varepsilon^2$.*

In the case of stochastic policies $\pi$ in the autoregressive Markov Decision Process $\mathcal{M}^*$, these policies are equivalent to defining arbitrary joint distributions over the sequence $(a_1, \ldots, a_H)$ using Bayes' rule. Consequently, since $J(\pi) = \mathbb{E}^{\pi} \left[ \sum_{h=1}^{H} r_h \right]$, Lemma C.33 ensures that for any $\varepsilon^2 \leq \frac{\mathrm{Var}^{\pi_e} \left[ \sum_{h=1}^{H} r_h \right]}{4R^2}$, there exists a policy $\tilde{\pi}_e$ such that:

$$D_{\mathrm{H}} \left( \mathbb{P}^{\pi_e}, \mathbb{P}^{\tilde{\pi}_e} \right) \leq D_{\mathrm{KL}} \left( \mathbb{P}^{\pi_e}, \mathbb{P}^{\tilde{\pi}_e} \right) \leq \varepsilon^2,$$

and:

$$J(\tilde{\pi}_e) - J(\pi_e) \geq 2^{-3} \sqrt{\mathrm{Var}^{\pi_e} \left[ \sum_{h=1}^{H} r_h \right] \cdot \varepsilon^2}.$$

This establishes the desired inequality. Setting $\varepsilon^2 = \frac{c}{n}$ for some constant $c > 0$, we achieve

$$\sqrt{n \cdot D_H\left(\mathbb{P}^{\pi_e}, \mathbb{P}^{\tilde{\pi}_e}\right)} \leq \frac{1}{2},$$

which is valid provided that $n \geq c' \cdot \frac{R^2}{\sigma^2_{\pi_e}}$.

## D. Additional Experiments in the Didactic Setup

**Details on the setup.** We generalize the planted subsequence problem from (Setlur et al., 2024b). The input prompt is a sequence of length 5 with the tokens chosen randomly from the set $\{1, 2, 3, \ldots, 10\}$. We fix the unknown function to be $g(x) = 2x + 5$. We fix the vocabulary for the policy we are training to be the set $\mathcal{V} =: \{0, \ldots, 30\}$. Here 0 is treated as the padding token. Concretely, for an input problem $\mathbf{x} = (x_1, .., x_5)$, we say that a response $\mathbf{y}$ with $H$ tokens from the vocabulary $\mathcal{V}$ is a correct trace if there exists a *gold* contiguous subsequence $(g(x_1), .., g(x_5))$ planted in $\mathbf{y}$. Here, the underlying mapping $g : [10] \mapsto [30]$ is fixed but unknown. For a state $\mathbf{s} =: (\mathbf{x}, a_1, \ldots, a_h)$, the bi-level reward $r(\mathbf{s}) = 1$ if and only if there exists some $h' \leq h$ such that the last 5 tokens before $h'$ *i.e.*, $(a_{h'-4}, .., a_{h'})$ match the gold subsequence. In order to use the same performance scale to compare methods trained for different horizon $H$ values (test-time compute budget), we $J_r(\pi)$ and divide it by the maximum reward of $H - 4$.

We wish to construct base policies $\pi_b$ that: **(i)** differ in heterogeneity, and **(ii)** satisfy the anti-concentration condition. To do so, we finetune GPT2-xl (Radford et al., 2019) on samples obtained from a mixture of hand-designed "procedural" policies. Inspired from Setlur et al. (2024b), a procedural policy $\mu_\gamma(\mathbf{y}^\star_{k+1}|\mathbf{s}) \propto \gamma$, when the last $k$ tokens in the state $\mathbf{s}$, match the first $k$ tokens in the gold subsequence $\mathbf{y}^\star$. Thus, the normalized return for $\mu_\gamma \to 1$, as $\gamma \to \infty$. We vary the heterogeneity of $\pi_b$ by finetuning GPT2-xl on data from a mixture of procedural policies with $\gamma \in \{5, 10, 20, 50, 100, 500\}$. Once the last 5 tokens match the gold sequence, the procedural policy puts mass $\propto \gamma$ on the padding token 0. See Figure 9 for an illustration of data sampled from different procedural policies.

For any compute budget $H$ (token length), we train separate SFT and RL policies, where SFT is run on traces that are $H$ tokens long. We also run RL on the same token budget, against a trained verifier. The verifier is trained on samples from the base policy. For this, we train a GPT2-xl transformer as a multiclass classifier, that takes in an $H$ length sequence and outputs a single value in 0 to $H$ (i.e., it is an $H + 1$-way classifier).

**Experiment details.** For the RL runs, we use REINFORCE (Ahmadian et al., 2024) train for 20k iterations in both with a batch size of 64, and a constant learning rate of $1e - 4$, with the Adam optimizer. The RL runs are initialized with the base policy, and to prevent reward hacking we also use a KL penalty (with weight 0.2), in addition to the REINFORCE training objective. For every trace in a batch, we query the trained verifier, which outputs a value between 0 and $H$, which directly tells us where the "staircase" appears in the bi-level reward. For example, a value of 2 implies that the staircase appears on the second last token. We convert this outcome supervision into token-level 0/1 rewards and update the policy with the computed policy gradient. For SFT, we also use the Adam optimizer with a learning rate of $2e - 4$, and a batch size of 64. Similar to RL, we apply a KL regularization term in addition to the next token prediction loss (ignoring the padding token 0), where the strength of the KL term is the same as RL. SFT runs are also initialized with the base policy. Using the same hyperparameters, we obtain the base policy by running SFT on 200k data points sampled i.i.d. from the uniform mixture over procedural policies outlined above. To collect training data for the verifier, we draw a random sample of $n/\log H$ prompts in $\mathcal{D}_{tr}$, and then make $\log(H)$ calls on each of them to binary search for the token where the correct answer first appeared. This way, we only query reward annotator $n$ times. Finally, for our experiments, where we vary base and expert policy heterogeneity, we simply change $\gamma$ (reducing variance over it), in a way that the average performance of the base/expert policy remains roughtly the same.

**Accuracy of trained verifier.** In Figure 10(left), we plot the accuracy of the verifier (black line), as we scale the horizon. We fix the data budget to $n = 2^14$ here. Since, here budget implies a multi-class classification over more classes, the problem hardness increases for the verifier, which explains the performance drop. Initially, we do see an improvement with $H$, since the coverage over high reward trajectories improves with $H$, as we sample the base policy for longer. We also plot the upper bound on RL performance, where we train the RL policy with ground-truth staircase rewards. Looking at its performance, it is clear that across all horizons, RL with trained verifier mainly suffers from the inaccuracy of the trained verifier (i.e., reward hacking issues). In Figure 10(right), we plot the accuracy of the learned verifier on two distributions (base policy), and the policy learned by RL. As we reduce base policy heterogeneity, it is easier to generalize on the base policy, but the verifier is inaccurate outside the narrow distribution of the base policy, making it more susceptible to reward hacking. As a

| Input
(Context) | Procedural policy
$\gamma = 10$ | Procedural policy
$\gamma = 1000$ |
|---|---|---|
| 1, 2, 5, 3, 8 | 7, 9, 3, 7, 25, 7, 9, 15, 14, 20
Reward: 0, Normalized: 0 | 7, 9, 15, 11, 21, 0, 0, 0, 0, 0
Reward: 6, Normalized: 1 |
| 3, 1, 7, 2, 6 | 11, 7, 3, 11, 7, 19, 9, 17, 2, 5
Reward: 3, Normalized: 0.5 | 11, 7, 19, 9, 17, 0, 0, 0, 0, 0
Reward: 6, Normalized: 1 |

Unknown mapping: $g(x) = 2x + 5$  ☐ Gold subsequence

Figure 9: ***Procedural policies for the generalized planted subsequence problem:*** For two values of $\gamma$: 10, and 1000, we show examples of two draws, over $H = 10$ tokens from each. Here, the unknown mapping is $g(x) = 2x + 5$. When $\gamma$ is 1000, the policy (over the first 5 tokens) is almost like a dirac delta distribution on the gold subsequence, followed by which it samples the padding tokens. On the other hand, when $\gamma = 10$, it makes multiple attempts and completing the sequence. Once it fails, it makes a new attempt. In the second sample, we see that after a few tokens it gets the correct sequence, achieving a total bi-level reward of 3, and normalizing it with $H - 4$, we get a normalized reward of 0.5.

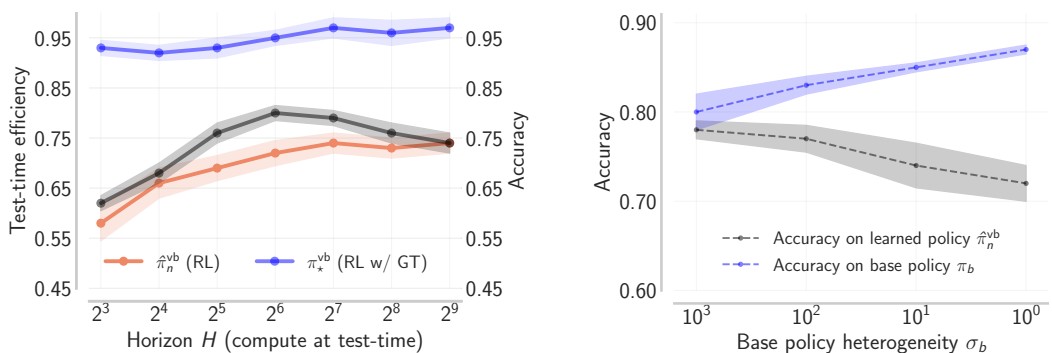

Figure 10: ***Accuracy of trained verifier:*** (Left) we plot the accuracy of the verifier (black line), as we scale the horizon (black line). We also plot the performance of RL with ground-truth (GT) bi-level rewards, and compare it with RL using the trained verifier. (Right) As we vary base policy heterogeneity we plot the accuracy of the verifier on two distributions: (i) on base policy $\pi_b$, (ii) on policy learned by running RL $\hat{\pi}_n^{\text{vb}}$.

result, we observe poor accuracy on the learned policy's distribution.

## E. Additional Experiments on MATH

**Experiment details.** We run all our training on the questions in the training set of MATH (Hendrycks et al., 2021), and run our test on the MATH500 evaluation benchmark. Any problem instance is determined by the data budget $n$ and compute budget $H$. For a fixed compute budget $H$, we run SFT on trajectories of search traces. A single search trace consists of multiple turns of responses. Each response is at most 512 tokens long. A trace can thus comprise of anywhere between 1 to $H/512$ turns (sometimes more for shorter responses). Here, all turns except the last one are responses that result in incorrect final answers for the problem. This is exactly the recipe followed by Qu et al. (2024) and Snell et al. (2024), for learning to self-correct, revise responses, or more generally use sequential compute at test-time. See below for examples of search traces of SFT. For training the verifier, we simply use samples from the base LLM, and use 0/1 rewards to train a model of the same size as the policy with cross-entropy loss. We use a batchsize of 32 and learning rate of $1e - 6$ for all our experiments. We run SFT and verifier training for 10000 iterations on each instance. We use a weight decay of 0.01 for training both. At test-time, we measure the performance of SFT using majority voting over sequential revisions, following Snell et al. (2024). We sample rollouts with temperature 1.0.

**Verifier accuracy.** In Figure 11 we plot the test accuracy of the trained verifier as we scale data. Interestingly, we find that the bigger model always generalizes better, and models of both sizes improve performance at the same rate, as we scale the data budget $n$.

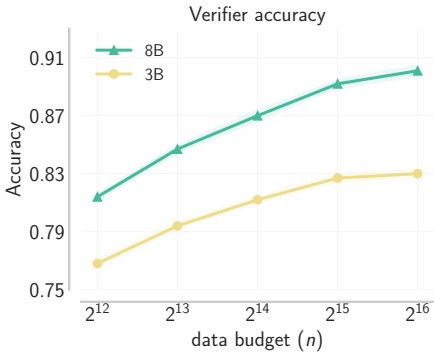

Figure 11: *Accuracy of verifier:* We plot the accuracy of verifier as we scale data, for Llama3-3b and 8b sized models. We find that the larger model generalizes better across all data budgets.

---

**Search trace with sequential revisions for SFT (example with 3 episodes)**

**Query:** A rectangle's length is five more than twice its width. If its perimeter is 58 cm, what is its width?

**Turn-1 Response:** Step 1: Let's assume the width of the rectangle as $x$. Step 2: According to the given information, the length is five more than twice the width. So, the length can be expressed as $2x + 5$. Step 3: The perimeter of a rectangle is given by the formula:

$$\text{Perimeter} = 2(\text{Length} + \text{Width})$$

Step 4: Substituting the given perimeter of 58 cm and our expressions for length and width:

$$2(x + 2x + 5) = 58$$

$$x = \frac{53}{6}$$

Answer: $\boxed{\frac{53}{6}}$ INCORRECT

**Turn-2 Response (Subtle Error Introduced):** Step 1: The width is $x$ and the length is $2x + 5$. Step 2: Using the perimeter formula:

$$2(x + 2x + 5) = 58$$

$$6x + 5 = 58$$

$$x = 7$$

Answer: $\boxed{7}$ INCORRECT

**Turn-3 Response:** Let's solve the problem step by step: Step 1: Let's denote the width of the rectangle as $x$. Step 2: The length is five more than twice the width, so it can be expressed as $2x + 5$. Step 3: The perimeter formula for a rectangle is:

$$\text{Perimeter} = 2(\text{Length} + \text{Width})$$

Step 4: Substituting the given perimeter of 58 cm:

$$2(x + 2x + 5) = 58$$

$$x = 8$$

Answer: $\boxed{8}$ INCORRECT

---

---

**Search trace with sequential revisions for SFT (example with 2 episodes)**

**Query:** A farmer has three times as many chickens as cows on his farm. If he has 15 cows, how many chickens does he have?

**Turn-1 Response:** Step 1: Let's assume the number of cows as $x$. Step 2: According to the given information, the farmer has three times as many chickens as cows. So, the number of chickens can be expressed as $3x$. Step 3: We are given that there are 15 cows, so let's substitute $x = 15$ into the equation:

$$3x = 3(15) = 30$$

Answer: $\boxed{30}$ INCORRECT

**Turn-3 Response:** Let's solve the problem step by step: Step 1: Let's denote the number of cows as $x$. Step 2: We are given that the farmer has three times as many chickens as cows. So, the number of chickens can be expressed as $3x$. Step 3: We are also given that there are 15 cows, so we can substitute $x = 15$ into the expression for the number of chickens:

$$3 \times 15 = 45$$

Answer: $\boxed{45}$ CORRECT

---

