# OpenReview forum: "Scaling Test-Time Compute Without Verification or RL is Suboptimal"
_ICML.cc/2025/Conference — ICML 2025 spotlightposter_

### Official Review · Reviewer_qsM1 · 2025-03-08

**Overall Recommendation:** 4

**Summary:**

The paper theoretically studies inference time scaling by modeling LLM generation as a Markov process with horizon $H$ and comparing verifier-based (VB) methods to verifier-free (VF) methods. For binary and nondecreasing rewards, and under a heterogeneity assumption of the base model search trace solutions, it is shown via an information theoretic construction that any VF algorithm with access to $n$ data points from an expert has suboptimality gap $\Omega(H/\sqrt{n})$, in the worst-case sense over a class of alternate experts/rewards. In contrast, it is shown that a specific VB algorithm with access to $n$ annotated search traces, optimizing a pessimistic reward over a class of approximate reward functions, achieves suboptimality gap $\widetilde{O}(H/n)$ under an anti-concentration assumption on the distribution of rewards. Hence VB can asymptotically outperform VF algorithms as test time compute $H$ is scaled. The assumptions and results are also verified experimentally on synthetic and math reasoning tasks.

* Update after rebuttal: I am satisfied with the authors' response and maintain my high rating of the contribution.

**Claims And Evidence:**

The claims are supported by convincing theoretical and empirical evidence.
* It is shown that the suboptimality of any VF estimator scales as $\tilde{\sigma}_b/\sqrt{n}$ where $\tilde{\sigma}_b$ measures the median (over prompts) expected variation of the Q-function under the base LLM policy. This results in a $H/\sqrt{n}$ separation assuming $\tilde{\sigma}_b=\Theta(H)$.
* Moreover, the paper constructs a VB estimator whose suboptimality scales as $H/n$ under an anti-concentration assumption for the base policy reward distribution. (This is required to hold uniformly over prompts which is slightly stronger, but I think this is fine.)
* Combining these establishes a $\sqrt{H}$ separation w.r.t. the generation horizon when $n=\Theta(H)$.
* Moreover, experiments on the MATH benchmark confirm the assumptions and show that heterogeneity hurts VF methods as predicted by the theory.

These results provide a solid framework to theoretically understanding the effect of verifier model vs expert cloning for inference time scaling.

**Essential References Not Discussed:**

None in particular

**Experimental Designs Or Analyses:**

There do not seem to be any glaring issues with experiment design

**Methods And Evaluation Criteria:**

Yes, see Claims And Evidence

**Other Comments Or Suggestions:**

While the many intuitive explanations are appreciated, the paper could benefit from being a bit more precise in the statements (e.g. theorems) of the main text for the theoretically inclined reader. In particular, it would be helpful to mathematically describe exactly what samples the VB and VF models have access to (expert trace vs annotated trace).

Typos: $r\to r'$ in Theorem 5.4, verifier-based $\to$ verifier-free in Theorem B.10(4)

**Other Strengths And Weaknesses:**

See above

**Questions For Authors:**

Some bounds in the proofs seem to rely on the rewards being binary, such as the rounding argument; can the results be generalized to when rewards are real-valued and monotonic?

**Relation To Broader Scientific Literature:**

Test time scaling for LLM reasoning is obviously a topic of crucial importance, however the theoretical understanding of chain of thought and test time scaling is still very shallow. I think the paper is a strong contribution towards this direction, proposing an MDP framework to study generation and interpreting verifier and behavior cloning from a novel information theoretic point of view.
From a theoretical perspective, the paper borrows from but expands on existing regret analyses of Markov decision processes (e.g. Foster et al, 2024).

**Theoretical Claims:**

The proofs are correct to the best of my knowledge

---

> ### Author Rebuttal · Authors · 2025-04-01
>
> Thank you for the review and a positive assessment of our work! To address your remaining concerns, we clarify the data annotations for VB and VF methods and show that our analysis can be extended to settings beyond binary rewards, while proving a similar separation between VB and VF methods. We also point to discussion on a relaxation of uniform anti-concentration.
>
> >> **Extending our analysis to non-binary rewards**
>
> While our algorithms and analysis are tailored for $0$-$1$ rewards with the bilevel structure, in the non-stochastic setting (no noise in observed rewards) we can extend these results to the case of general rewards functions, while essentially preserving the final guarantees.
>
> In general, given access to the trajectory-level reward, we can quantize the rewards to $n$ levels, and apply the same classification-based algorithm for learning the reward function. By doing so, we obtain an algorithm for learning general rewards in $L_1$ error to a test error of $H \log(|\mathcal{R}|) \log(n)/n$. The only differences compared to the bilevel case are: there is an additional $\log(n)$ blow up compared to the bilevel case, the learner can directly query the outcome reward for a trajectory directly (rather than the per-step reward).
>
> The intuition is as follows: suppose the outcome reward belongs to some class $\mathcal{R}$, we can consider an induced quantized reward class $\mathcal{R}_q$, which rounds the rewards to a grid where consecutive points are $H/n$ apart. Note that $|\mathcal{R}_q| \le |\mathcal{R}|$. By quantizing the reward observations, and training a multiclass classifier over the outcome space $\mathcal{Y} = \\{ 0,\frac{H}{n}, \frac{2H}{n},\cdots,H\\}$ of size $n+1$, we can train a classifier incurring $0$/$1$ error of at most $d_G (\mathcal{H}) \log(|\mathcal{Y}|)/n \lesssim \log(|\mathcal{R}|) \log(n)/n$, where $d_G$ denotes the graph dimension (see definition B.29 in the submission).
> When the classifier is correct (in a $0$/$1$ sense), the induced reward function makes an error of at most $H/n$. If the classifier is incorrect, the induced reward function makes an error of at most $H$. Overall, this implies that the $L_1$ error of the reward function induced by the classifier (which is essentially a discretized reward $\in \\{ 0, \frac{H}{n},\cdots,H \\}$) is at most $ H\log(|\mathcal{R}|) \log(n)/n + H/n \lesssim H\log(|\mathcal{R}|) \log(n)/n$. Hence, the same lemma, and the same guarantees, will also apply in our setting. We will add this discussion in the paper.
>
> >> **Data annotations for VB and VF methods**
>
> For VF methods, we simply have $n$ datapoints of the form $\(x, \tau)$, consisting of an input $x \sim \rho$, and a trace $\tau \sim \pi_e(\cdot \mid x)$ sampled from expert $\pi_e$. For VB methods, we have $n$ datapoints of the form $(x, \tau, r(\tau))$, where $x \sim \rho$, but the trace $\tau \sim \pi_b(\cdot \mid x)$ is sampled from the base LLM $\pi_b$, and $r(\tau)$ is the reward annotation $\in [0, H]$ under the bi-level reward $r$. We will also move details from the formal theorem statements in Appendix (e.g., the formal lower bound result in Thm. B.10) into the main paper, in our final version, which we hope will make the theorem presentation more clear.
>
> >> **Relaxing uniform anti-concentration to an average case notion.**
>
> Please see our response to reviewer eEQB where we show that our uniform anti-concentration can be relaxed to $L_p$ concentration, and when $p>2$, we recover the same asymptotic separation of $\Omega(H/\sqrt{n})$ between VB and VF methods, as we obtain under uniform anti-concentration in Thm. 5.8. We show the full derivation in a figure here: **https://sites.google.com/view/ttcwv/home**.

---

### Official Review · Reviewer_eEQB · 2025-03-13

**Overall Recommendation:** 3

**Summary:**

This paper studies the performance of two prevalent methods which are named as verifier-based (VB) and verifier-free (VF) methods in terms of scaling test-time compute. VB methods utilize a verifier or reward signals to improve a policy while VF approaches use expert data to supervise the policy training. They compare the performance of these two category of methods at different test-time compute by introducing a bi-level reward to represent the compute. The authors show VB methods scale test-time compute better than VF methods if the base policy satisfy heterogeneity and anti-concentration. Besides, they implement experiments to support this finding.

**Claims And Evidence:**

No, the claim made is not supported clearly or solidly.

The main claim in this paper is that finetuning LLMs with VB methods is superior to VF approaches in terms of scaling the compute resources.

First, the title of the paper is scaling test-time compute, but this paper seems talking about the fine-tuning phase. I also raised my question in the question part below.

Second, to show the advantage of VB methods, the base policy should possess some properties which are restrictive (I elaborate in the weaknesses part). And I don't think their experiments successfully show that in practice, pre-trained policies satisfy these conditions.

**Essential References Not Discussed:**

No, I didn't see any essential related works are not discussed.

**Experimental Designs Or Analyses:**

Yes, I checked the soundness of the experimental part about didactic setup. The authors compare results at different data budget, horizon (which represents compute in their framework) and policy heterogeneity. However, policy anti-concentration is less discussed. They show and control the policy heterogeneity of their choosing base policy, but they didn't show how they control the anti-concentration which is important for the soundness of the experiments. In the experiments for math reasoning, they include a Figure 7 to show that practical pre-training policies satisfy the anti-concentration property, but this conclusion is too optimistic. At least more different policies should be tested. And I also doubt the definition of anti-concentration (please refer to the weaknesses part).

**Methods And Evaluation Criteria:**

Yes, so the authors want to show verifier-based algorithms have advantages at scaling the test-time compute compared with verifier-free methods. They first model the scaling of test-time compute by the horizon and defining a bi-level reward. Then they show the lower bound of sub-optimality of VF algorithms and upper bound of sub-optimality of VB algorithms under certain condition of the base policy.

**Other Comments Or Suggestions:**

1. Some typos: In line 183, For e.g. -> For example. On the right column in line 62, distinction -> direction. In line 305, this results in Theorem 5.7 which is an incomplete sentence.
2. It would be better to include a lemma that can formally describe the relationship between the definition of policy heterogeneity and anti-concentration. I noticed that the authors explain in somewhere in a paragraph that 'while a non-heterogeneous base policy will not satisfy Property 5.6, hetergeoneous distributions can still be anti-concentrated'. This is crucial to derive the final conclusion in this paper which is VB algorithms can perform better than VF algorithms for some base policy. It is not very straightforward at the first glance, hence I think adding a lemma would be beneficial.

**Other Strengths And Weaknesses:**

Strengths:
1. Comparing the methods of learning from rewards and learning from expert is a very interesting and important direction.

Weaknesses:
1. The authors show by Theorem 5.1, there exists a base policy $\pi_b$ such that verifier-based algorithm scales better with test-time compute. But they didn't show if it is possible that there also exists another $\pi_b$ such that verifier-free algorithm scales better. It is important to derive the final conclusion.
2. The anti-concentration parameter which they define as $c$ in Property 5.6 depends on 'worst' problem since they define $\min_x c_x(\epsilon_x)\ge c$. This a very restrictive definition/requirement. It is usually the case where we have a huge number of different problems for a specific task. The anti-concentration definition needs the base policy to have nice shape of the answer distribution over all problems. I think the authors should relax this definition. Besides, how is Figure 7 plotted, is it for just one problem? If it is for a set of problems, how are these metrics calculated?
3. The authors should include more detailed discussions on the related work [1] since this work also models the problem as a MDP and their work is very close to this work. In the paragraph before Section 6, the authors give some discussions but are vague to a reader.


[1]. Kumar A, Hong J, Singh A, et al. When should we prefer offline reinforcement learning over behavioral cloning?[J]. arXiv preprint arXiv:2204.05618, 2022.

**Questions For Authors:**

1. I am confused at what the authors refer as test-time compute. From my understanding, the test-time compute usually refers to the compute applied to the inference phase of the model which has been trained or fine-tuned. But in this paper, it seems to refer to the compute applied to the fine-tuning phase. Do I misunderstand something here?
2. In line 182, the condition (b) of the expert is the expert's distribution should be 'close' to the base policy to prevent some issues, but these issues arise in practice. From the theoretical perspective (which is this paper's perspective), do we need this condition and why?
3. In Proposition 5.5, doesn't one need realization assumption of the reward function class, i.e., the true underlying reward $R^*$ is in the set of reward functions $R^*\in \mathcal{R}$?
4. What does the y-axis (efficiency) in Figure 4 refer to, accuracy?

**Relation To Broader Scientific Literature:**

The contributions of this paper are related to two broad directions. The first is how scaling test-time compute can improve a model. It is an active area and there are some promising results [1][2]. The second is which is better: learning from experts or learning from rewards. Some works [3] study this question in reinforcement learning, especially imitation learning, but it is relatively under-explored in language models.

[1]. Snell C, Lee J, Xu K, et al. Scaling llm test-time compute optimally can be more effective than scaling model parameters[J]. arXiv preprint arXiv:2408.03314, 2024.
[2]. Hübotter J, Bongni S, Hakimi I, et al. Efficiently learning at test-time: Active fine-tuning of llms[J]. arXiv preprint arXiv:2410.08020, 2024.
[3]. Kumar A, Hong J, Singh A, et al. When should we prefer offline reinforcement learning over behavioral cloning?[J]. arXiv preprint arXiv:2204.05618, 2022.

**Theoretical Claims:**

No, I didn't check the correctness of the theorems. But I think that Proposition 5.5 needs a realization assumption of the reward function class.

---

> ### Author Rebuttal · Authors · 2025-04-01
>
> Thank you for the feedback! We believe that many concerns perhaps stem from a misunderstanding of the setup. We study the efficacy of different fine-tuning algorithms to train LLMs to attain better performance as they utilize more test-time compute, measured in terms of the token length. Verifier-based (VB), or verifier free (VF) are finetuning methods, like DeepSeek-R1 or SCoRe, training LLMs to use more test-time compute to solve more problems. We expand on this, provide more evidence of anti-concentration holding in practice, and explain how we can relax the worst-case (over prompts) anti-concentration to average case while arriving at the same result in Thm 5.8.
>
> >> **I am confused at what the authors refer to as test-time compute? It is applied to finetuning**
>
> As we explain in L131-152, our goal is to compare LLMs obtained by running different finetuning algorithms that train LLMs to utilize test-time compute (measured by the length of CoT H) more effectively. So, either we finetune LLMs with VF, e.g. by running SFT on expert traces of length at most H containing behaviors like planning, search, backtracking etc., or train LLMs with a verifier based approach like RL that samples traces of length H from the base LLM, and optimizes it with 0/1 rewards (L165-175, L200-209). Our analysis is novel because it compares finetuning algorithms that train LLMs to optimize test compute efficiency, as opposed to only analyzing the efficiency of known test-time search algorithms like best-of-N.
>
> >> **How is Fig7 plotted,is it for 1 problem. Ablations on  anticonc?**
>
> See figure & caption in **https://sites.google.com/view/ttcwv/home**.
>
> >> **Prop. 5.6 is restrictive since it needs to hold on all problems. Consider relaxing it.**
>
> The notion of anti-concentration can indeed be weakened to accommodate  *average-case* at the cost of a slightly weaker dependence on $n$.
>
> See fig in **https://sites.google.com/view/ttcwv/home** for the derivation. Note that this results in the same conclusion as Thm 5.8 in the current paper, but the bound depends on a weaker notion of anti-concentration.
>
> Additionally:
> - *Common in literature*: This style of uniform lower bound is common in RL theory (e.g., Uehara el al., Kumar et al.). Prior work assumes strong density ratio bounds, whereas we only require that base policy generates traces exceeding one std. dev. of the mean reward with constant probability.
> - *Poor anti-concentration can hurt VF too*: In practice, VF gets expert traces by running rejection sampling. However, rejecting more traces results in smaller SFT data. This tradeoff is worse for problems with poor anti-concentration. To achieve a high reward dataset, the learner would need to discard most traces, biasing the dataset and degrading SFT performance. We will add this to final version.
>
> >> **Thm 5.1: does there exist a base policy where VF > VB?**
>
> No matter the base LLM, whenever the expert is non-heterogenous, and covers high reward traces, we expect VF algos like SFT to learn good policies. It is possible that in some cases when heterogeneity and anti-concentration are not satisfied, VF methods may outperform VB, as we also show in Fig 6 (left). In practice, however, we see the assumptions are satisfied(Figs 6, 7).
>
> >> **L281, lemma implying anti-concentration needed for VB > VF**
>
> Heterogeneity (Prp. 5.2) & anti-concentration (Prp. 5.6) are largely orthogonal conditions, which we show are sufficient to prove VB > VF. We show in Sec 7 (Fig 6, 7) that *both* conditions hold in practice. As seen in Fig 3, base LLM can satisfy these conditions separately. In L281, we are referring to the fact that a heterogeneous base LLM need not satisfy anti-concentration (3rd col in Fig 3), which is why we need both Prp. 5.6 & 5.2 to hold for the base LLM.
>
> >> **Discussion on Kumar et al**
>
> In Remark 5.9 (L299), we already include a detailed comparison with prior work comparing BC and offline RL methods.
>
> >> **Condition on expert distribution, is it needed in theory?**
>
> Yes, one may prove a result assuming the expert policy is far away from the base policy and only samples high reward traces. But this is less practically useful: 1) it is unclear how to identify and sample from such an expert; and 2) the expert needs to be close to the base policy to: a) prevent optimization and memorization issues (Kang et al., Tajwar et al.); and b) it is common to collect expert traces by running rejection sampling (Zelikman et al.).
>
> >> **Realizability assm. in Prp 5.5**
>
> Yes, the true reward function is assumed to be present in reward class R. We state this in our setup (L114), and we will highlight this better.
>
> >> **Y-axis in Fig 4**
>
> To cleanly compare performance of policies trained for different values of token length H, we divide the average reward (in [0, H]) that measures test-time compute efficiency by H (L347). Hence, the normalized rewards are in [0,1], and can be viewed as accuracy normalized by the test-time compute budget.

---

> > ### Comment · Reviewer_eEQB · 2025-04-04
> >
> > Thank the authors for their efforts on the rebuttal. The elaborations are very detailed. I admit I have some misunderstanding towards the setting. Besides, my main concerns are satisfactorily solved including relaxing the restrictive definition of anti-concentration in Property 5.6 and showing the anti-concentration condition usually satisfies in practice by additional experiments. With these further evidences, I think the claims made in this work are supported and I am glad to increase my score to 3.

---

> > > ### Author Response · Authors · 2025-04-07
> > >
> > > Thank you for raising the score. We are glad to note that your main concerns are "**satisfactorily solved**", including the relaxation of anti-concentration, and showing that it holds in practice. Please let us know if there are any additional concerns, and **if there are no additional concerns, we would really appreciate if you would consider raising the score further, thank you!**.

---

### Official Review · Reviewer_nSQF · 2025-03-18

**Overall Recommendation:** 3

**Summary:**

- This paper seeks to understand the best way to finetune LLMs that leads to performance that is most scalable in terms of test-time compute
- These are split into VF methods that don't use a verifier (eg SFT on expert trajectories) and VB methods that do (eg RL on 1/0 rewards)
- The focus is on theoretical results. First they define heterogeneity (roughly how diverse is the base policy in terms of q value) and anti-concentration (roughly a high value means there is a lump of probability mass at a reward value much higher than the average value)
- $H$ is test time budget per answer (number of tokens) and $n$ is number of examples for finetuning phases.
- The first main result (theorem 5.4) is sub-optimality of VF methods deteriorates with heterogeneity (which itself scales $\Omega(H)$), which leads to overall bound $H/\sqrt{n}$
- The second is that sub-optimality of VB methods do not, and instead is constant (assuming $n$ is set to be linear in $H$)
- Combining these (theorem 5.8), one gets the gap between VB and VF methods is bounded by $\Omega(H/\sqrt{n})$ where $n$ is set as $n=\Omega(H)$, giving the main result that the gap grows $\sqrt{H}$.
- Two experimental results are presented, a didactic toy distribution (but still using real LLMs) and math reasoning.

**Claims And Evidence:**

fine

**Essential References Not Discussed:**

fine

**Experimental Designs Or Analyses:**

fine

**Methods And Evaluation Criteria:**

fine

**Other Comments Or Suggestions:**

fine

**Other Strengths And Weaknesses:**

Strengths
- I'm sure much of the community would welcome this kind of deeper understanding of differences between LLM finetuning methods and their scalability
- Precisely written, no obvious errors (although I was not able to fully interrogate all mathematical details)
- Good job of making effort to provide intuition and link to practice, rather than solely stating results in mathematical language
- LLM experiments in section 7 were excellent, with an attempt to even measure and visualize heterogeneity and anti-concentration
- Tackling this kind a problem through theoretical analysis is tough, and I feel the authors have done a great job at making progress (though I am only familiar with high-level details of related work so cannot comment too precisely)

Weaknesses
- The paper is dense. The concepts took me a long time to absorb, and there are various lemmas outside of the main results I did not have time to fully process. Less content could be more from a readability perspective.
- The didactic experiment was less informative than the math experiment. The gap between the two methods didn't seem to grow with $\sqrt{H}$ (fig 4 a), and there was little extra intuition I felt I gained from it.
- Anyway these didactic results are supersceded by the math experiments, which end up being cramped and rushed for the sake of space. I might suggest removing section 6 from the main paper altogether to uncompress the technical results and provide readers a deeper dive into section 7.
- Anti-concentration felt like a very specific assumption to require -- intuitively it says we need a large amount of probability mass producing rewards that are far from the average output of the model. It's not a surprising consequence then that RL should be able to seek this mode out.
- I did not follow the intuition for why heterogeneity harms VF methods. Did I miss this in the paper? It seemed like one of the two cornerstones leading to the main result, but I was unable to follow why it would have a detrimental affect.

Questions
- I wonder if the results say anything about the common practice of firstly SFT finetuning followed by RL finetuning. The paper's results suggest the latter is strictly better than the former?

**Questions For Authors:**

see above

**Relation To Broader Scientific Literature:**

see strengths weaknesses

**Theoretical Claims:**

I only looked at main paper content and skimmed some appendix section, but have not interrogated in depth

---

> ### Author Rebuttal · Authors · 2025-04-01
>
> Thank you for the review and a positive assessment of our work! To address your concerns, we added an experiment for the didactic setup where we show an increasing performance gap between VF and VB methods, as we scale both n (the number of prompts) and H (the output length), matching our results on MATH in Fig 5c. We also answer other questions on heterogeneity, anti-concentration, and running RL followed by SFT. We will also use the extra page in the final version to uncompress Sec 7, and if needed shorten Sec 6.
>
> >> **[New Expt.] Gap between VF and VB methods in the didactic setup too scales as $\sqrt{H}$.**
>
> Similar to our MATH experiment (Fig 5c), we plot the performance of VB (RL) and VF (SFT) in our didactic setup here: **https://sites.google.com/view/ttcwv/home**. In Fig 5a we fix the data budget to $n=2^{10}$, and in 5a fix the compute budget $H=2^6$. Now, we scale $n$ with $H$, and note the performance gap between RL and SFT scale superlinearly (as $\sqrt H$), in agreement with our result in Theorem 5.1.
>
> >> **Why does heterogeneity hurt VF methods?**
>
> A VF algorithm (e.g. SFT) finetunes the base LLM on expert traces (e.g. traces collected by rejection sampling correct answers from base LLM). VF methods fail under heterogeneity for the following reasons:
>
> - **Diverse solution traces**: When the base policy (and thus the expert induced by rejection sampling) is heterogeneous, the expert data consists of diverse solution traces for the same problem. Here, each trace gets to the correct answer, spending a varying number of tokens (e.g., a short solution that directly outputs the final answer, and a longer solution that is composed of a failed attempt, followed by a correct one that solves the problem). Thus, the bi-level reward carried by these traces has a high variance for each input.
>
> - **VF methods are forced to equally mimic diverse traces**: VF methods like SFT finetune the base LLM on expert traces where reward annotations are absent. Thus, VF methods cannot distinguish between the diverse expert traces, and are forced to mimic each trace equally, irrespective of their reward. Thus, the VF learned policy fails to only learn high reward solutions. Also, in practice it is challenging to collect multiple traces that are less diverse, i.e., with matching rewards, and naturally the expert data is heterogeneous (as we also show in Sec 7). On the other hand, VB methods like RL have access to reward annotations from a learned verifier or ground truth 0/1 rewards, and need only mimic or increase likelihood on high reward traces, as judged by the verifier.
>
> Note that the verifier for VB is trained on heterogeneous, and suboptimal data (includes incorrect answers) sampled from the base LLM, and the learned verifier is also forced to equally reduce estimation error on each mode of the reward distribution (low and high rewards). But, the base LLM is finetuned to *maximize* the rewards under the learned verifier. So, the heterogeneity of traces sampled from the base policy does not hurt the policy learned by VB methods.
>
> We will add the above discussion after Theorem 5.4 in the final version.
>
> >> **Does running SFT before RL help, as done in practice?**
>
> Yes, you are correct, our results imply that running SFT before RL can help improve coverage of correct solutions under the base LLM. Note that, our performance guarantee (Thm 5.7) for VB methods like RL bounds the suboptimality gap by $O(H/n)$ for the RL trained policy against the best expert in the $\epsilon$-$\chi^2$ ball around the base LLM. Thus, for the RL policy to have good performance in absolute terms, we need to ensure that there exists an LLM in an $\epsilon$-$\chi^2$ ball around the base LLM that performs well on the task. One way of ensuring this is to improve the coverage of the base LLM over high reward traces or at least correct solution traces, and SFT does exactly that. Thank you for this question; we will add the above discussion after Thm 5.7.
>
> >> **Anti-concentration felt like a very specific assumption. It is not surprising RL can seek this mode.**
>
>
> Anti-concentration is a weaker form of the trajectory level coverage assumption made in offline RL literature (e.g., Kumar et al. BC or Offline RL). The functional form of the definition (one std. dev. from mean) stems from the fact that VB methods need to improve over the best expert in $\epsilon$-$\chi^2$ ball around base LLM, and the reward for such an expert assumes this form. Yes, it is not surprising that online RL can seek this mode of higher rewards given access to the ground-truth bi-level reward function. The non-trivial finding of our analysis is that training a learned verifier *using only IID samples from the base policy* and optimizing it with online RL is enough for a policy to seek this mode (Alg. 1). Contrast this with prior works on reward hacking when optimizing learned verifiers (Gao et al.).

---

> > ### Comment · Reviewer_nSQF · 2025-04-03
> >
> > Thank you for this response, particularly the intuition around heterogeity. I will continue the discussion with other reviewers.

---

> > > ### Author Response · Authors · 2025-04-03
> > >
> > > We are glad that the intuition on heterogeneity is more clear. We will certainly add this discussion to the final version. Are there any additional concerns or comments that we can help address? If there are no additional concerns, we would greatly appreciate if you would be willing to increase your score. Thank you.

---

### Decision · Program_Chairs · 2025-05-01

**Decision:**

Accept (spotlight poster)

**Comment:**

This paper focusing on comparing the verifier-based (VB) methods based on RL or search and verifier-free (VF) approaches based on distilling or cloning search traces for LLM finetuning, and it draws conclusion that VB methods are more reliable and scalable when the  test-time compute budget grows.

Reviewers generally agree with the significant of this topic and the conclusion is supported by both theoretical and experimental results.